# HOG-Diff: Higher-Order Guided Diffusion for Graph Generation

**Yiming Huang, Tolga Birdal**
Department of Computing, Imperial College London, UK
{y.huang24, t.birdal}@imperial.ac.uk

## Abstract

Graph generation is a critical yet challenging task, as empirical analyses require a deep understanding of complex, non-Euclidean structures. Diffusion models have recently made significant advances in graph generation, but these models are typically adapted from image generation frameworks and overlook inherent higher-order topology, limiting their ability to capture graph topology. In this work, we propose Higher-order Guided Diffusion (HOG-Diff), a principled framework that progressively generates plausible graphs with inherent topological structures. HOG-Diff follows a coarse-to-fine generation curriculum, guided by higher-order topology and implemented via diffusion bridges. We further prove that our model admits stronger theoretical guarantees than classical diffusion frameworks. Extensive experiments across eight graph generation benchmarks, spanning diverse domains and including large-scale settings, demonstrate the scalability of our method and its superior performance on both pairwise and higher-order topological metrics. Our project page is available here.

## 1 Introduction

Graphs provide an elegant abstraction for representing complex systems by encoding entities as vertices and their relationships as pairwise edges. As such, they have played a key role in generative modeling in unstructured domains, enabling the synthesis of novel graphs faithful to the data distribution. This representational power has positioned graph generative models as a crucial tool for discovering new molecules, materials, and biostructures (Jumper et al., 2021).

Despite these advances, graph generation still lags behind its Euclidean counterparts. Classically, oversmoothing and oversquashing limit expressive capacity (Alon & Yahav, 2021), but an even deeper obstacle persists: most generative frameworks treat graphs purely as collections of pairwise edges, overlooking the higher-order structures that govern the organization of real-world systems, such as triangles, cliques, rings, and motifs (Battiston et al., 2020). These higher-order structures play a decisive role in fields ranging from chemistry to neuroscience: molecules function through coordinated multi-atom assemblies; collaborative teams operate effectively only at specific group sizes; and neural populations exhibit coherent activity patterns that fundamentally rely on multi-unit interactions (Gardner et al., 2022; Papamarkou et al., 2024). A growing body of empirical and theoretical work underscores that such higher-order topology is not an optional structure, but a defining characteristic of real data (Segler et al., 2018; Ertl et al., 2025).

Recent studies make this picture even sharper. Approved drug molecules, for instance, inhabit only a few hundred distinct ring systems (Ertl et al., 2025), far fewer than the astronomical chemical space of plausible compounds ($10^{23}$–$10^{60}$) (Segler et al., 2018). Meanwhile, advances in topological deep learning (TDL) show that explicitly modeling complex structures boosts the expressivity and stability of graph and molecular representation learning (Hajij et al., 2022; Papamarkou et al., 2024; Liu et al., 2024; Wang et al., 2025; Hajij et al., 2025). Nevertheless, no existing graph generative model integrates higher-order structure as an explicit guiding signal.

In this work, motivated by this gap, we propose the **Higher-order Guided Diffusion** (HOG-Diff) framework, a principled, topology-aware framework that places higher-order topology at the core of the generative process. As illustrated in Fig. 1, HOG-Diff implements a coarse-to-fine generation curriculum: it first synthesizes the higher-order skeleton of a graph, its key 2-cells, faces, or other

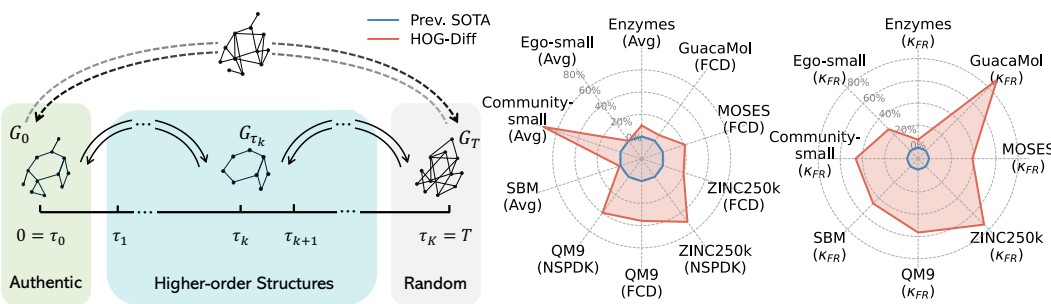

Figure 1: Overview of the **HOG-Diff** framework and its performance. (**Left**) The dashed trajectory above illustrates the classical generation process, where graphs rapidly degrade into structureless randomness. In contrast, as shown in the coloured region below, HOG-Diff adopts a coarse-to-fine generation curriculum via a diffusion bridge, explicitly learning higher-order structures during intermediate steps. (**Middle** and **right**) Relative improvements over prior state-of-the-art on pairwise and higher-order topological metrics.

motifs, and then refines these coarse structures into full pairwise connectivity. To realize this, we combine cell-complex filtering with a novel *generalized Ornstein–Uhlenbeck (OU) diffusion bridge* in the spectral domain, enabling the diffusion process to transition smoothly between increasingly detailed topological states while preserving the global structure implied by higher-order interactions.

This coordinated two-stage design brings several benefits. It aligns the generative trajectory with the intrinsic hierarchical organization of real-world graphs; it avoids the collapse of intermediate states into meaningless noisy adjacency matrices; and it allows us to leverage closed-form bridge dynamics for stable, simulation-free training. Theoretically, we prove that HOG-Diff enjoys faster convergence in score matching and tighter reconstruction error bounds than classical diffusion models. Practically, the explicit presence of topological guides enables interpretability: by varying the guide structures, we can directly probe which topological motifs are most influential in determining the generative process. Across molecular and generic graph benchmarks, HOG-Diff consistently achieves state-of-the-art performance, sharply reducing both statistical and topological discrepancies relative to real data. Our findings underscore a central message: higher-order topology is a powerful generative signal and incorporating it transforms graph diffusion from an edge-level denoising procedure into a truly structure-aware generative paradigm. Our concrete contributions are:

- We introduce *cell complex filtering* to extract higher-order skeletons from graphs as valuable generation guides.
- We propose a principled, coarse-to-fine graph generation framework guided by higher-order topology and implemented via the generalized OU diffusion bridge.
- We theoretically show that HOG-Diff achieves faster convergence during score-matching and a sharper reconstruction error bound compared to classical diffusion models.
- Comprehensive evaluations across eight benchmarks show that topology-informed HOG-Diff achieves state-of-the-art performance on both pairwise and higher-order metrics, highlighting the value of topological guidance.

## 2 PRELIMINARIES

**Higher-order Networks**. Graphs are elegant and useful abstractions for various empirical objects. Formally, a graph can be represented as $G \triangleq (V, E, X)$, where $V$ denotes the node set, $E \subseteq V \times V$ the edges, and $X$ the node feature matrix. However, many empirical systems exhibit group interactions that go beyond simple pairwise relationships (Battiston et al., 2020). To capture these complex interactions, higher-order networks—such as hypergraphs, simplicial complexes (SCs), and cell complexes (CCs)—offer more expressive alternatives by capturing higher-order interactions among multiple entities (Papamarkou et al., 2024). Among these, cell complexes are fundamental in algebraic topology, offering a flexible generalization of pairwise graphs (Hatcher, 2001).

**Definition 1** (Regular cell complex). *A regular cell complex is a topological space $\mathcal{S}$ with a partition into subspaces (cells) $\{x_\alpha\}_{\alpha \in P_{\mathcal{S}}}$, where $P_{\mathcal{S}}$ is an index set, satisfying the following conditions:*

*1. For any $x \in \mathcal{S}$, every sufficiently small neighborhood of $x$ intersects finitely many cells.*
*2. Each cell $x_\alpha$ is homeomorphic to $\mathbb{R}^{n_\alpha}$, where $n_\alpha = \dim(x_\alpha)$ denotes the dimension of $x_\alpha$.*

3. *For each cell $x_\alpha$, the boundary $\partial x_\alpha$ is a finite union of cells of dimension less than $\dim(x_\alpha)$.*
4. *For every $\alpha \in P_S$, there exists a homeomorphism $\phi_\alpha$ of a closed ball $\mathbb{B}^{n_\alpha} \subset \mathbb{R}^{n_\alpha}$ to the closure $\overline{x_\alpha}$ such that the restriction of $\phi_\alpha$ to the interior of the ball is a homeomorphism onto $x_\alpha$.*

**Lifting: From Graphs to Cell Complexes**. A cell complex can be constructed hierarchically through a gluing procedure, which is known as *lifting*. It begins with a set of vertices (0-cells), to which edges (1-cells) are attached by gluing the endpoints of closed line segments, thereby forming a graph. This process can be extended by taking a two-dimensional closed disk and gluing its boundary to a simple cycle in the graph, see Fig. 2 for illustration. While we typically focus on dimensions up to two, this framework can be further generalized by gluing the boundary of $n$-dimensional balls to specific $(n - 1)$-cells in the complex.

From the definition, we can derive that the cell complex $S$ is the union of the interiors of all cells. In this work, we also consider simplicial complexes (SCs), a class of topological spaces represented by finite sets of elements that are closed under the inclusion of subsets. Intuitively, SCs can be viewed as a more constrained subclass of cell complexes, where 2-cells are limited to triangle shapes. A comprehensive introduction to higher-order networks can be found in App. B.1.

**Score-based Diffusion Models**. A fundamental goal of generative models is to produce plausible samples from an unknown target data distribution $p(\mathbf{x}_0)$. Score-based diffusion models (Song & Ermon, 2019; Song et al., 2021) achieve this by progressively corrupting the authentic data with noise and subsequently training a neural network to reverse this corruption process, thereby generating meaningful data from a tractable prior distribution, *i.e.*, $\mathbf{x}_{\text{generated}} \sim p(\mathbf{x}_0)$.

Specifically, the time-dependent forward process of the diffusion model can be described by the following stochastic differential equation (SDE):

$$\mathrm{d}\mathbf{x}_t = \mathbf{f}_t(\mathbf{x}_t)\,\mathrm{d}t + g_t\mathrm{d}\mathbf{w}_t, \tag{1}$$

where $\mathbf{f}_t : \mathbb{R}^n \to \mathbb{R}^n$ is a vector-valued drift function, $g_t : [0, T] \to \mathbb{R}$ is a scalar diffusion coefficient, and $\mathbf{w}_t$ represents a Wiener process. Typically, $p(\mathbf{x}_0)$ evolves over time $t$ from 0 to a sufficiently large $T$ into $p(\mathbf{x}_T)$ through the SDE, such that $p(\mathbf{x}_T)$ will approximate a tractable prior distribution, for example, a standard Gaussian distribution.

Starting from time $T$, $p(\mathbf{x}_T)$ can be progressively transformed back to $p(\mathbf{x}_0)$ by following the trajectory of the reverse-time SDE $\mathrm{d}\mathbf{x}_t = [\mathbf{f}_t(\mathbf{x}_t) - g_t^2 \nabla_{\mathbf{x}_t} \log p_t(\mathbf{x}_t)]\,\mathrm{d}\bar{t} + g_t\mathrm{d}\bar{\mathbf{w}}_t$ (Anderson, 1982), where $p_t(\cdot)$ denotes the probability density function of $\mathbf{x}_t$ and $\bar{\mathbf{w}}$ is a reverse-time Wiener process. The score function $\nabla_{\mathbf{x}_t} \log p_t(\mathbf{x}_t)$ is typically parameterized by a neural network $\boldsymbol{s}_{\boldsymbol{\theta}}(\mathbf{x}_t, t)$ and trained using the conditional score-matching loss function (Vincent, 2011):

$$\ell(\boldsymbol{\theta}) \triangleq \mathbb{E}_{t,\mathbf{x}_t}\left[\omega(t)\,\|\boldsymbol{s}_{\boldsymbol{\theta}}(\mathbf{x}_t, t) - \nabla_{\mathbf{x}_t} \log p_t(\mathbf{x}_t)\|^2\right] \propto \mathbb{E}_{t,\mathbf{x}_0,\mathbf{x}_t}\left[\omega(t)\,\|\boldsymbol{s}_{\boldsymbol{\theta}}(\mathbf{x}_t, t) - \nabla_{\mathbf{x}_t} \log p_t(\mathbf{x}_t|\mathbf{x}_0)\|^2\right],$$

where $\omega(t)$ is a weighting function. The second expression is more commonly used since the conditional probability $p_t(\mathbf{x}_t|\mathbf{x}_0)$ is generally accessible. Ultimately, the generation process is complete by first sampling $\mathbf{x}_T$ from a tractable prior distribution $p(\mathbf{x}_T) \approx p_{\text{prior}}(\mathbf{x})$ and then generating $\mathbf{x}_0$ by numerically solving the reverse-time SDE.

**Doob's $h$-transform**. Doob's $h$-transform is a mathematical framework widely used to modify stochastic processes, enabling the process to satisfy specific terminal conditions. By introducing an $h$-function into the drift term of an SDE, this technique ensures that the process transitions to a predefined endpoint while preserving the underlying probabilistic structure. Specifically, given the SDE in Eq. (1), Doob's $h$-transform alters the SDE to include an additional drift term, ensuring that the process reaches a fixed terminal state at $t = T$. The modified SDE is expressed as:

$$\mathrm{d}\mathbf{x}_t = [\mathbf{f}_t(\mathbf{x}_t) + g_t^2\boldsymbol{h}(\mathbf{x}_t, t, \mathbf{x}_T, T)]\mathrm{d}t + g_t\mathrm{d}\mathbf{w}_t, \tag{2}$$

where $\boldsymbol{h}(\mathbf{x}_t, t, \mathbf{x}_T, T) = \nabla_{\mathbf{x}_t} \log p(\mathbf{x}_T|\mathbf{x}_t)$. Crucially, the construction drives the diffusion process towards a Dirac distribution at $\mathbf{x}_T$, *i.e.*, $\lim_{t \to T} p(\mathbf{x}_t|\mathbf{x}_0, \mathbf{x}_T) = \delta(\mathbf{x}_t - \mathbf{x}_T)$.

## 3 HIGHER-ORDER GUIDED DIFFUSION MODEL

We now present our *Higher-order Guided Diffusion* (HOG-Diff) model, which enhances graph generation by exploiting higher-order structures. We first describe our coarse-to-fine generation framework, followed by an introduction to the supporting diffusion bridge technique. Finally, we provide theoretical evidence to validate the efficacy of HOG-Diff.

### 3.1 COARSE-TO-FINE FRAMEWORK WITH TOPOLOGICAL FILTERING

We draw inspiration from *curriculum learning*, a paradigm that mimics human learning by systematically organizing data in a progression from simple (coarse) to complex (fine) (Abbe et al., 2021; Soviany et al., 2022). Specifically, we model coarse intermediary structures as *higher-order cells*, which encapsulate rich structural properties beyond pairwise interactions (Battiston et al., 2020). These cells can be obtained by *lifting* the original graph and retaining associated 2-faces as the higher-order skeleton. Our generative processes then follow a curriculum to progressively generate graphs, starting with higher-order cells and gradually refining them into the full complex graph.

To implement our coarse-to-fine generation, we first introduce a key operation termed *cell complex filtering* (CCF).

**Proposition 2** (Cell complex filtering). *Given a graph $\boldsymbol{G} = (\boldsymbol{V}, \boldsymbol{E})$ and its associated cell complex $\mathcal{S} = \cup_\alpha x_\alpha$ (obtained via lifting). The p-cell complex filtering operation defines a filtered graph $\boldsymbol{G}_{[p]} = (\boldsymbol{V}_{[p]}, \boldsymbol{E}_{[p]})$, where $\boldsymbol{V}_{[p]} = \{v \in \boldsymbol{V} \mid \exists\, x_\alpha \text{with} \dim(x_\alpha) = p : v \in \overline{x_\alpha}\}$, and $\boldsymbol{E}_{[p]} = \{(u,v) \in \boldsymbol{E} \mid \exists\, x_\alpha \text{with} \dim(x_\alpha) = p : \{u,v\} \subseteq \overline{x_\alpha}\}$. Here, $\overline{x_\alpha}$ denotes the closure of $x_\alpha$.*

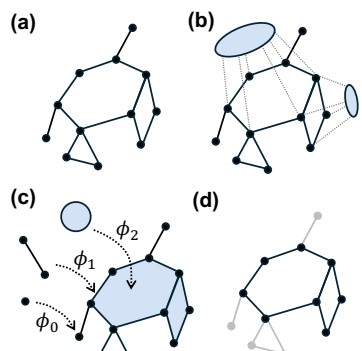

As illustrated in Fig. 2, we first lift the graph to a cell complex and then apply CCF to generate intermediate states by pruning nodes and edges that do not belong to higher-order cells. In practice, CCF offers a substantial speedup, as it avoids the expensive enumeration of all cells required by lifting (see App. D for detailed complexity analysis). The filtering operation decomposes the graph generation task into hierarchically structured and manageable subtasks.

Figure 2: Cell Complex transformations. (**a**) An example graph. (**b**) Lifting: closed 2D disks are glued to the boundary of the rings to form the 2-cell complex. (**c**) The resulting cell complex and the corresponding homeomorphisms to the closed balls for three representative cells of different dimensions in the complex. (**d**) Black elements represent higher-order structures extracted through 2-cell filtering, while grey elements denote corresponding peripheral structures pruned by the filtering operation.

Based on the filtering results, the diffusion process is structured into $K$ hierarchical time windows, denoted as $\{[\tau_{k-1}, \tau_k]\}_{k=1}^K$, where $0 = \tau_0 < \cdots < \tau_{k-1} < \tau_k < \cdots < \tau_K = T$, with the filtered results serving as natural intermediaries in the hierarchical generation process. The overall framework of HOG-Diff is depicted in Fig. 1(left). In general, we first generate coarse-grained higher-order skeletons and subsequently refine them into finer pairwise relationships, thereby simplifying the task of capturing complex graph distributions. Formally, our generation process factorizes the joint distribution of the final graph $\boldsymbol{G}_0$ into a product of conditional distributions across these time windows:

$$p(\boldsymbol{G}_0) = p(\boldsymbol{G}_0|\boldsymbol{G}_{\tau_1})p(\boldsymbol{G}_{\tau_1}|\boldsymbol{G}_{\tau_2})\cdots p(\boldsymbol{G}_{\tau_{K-1}}|\boldsymbol{G}_T). \tag{3}$$

Here, the intermediate states $\boldsymbol{G}_{\tau_{K-1}}, \cdots, \boldsymbol{G}_{\tau_2}, \boldsymbol{G}_{\tau_1}$ represent progressively finer cell complex filtered graph representations, aligning intermediate diffusion stages with realistic hierarchical graph structures. This coarse-to-fine approach enables our model to first focus on fundamental topological structures and then add finer connectivity, inherently aligning with the hierarchical nature of many empirical systems. Consequently, our model benefits from smoother training and improved sampling performance (see Sec. 3.3 for theoretical analysis).

To ensure smooth transitions between intermediate states within each interval $[\tau_{k-1}, \tau_k]$, the graph evolves according to the general form of a diffusion bridge process (see Sec. 3.2 for details):

$$d\boldsymbol{G}_t^{(k)} = \mathbf{f}_{k,t}(\boldsymbol{G}_t^{(k)})dt + g_{k,t}d\boldsymbol{W}_t, t \in [\tau_{k-1}, \tau_k]. \tag{4}$$

The forward diffusion process introduces noise in a stepwise manner while preserving intermediate structural information. Reversing this process enables the model to generate authentic samples with desirable higher-order information. Moreover, integrating higher-order structures into graph generative models improves interpretability by allowing analysis of their significance in shaping the graph's properties. Rather than directly conditioning on higher-order information, HOG-Diff employs it incrementally as a guiding structure. This strategy allows the model to build complex graph structures progressively, while maintaining meaningful structural integrity at each stage.

## 3.2 Guided Generation via Diffusion Bridge Process

As a building block of our generative framework, we leverage a class of diffusion processes with fixed terminal states, namely the *generalized Ornstein-Uhlenbeck (GOU) bridge*, to realize the proposed guided diffusion process in Eq. (4), while enabling simulation-free training.

The generalized Ornstein-Uhlenbeck (GOU) process (Ahmad, 1988; Luo et al., 2023b), also known as the time-varying OU process, is a stationary, Gaussian-Markov process characterized by its mean-reverting property. Specifically, the marginal distribution of the GOU process asymptotically approaches a fixed mean and variance. The GOU process is governed by the following SDE:

$$\mathrm{d}\boldsymbol{G}_t = \theta_t(\boldsymbol{\mu} - \boldsymbol{G}_t)\mathrm{d}t + g_t\mathrm{d}\boldsymbol{W}_t, \tag{5}$$

where $\boldsymbol{\mu}$ is the target terminal state, $\theta_t$ denotes a scalar drift coefficient and $g_t$ represents the diffusion coefficient. For analytical tractability, $\theta_t$ and $g_t$ are constrained by $g_t^2/\theta_t = 2\sigma^2$ (Luo et al., 2023b), where $\sigma^2$ is a fixed constant, yielding a closed-form transition probability:

$$p(\boldsymbol{G}_t \mid \boldsymbol{G}_s) = \mathcal{N}(\mathbf{m}_{s:t}, v_{s:t}^2\boldsymbol{I}) = \mathcal{N}\left(\boldsymbol{\mu} + (\boldsymbol{G}_s - \boldsymbol{\mu})\,e^{-\bar{\theta}_{s:t}}, \sigma^2(1 - e^{-2\bar{\theta}_{s:t}})\boldsymbol{I}\right). \tag{6}$$

Here, $\bar{\theta}_{s:t} = \int_s^t \theta_z dz$. For notional simplicity, $\bar{\theta}_{0:t}$ is replaced by $\bar{\theta}_t$ when $s = 0$.

**Diffusion Bridge**. Applying Doob's $h$-transform (Doob & Doob, 1984) to the GOU process under the terminal condition $\boldsymbol{\mu} = \boldsymbol{G}_{\tau_k}$, we can derive the GOU bridge process as follows (detailed derivation of the bridge process provided in App. A.1):

$$\mathrm{d}\boldsymbol{G}_t = \theta_t\left(1 + \frac{2}{e^{2\bar{\theta}_{t:\tau_k}} - 1}\right)(\boldsymbol{G}_{\tau_k} - \boldsymbol{G}_t)\mathrm{d}t + g_{k,t}\mathrm{d}\boldsymbol{W}_t. \tag{7}$$

The conditional transition probability admits an analytical form $p(\boldsymbol{G}_t|\boldsymbol{G}_{\tau_{k-1}}, \boldsymbol{G}_{\tau_k}) = \mathcal{N}(\bar{\mathbf{m}}_t, \bar{v}_t^2\boldsymbol{I})$:

$$\bar{\mathbf{m}}_t = \boldsymbol{G}_{\tau_k} + (\boldsymbol{G}_{\tau_{k-1}} - \boldsymbol{G}_{\tau_k})e^{-\bar{\theta}_{\tau_{k-1}:t}}\frac{v_{t:\tau_k}^2}{v_{\tau_{k-1}:\tau_k}^2}, \quad \bar{v}_t^2 = v_{\tau_{k-1}:t}^2 v_{t:\tau_k}^2 / v_{\tau_{k-1}:\tau_k}^2. \tag{8}$$

Here, $\bar{\theta}_{a:b} = \int_a^b \theta_s \mathrm{d}s$, and $v_{a:b} = \sigma^2(1 - e^{-2\bar{\theta}_{a:b}})$.

The GOU bridge process eliminates variance in the terminal state by directing the diffusion toward a Dirac distribution centered at $\boldsymbol{G}_{\tau_k}$, making it well-suited for stochastic modelling with terminal constraints (Heng et al., 2021; Yue et al., 2024). Moreover, we can directly use the closed-form solution for one-step forward sampling without expensive SDE simulation. Note that the Brownian bridge process used in previous works (Wu et al., 2022) is a special case of the GOU bridge process when $\theta_t \to 0$ (see App. A.1).

**Training and Sampling**. Classical graph diffusion approaches typically inject isotropic Gaussian noise directly into the adjacency matrices $\boldsymbol{A}$, leading to various fundamental challenges, such as permutation ambiguity, sparsity-induced signal degradation, and poor scalability (see App. C.2 for detailed discussion). To address these challenges, inspired by Luo et al. (2023a), we introduce noise in the spectra of the graph Laplacian $\boldsymbol{L} = \boldsymbol{D} - \boldsymbol{A}$, instead of the adjacency matrix $\boldsymbol{A}$, where $\boldsymbol{D}$ denotes the diagonal degree matrix. As a symmetric positive semi-definite matrix, the graph Laplacian can be diagonalized as $\boldsymbol{L} = \boldsymbol{U}\boldsymbol{\Lambda}\boldsymbol{U}^{\top}$. Here, the orthogonal matrix $\boldsymbol{U} = [\boldsymbol{u}_1, \cdots, \boldsymbol{u}_n]$ comprises the eigenvectors, and the diagonal matrix $\boldsymbol{\Lambda} = \mathrm{diag}(\lambda_1, \cdots, \lambda_n)$ holds the corresponding eigenvalues. Therefore, the target graph distribution $p(\boldsymbol{G}_0)$ represents a joint distribution of $\boldsymbol{X}_0$ and $\boldsymbol{\Lambda}_0$, exploiting the permutation invariance and structural robustness of the Laplacian spectrum.

Consequently, the GOU bridge process in Eq. (7), along with its time-reversed counterpart, can be formulated as the following system of SDEs for graph $\boldsymbol{G}$:

$$\begin{cases} \mathrm{d}\boldsymbol{X}_t = \mathbf{f}_{k,t}(\boldsymbol{X}_t)\mathrm{d}t + g_{k,t}\mathrm{d}\bar{\boldsymbol{W}}_t^1 \\ \mathrm{d}\boldsymbol{\Lambda}_t = \mathbf{f}_{k,t}(\boldsymbol{\Lambda}_t)\mathrm{d}t + g_{k,t}\mathrm{d}\bar{\boldsymbol{W}}_t^2 \end{cases}, \begin{cases} \mathrm{d}\boldsymbol{X}_t = \left[\mathbf{f}_{k,t}(\boldsymbol{X}_t) - g_{k,t}^2\nabla_{\boldsymbol{X}}\log p_t(\boldsymbol{G}_t|\boldsymbol{G}_{\tau_k})\right]\mathrm{d}\bar{t} + g_{k,t}\mathrm{d}\bar{\boldsymbol{W}}_t^1 \\ \mathrm{d}\boldsymbol{\Lambda}_t = \left[\mathbf{f}_{k,t}(\boldsymbol{\Lambda}_t) - g_{k,t}^2\nabla_{\boldsymbol{\Lambda}}\log p_t(\boldsymbol{G}_t|\boldsymbol{G}_{\tau_k})\right]\mathrm{d}\bar{t} + g_{k,t}\mathrm{d}\bar{\boldsymbol{W}}_t^2 \end{cases}.$$

Here, the reverse-time dynamics of the bridge process are derived using the theory of SDEs, the superscript of $\boldsymbol{X}_t^{(k)}$ and $\boldsymbol{\Lambda}_t^{(k)}$ are dropped for simplicity, and $\mathbf{f}_{k,t}$ is determined according to Eq. (7).

To approximate the score functions $\nabla_{\boldsymbol{X}_t} \log p_t(\boldsymbol{G}_t|\boldsymbol{G}_{\tau_k})$ and $\nabla_{\boldsymbol{\Lambda}_t} \log p_t(\boldsymbol{G}_t|\boldsymbol{G}_{\tau_k})$, we employ a neural network $\boldsymbol{s}_{\boldsymbol{\theta}}^{(k)}(\boldsymbol{G}_t, \boldsymbol{G}_{\tau_k}, t)$, which outputs predictions for both node-level ($\boldsymbol{s}_{\boldsymbol{\theta},\boldsymbol{X}}^{(k)}(\boldsymbol{G}_t, \boldsymbol{G}_{\tau_k}, t)$) and spectrum ($\boldsymbol{s}_{\boldsymbol{\theta},\boldsymbol{\Lambda}}^{(k)}(\boldsymbol{G}_t, \boldsymbol{G}_{\tau_k}, t)$) components. The network is optimized by minimizing:

$$\ell^{(k)}(\boldsymbol{\theta}) = \mathbb{E}_{t,\boldsymbol{G}_t,\boldsymbol{G}_{\tau_{k-1}},\boldsymbol{G}_{\tau_k}} \{\omega(t)[c_1\|\boldsymbol{s}_{\boldsymbol{\theta},\boldsymbol{X}}^{(k)} - \nabla_{\boldsymbol{X}} \log p_t(\boldsymbol{G}_t|\boldsymbol{G}_{\tau_k})\|_2^2 + c_2\|\boldsymbol{s}_{\boldsymbol{\theta},\boldsymbol{\Lambda}}^{(k)} - \nabla_{\boldsymbol{\Lambda}} \log p_t(\boldsymbol{G}_t|\boldsymbol{G}_{\tau_k})\|_2^2]\}, \tag{9}$$

where $\omega(t)$ is a positive weighting function, and $c_1, c_2$ control the relative importance of vertices and spectrum. The training procedure is detailed in Alg. 1 in App. C.

In the inference procedure, we sample $(\hat{\boldsymbol{X}}_{\tau_K}, \hat{\boldsymbol{\Lambda}}_{\tau_K})$ from the prior distribution and select $\hat{\boldsymbol{U}}_0$ as an eigenbasis drawn from the training set. Reverse diffusion is then applied across multiple stages to sequentially generate $(\hat{\boldsymbol{X}}_{\tau_{K-1}}, \hat{\boldsymbol{\Lambda}}_{\tau_{K-1}}), \cdots, (\hat{\boldsymbol{X}}_{\tau_1}, \hat{\boldsymbol{\Lambda}}_{\tau_1}), (\hat{\boldsymbol{X}}_0, \hat{\boldsymbol{\Lambda}}_0)$, where each stage is implemented via the diffusion bridge and initialized from the output of the previous step. Finally, plausible samples with higher-order structures can be reconstructed as $\hat{\boldsymbol{G}}_0 = (\hat{\boldsymbol{X}}_0, \hat{\boldsymbol{L}}_0 = \hat{\boldsymbol{U}}_0\hat{\boldsymbol{\Lambda}}_0\hat{\boldsymbol{U}}_0^\top)$. Further details of the spectral diffusion process and the complete sampling procedure are provided in App. C, while ablation studies comparing diffusion in the spectral domain versus the adjacency matrix are presented in App. F.3.

**Score Network Architecture**. The score network plays a critical role in estimating the score functions required to reverse the diffusion process. Standard graph neural networks designed for classical tasks such as graph classification and link prediction may be inappropriate for graph distribution learning due to the complicated requirements. For example, an effective model for molecular graph generation should capture local node-edge dependence for chemical valency rules and attempt to recover global graph patterns like edge sparsity, frequent ring subgraphs, and atom-type distribution.

To achieve this, we introduce a unified score network that explicitly integrates node and spectral representations. As illustrated in Fig. 7 in the Appendix, the network comprises two different graph processing modules: a standard graph convolution network (GCN) (Kipf & Welling, 2017) for local feature aggregation and a graph transformer network (ATTN) (Dwivedi & Bresson, 2021; Vignac et al., 2023) for global information extraction. The outputs of these modules are fused with time information through a Feature-wise Linear Modulation (FiLM) layer (Perez et al., 2018), and the resulting representations are concatenated to form a unified hidden embedding. This hidden embedding is further processed by separate multilayer perceptrons (MLPs) to produce predictions for $\nabla_{\boldsymbol{X}} \log p(\boldsymbol{G}_t|\boldsymbol{G}_{\tau_k})$ and $\nabla_{\boldsymbol{\Lambda}} \log p(\boldsymbol{G}_t|\boldsymbol{G}_{\tau_k})$, respectively. It is worth noting that our score network is permutation equivalent, as each component of our model avoids any node ordering-dependent operations. Our model is detailed in App. C.3.

### 3.3 THEORETICAL ANALYSIS

We now provide theoretical evidence for the efficacy of HOG-Diff, demonstrating that the proposed framework achieves faster convergence in score-matching and tighter reconstruction error bounds compared to standard graph diffusion. We experimentally verify our theories in Sec. 4.4.

**Theorem 3** (Informal). *Suppose the loss function $\ell^{(k)}(\boldsymbol{\theta})$ in Eq. (9) is $\beta$-smooth and satisfies the $\mu$-PL condition in the ball $B(\boldsymbol{\theta}_0, R)$. Then, the expected loss at the $i$-th training iteration satisfies:*

$$\mathbb{E}\left[\ell^{(k)}(\boldsymbol{\theta}_i)\right] \leq \left(1 - \frac{b\mu^2}{\beta N(\beta N^2 + \mu(b-1))}\right)^i \ell^{(k)}(\boldsymbol{\theta}_0),$$

*where $N$ denotes the size of the training dataset, and $b$ is the mini-batch size. Furthermore, it holds that $\beta_{\text{HOG-Diff}} \leq \beta_{\text{classical}}$, implying that the distribution learned by the proposed framework converges to the target distribution faster than classical generative models.*

Following Luo et al. (2023a), we define the expected reconstruction error at each generation process as $\mathcal{E}(t) = \mathbb{E}\left\|\bar{\boldsymbol{G}}_t - \hat{\boldsymbol{G}}_t\right\|^2$, where $\bar{\boldsymbol{G}}_t$ represents the data reconstructed with the ground truth score $\nabla \log p_t(\cdot)$ and $\hat{\boldsymbol{G}}_t$ denotes the data reconstructed with the learned score function $\boldsymbol{s}_{\boldsymbol{\theta}}$. Next, we establish that the reconstruction error in HOG-Diff is bounded more tightly than in classical graph generation models, thereby ensuring superior sample quality.

Table 1: Comparison of different methods on molecular datasets. The **best** results are highlighted in bold. The full version with more baselines, metrics, and standard deviations is provided in App. F.1.

| Method | QM9 | | | | | ZINC250k | | | | |
|---|---|---|---|---|---|---|---|---|---|---|
| | Val. ↑ | Uni. ↑ | Nov. ↑ | FCD ↓ | NSPDK ↓ | Val. ↑ | Uni. ↑ | Nov. ↑ | FCD ↓ | NSPDK ↓ |
| GraphAF | 74.43 | 88.64 | 86.59 | 5.625 | 0.021 | 68.47 | 98.64 | 99.99 | 16.023 | 0.044 |
| GraphDF | 93.88 | 98.58 | 98.54 | 10.928 | 0.064 | 90.61 | 99.63 | 100.00 | 33.546 | 0.177 |
| GraphArm | 90.25 | 95.62 | 70.39 | 1.220 | 0.002 | 88.23 | 99.46 | 100.00 | 16.260 | 0.055 |
| MiCaM | **99.93** | 93.89 | 83.25 | 1.045 | 0.001 | **100.00** | 88.48 | 99.98 | 31.495 | 0.166 |
| SPECTRE | 87.30 | 35.70 | **97.28** | 47.960 | 0.163 | 90.20 | 67.05 | 100.00 | 18.440 | 0.109 |
| GSDM | 99.90 | - | - | 2.650 | 0.003 | 92.70 | - | - | 12.956 | 0.017 |
| EDP-GNN | 47.52 | 99.25 | 86.58 | 2.680 | 0.005 | 82.97 | 99.79 | 100.00 | 16.737 | 0.049 |
| GDSS | 95.72 | 98.46 | 86.27 | 2.900 | 0.003 | 97.01 | 99.64 | 100.00 | 14.656 | 0.019 |
| DiGress | 99.00 | 96.66 | 33.40 | 0.360 | 0.0005 | 91.02 | 81.23 | 100.00 | 23.060 | 0.082 |
| MoFlow | 91.36 | 98.65 | 94.72 | 4.467 | 0.017 | 63.11 | 99.99 | 100.00 | 20.931 | 0.046 |
| Cometh | 99.59 | 96.75 | 72.06 | 0.248 | 0.0005 | - | - | - | - | - |
| CatFlow | 99.81 | **99.95** | - | 0.441 | - | 99.95 | **99.99** | - | 13.211 | - |
| DeFoG | 99.26 | 96.61 | 72.57 | 0.268 | 0.0005 | 94.97 | 99.98 | 100.00 | 2.030 | 0.002 |
| **HOG-Diff** | 98.74 | 97.10 | 75.12 | **0.172** | **0.0003** | 98.56 | 99.96 | 99.53 | **1.633** | **0.001** |

| | Method | Val.↑ | Uni.↑ | Nov.↑ | FCD↓ | Filters↑ |
|---|---|---|---|---|---|---|
| MOSES | DiGress | 85.7 | **100.0** | 95.0 | 1.19 | 97.1 |
| | DisCo | 88.3 | **100.0** | **97.7** | 1.44 | 95.6 |
| | Cometh | 90.5 | 99.9 | 92.6 | 1.27 | **99.1** |
| | DeFoG | 92.8 | 99.9 | 92.1 | 1.95 | 98.9 |
| | **HOG-Diff** | **99.7** | **100.0** | 89.1 | **0.94** | 96.7 |

| | Method | Val.↑ | Uni. ↑ | Nov.↑ | FCD score↑ | KL div.↑ |
|---|---|---|---|---|---|---|
| GuacaMol | DiGress | 85.2 | **100.0** | 99.9 | 68.0 | 92.9 |
| | DisCo | 86.6 | 86.6 | 86.5 | 59.7 | 92.6 |
| | Cometh | 98.9 | 98.9 | 97.6 | 72.7 | 96.7 |
| | DeFoG | 99.0 | 99.0 | 97.9 | 73.8 | **97.7** |
| | **HOG-Diff** | 99.1 | **100.0** | 97.0 | **78.5** | 96.9 |

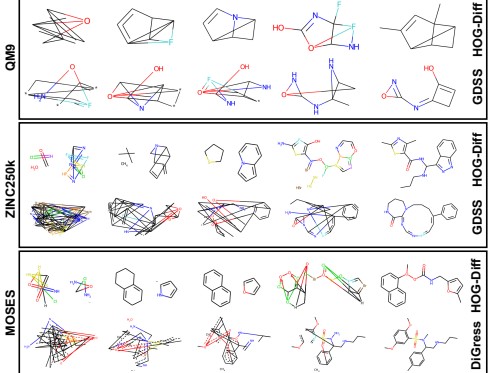

Figure 3: (**left**) Large molecule generation performance. **Best** results are highlighted in bold. Full versions are presented in App. F.2. (**right**) Generative trajectories progressing from left to right, illustrating that HOG-Diff preserves topological integrity across intermediate steps.

**Theorem 4.** *Under standard Lipschitz and boundedness assumptions, the reconstruction error of HOG-Diff satisfies the following bound:*

$$\mathcal{E}_{\text{hog}}(0) \leq A_{\text{hog}}\, e^{T_{\text{hog}}}, \tag{10}$$

*where* $A_{\text{hog}} = C^2 \sum_{k=1}^{K} \ell^{(k)}(\boldsymbol{\theta}) \int_{\tau_{k-1}}^{\tau_k} g_{k,s}^4 \mathrm{d}s$ *and* $T_{\text{hog}} = \sum_{k=1}^{K} \int_{\tau_{k-1}}^{\tau_k} \gamma_k(s)\mathrm{d}s + C \sum_{k=1}^{K-1} \int_{\tau_{k-1}}^{\tau_k} h_{k,s}^2 \mathrm{d}s$. *Moreover, under the same assumptions, HOG-Diff admits a tighter reconstruction error bound than classical diffusion models.*

The theorems above rely primarily on mild assumptions, such as smoothness and boundedness, without imposing strict conditions like the target distribution being log-concave or satisfying the log-Sobolev inequality. Their formal statements and detailed proofs are postponed to App. A. We experimentally verify these Theorems in Sec. 4.4.

## 4 EXPERIMENTS

We assess HOG-Diff against state-of-the-art baselines in both molecular and generic graph generation. Ablation studies are further conducted to analyze the impact of different topological guides. Complexity analysis and experimental settings are deferred to Appendices D and E, while App. F reports additional results, including scalability to large benchmarks, variance statistics, diffusion-domain analysis, the rationale for filtering choices, and visualizations.

### 4.1 MOLECULE GENERATION

Molecular design is a prominent application of graph generation. We evaluate our method on four widely used molecular benchmarks: QM9 (Ramakrishnan et al., 2014), ZINC250k (Irwin et al.,

Table 2: Generation performance on generic graph datasets. Best **bold** and second-best underlined. Hyphen (-) indicates missing results in the original paper.

| Method | Community-small | | | | Enzymes | | | | Ego-small | | | |
|---|---|---|---|---|---|---|---|---|---|---|---|---|
| | Deg.↓ | Clus.↓ | Orbit↓ | Avg.↓ | Deg.↓ | Clus.↓ | Orbit↓ | Avg.↓ | Deg.↓ | Clus.↓ | Orbit↓ | Avg.↓ |
| GraphRNN | 0.080 | 0.120 | 0.040 | 0.080 | 0.017 | 0.062 | 0.046 | 0.042 | 0.090 | 0.220 | 0.003 | 0.104 |
| GraphAF | 0.180 | 0.200 | 0.020 | 0.133 | 1.669 | 1.283 | 0.266 | 1.073 | 0.030 | 0.110 | **0.001** | 0.047 |
| GraphDF | 0.060 | 0.120 | 0.030 | 0.070 | 1.503 | 1.061 | 0.202 | 0.922 | 0.040 | 0.130 | 0.010 | 0.060 |
| GraphVAE | 0.350 | 0.980 | 0.540 | 0.623 | 1.369 | 0.629 | 0.191 | 0.730 | 0.130 | 0.170 | 0.050 | 0.117 |
| GNF | 0.200 | 0.200 | 0.110 | 0.170 | - | - | - | - | 0.030 | 0.100 | **0.001** | 0.044 |
| EDP-GNN | 0.053 | 0.144 | 0.026 | 0.074 | 0.023 | 0.268 | 0.082 | 0.124 | 0.052 | 0.093 | 0.007 | 0.051 |
| GPrinFlowNet | 0.021 | 0.068 | 0.021 | 0.037 | 0.021 | 0.088 | 0.009 | 0.039 | - | - | - | - |
| SPECTRE | 0.048 | 0.049 | 0.016 | 0.038 | 0.136 | 0.195 | 0.125 | 0.152 | 0.078 | 0.078 | 0.007 | 0.054 |
| GDSS | 0.045 | 0.086 | 0.007 | 0.046 | 0.026 | **0.061** | 0.009 | 0.032 | 0.021 | **0.024** | 0.007 | 0.017 |
| DiGress | 0.047 | 0.041 | 0.026 | 0.038 | **0.004** | 0.083 | **0.002** | 0.030 | **0.015** | 0.029 | 0.005 | **0.016** |
| **HOG-Diff** | **0.006** | **0.022** | **0.002** | **0.010** | 0.011 | **0.061** | 0.007 | **0.027** | **0.015** | 0.027 | 0.004 | **0.016** |

2012), and the large-scale MOSES (Polykovskiy et al., 2020) and GuacaMol (Brown et al., 2019) benchmarks. In contrast to many prior works that report results on only a subset of these datasets, our evaluation provides comprehensive coverage. Intermediate higher-order skeletons are extracted using 2-cell complex filtering; the rationale for this choice is discussed in App. F.4. For QM9 and ZINC250k, we follow the dataset split and evaluation protocols of Jo et al. (2022). For MOSES and GuacaMol, we adopt their official splits and benchmark-specific metrics. We report Validity without correction (Val.), Uniqueness (Uni.), Novelty (Nov.), Fréchet ChemNet Distance (FCD) (Preuer et al., 2018), NSPDK (Costa & Grave, 2010), Filters, and KL divergence (Brown et al., 2019).

We benchmark against a diverse set of representative molecular generators. Autoregressive approaches include GraphAF (Shi et al., 2020), GraphDF (Luo et al., 2021), GraphArm (Kong et al., 2023), and the fragment-based MiCaM (Geng et al., 2023). The remaining methods adopt a one-shot paradigm: SPECTRE (Martinkus et al., 2022) incorporates spectral conditioning within a GAN framework, while GDSS (Jo et al., 2022), DiGress (Vignac et al., 2023), and Cometh (Siraudin et al., 2025) are diffusion-based models. We also compare against advanced flow-based models, namely MoFlow (Zang & Wang, 2020), CatFlow (Eijkelboom et al., 2024), and DeFoG (Qin et al., 2025).

**Sampling Quality.** Tab. 1 and Fig. 3(left) show that HOG-Diff generally outperforms both autoregressive and one-shot models. Notably, the dramatic decrease in NSPDK and FCD implies that HOG-Diff is able to generate molecules with data distributions close to those of real molecules in both graph and chemical spaces. We further visualize the molecule generation process in Fig. 3(right) with more examples deferred to App. F.5. It can be observed that our model explicitly preserves higher-order structures during the generation process.

## 4.2 GENERIC GRAPH GENERATION

To display the topology distribution learning ability, we assess HOG-Diff over four common generic graph datasets: (1) Community-small, containing 100 random community graphs; (2) Ego-small, comprising 200 small ego graphs derived from the Citeseer network dataset; (3) Enzymes, featuring 587 protein graphs representing tertiary structures of enzymes from the BRENDA database; and (4) SBM, a larger-scale stochastic block model dataset. The SBM benchmark is presented separately in App. F.2 due to its distinct evaluation protocol. Intermediate higher-order skeletons are obtained via 3-simplicial complex filtering. We employ the same train/test split as Jo et al. (2022) for a fair comparison. Maximum mean discrepancy (MMD) is used to quantify the distribution differences across key graph statistics, including degree (Deg.), clustering coefficient (Clus.), and 4-node orbit counts (Orbit). A low MMD signifies a close alignment between the generated and evaluation datasets, suggesting superior generative performance. We also report the average MMD across all metrics as an overall indicator.

We compare the following graph generative models: GraphRNN (You et al., 2018) is an autoregressive model, while GraphVAE (Simonovsky & Komodakis, 2018), GNF (Liu et al., 2019), and GPrinFlowNet (Mo et al., 2024) are one-shot models. GraphAF, GraphDF, EDP-GNN, SPECTRE, GDSS, and DiGress were introduced previously. The results in Tab. 2 show that HOG-Diff is not only suitable for molecular generation but also proficient in generic graph generation, demonstrating its ability to capture the intricate topological interdependencies effectively.

## 4.3 TOPOLOGICAL PRESERVATION ANALYSIS

To further verify the capability of HOG-Diff in preserving higher-order structures, we conduct a quantitative evaluation using *Curvature Filtrations* (Southern et al., 2023). Unlike simple graph statistics, curvature filtrations combine discrete curvature notions with topological data analysis (TDA) to capture multi-scale topological features.

Specifically, we measure the distance between the mean persistence landscapes of generated graphs and the test set, using two filtration functions: (1) Balanced Forman–Ricci curvature $\kappa_{FR}$, which focuses on edge-based local clustering and cycles, and (2) Ollivier–Ricci curvature $\kappa_{OR}$, which captures global geometry and transport properties via Wasserstein distance. A lower distance indicates a generated distribution that is topologically closer to the test set.

Fig. 1 (right) reports the $\kappa_{FR}$ results, while Tab. 3 provides the full evaluation for both $\kappa_{FR}$ and $\kappa_{OR}$ across all molecular and generic datasets. HOG-Diff consistently achieves the lowest distance scores across both molecular and generic graph datasets. Notably, on complex molecular datasets, our method outperforms strong baselines by a significant margin, suggesting that while baseline models may capture basic chemical validity, HOG-Diff is superior in reconstructing the intrinsic topological backbone and higher-order geometric relations. Even on generic graphs like SBM and Enzymes, which possess distinct community structures, HOG-Diff maintains better topological fidelity, verifying that the performance improvement stems from the successful preservation of higher-order structures.

Table 3: Quantitative evaluation of higher-order topology preservation using Curvature Filtrations. The **best** results are in **bold**.

| Dataset | Method | $\kappa_{FR}$ | $\kappa_{OR}$ |
|---|---|---|---|
| QM9 | GDSS | 0.925 | 0.601 |
| | DiGress | 0.251 | 0.343 |
| | DeFoG | 0.177 | 0.286 |
| | Cometh | 0.216 | 0.314 |
| | **HOG-Diff** | **0.077** | **0.098** |
| ZINC 250k | MiCaM | 9.436 | 7.251 |
| | GDSS | 1.781 | 1.331 |
| | DeFoG | 0.728 | 0.498 |
| | **HOG-Diff** | **0.190** | **0.098** |
| MOSES | DiGress | 0.260 | 0.223 |
| | **HOG-Diff** | **0.159** | **0.183** |
| GuacaMol | DiGress | 0.862 | 0.075 |
| | **HOG-Diff** | **0.083** | **0.057** |
| SBM | DeFoG | 3.998 | 5.206 |
| | Cometh | 2.748 | 3.938 |
| | **HOG-Diff** | **1.453** | **3.740** |
| Comm. -small | GDSS | 12.515 | 14.949 |
| | **HOG-Diff** | **6.734** | **6.522** |
| Enzymes | GDSS | 13.031 | 12.399 |
| | **HOG-Diff** | **12.114** | **9.830** |
| Ego -small | GDSS | 2.311 | 1.301 |
| | **HOG-Diff** | **1.679** | **1.042** |

## 4.4 ABLATIONS: TOPOLOGICAL GUIDE ANALYSIS

Throughout the experiments, we observe that HOG-Diff exhibits superior performance on complex molecular datasets, but comparatively modest results on the Ego-small dataset. Statistics and visualizations in Apps. E and F.5 reveal that Ego contains the fewest higher-order structures among the datasets analyzed, suggesting that the choice of guide plays a pivotal role in the effectiveness of generation. To validate this hypothesis, we conduct further ablations using different types of topological information as guides.

We first compare two types of guides: structures derived from 2-cell filtering (Cell) and Gaussian random noise (Noise). Employing noise as a guide aligns with classical diffusion paradigms that generate samples by progressively denoising noisy data. Fig. 4 visualizes how the spectrum loss changes during the training process. It shows that our framework (red curve) converges faster than the classical method (blue curve), which is consistent with our theoretical results in Theorem 3.

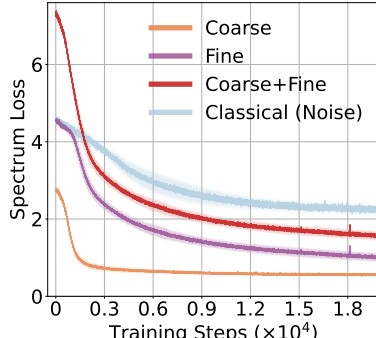

Figure 4: Training curves of the score-matching process. The entire process of HOG-Diff is divided into two stages, *i.e.*, $K = 2$, referred to as coarse and fine, respectively. The combined loss of these two stages is labelled as Coarse+Fine.

In the sampling procedure, we further evaluate peripheral structures (Periph.), obtained by removing cell components, as guides. As shown in Tab. 4, both peripheral and noise guides underperform cell-based guides, providing empirical support for the tighter reconstruction error bound established in Theorem 4. These results indicate that certain topologies, particularly cells, are more effective in guiding generation, highlighting the importance of selecting appropriate topological structures to

steer toward meaningful outputs. Moreover, this finding suggests that guides could serve as diagnostic tools for assessing whether specific topologies are essential to a given architecture. Systematic analysis of different guides promises to enhance interpretability, clarify how topologies affect generation, and inform the design of more effective graph generative models.

## 5 RELATED WORK

We review graph and higher-order generation methods, with a more detailed discussion in App. B.

**Graph Generative Models**. Early work on graph generative models dates back to random network models (Barabási & Albert, 1999), which offer foundational insights but are too simplistic for capturing real-world graph distributions. Recent advances in generative models have leveraged the power of deep neural networks, significantly improving the ability to learn graph distributions. Notable approaches include autoregressive or simultaneous models based on techniques such as variational autoencoders (VAE) (Jin et al., 2018; Simonovsky & Komodakis, 2018), recurrent neural networks (RNN) (You et al., 2018), normalizing flows (Zang & Wang, 2020; Shi et al., 2020; Luo et al., 2021), and generative adversarial networks (GAN) (Martinkus et al., 2022).

Table 4: Sampling results of various topological guides.

| Dataset | Guide | Val.↑ | FCD↓ | NSPDK↓ |
|---|---|---|---|---|
| QM9 | Noise | 91.52 | 0.829 | 0.0015 |
| | Peripheral | 97.58 | 0.305 | 0.0009 |
| | Cell | 98.74 | 0.172 | 0.0003 |
| ZINC 250k | Noise | 96.78 | 1.665 | 0.002 |
| | Peripheral | 97.93 | 1.541 | 0.002 |
| | Cell | 98.56 | 1.533 | 0.001 |

**Diffusion-based Graph Generation**. A breakthrough in graph generative models has been marked by the recent progress in diffusion-based generative models (Niu et al., 2020; Ho et al., 2020; Song et al., 2021). Recent models employ various strategies to enhance the generation of complex graphs, including capturing node-edge dependency (Jo et al., 2022), addressing discretization challenges (Vignac et al., 2023; Huang et al., 2023; Siraudin et al., 2025), exploiting low-to-high frequency generation curriculum (Mo et al., 2024; Huang & Birdal, 2025), and improving computational efficiency through low-rank diffusion processes (Luo et al., 2023a). CatFlow (Eijkelboom et al., 2024) and DeFoG (Qin et al., 2025) adopt flow matching as an alternative to diffusion, achieving more efficient generation. Recent studies have also enhanced diffusion-based generative models by incorporating diffusion bridge processes, *i.e.*, processes conditioned on the endpoints (Wu et al., 2022; Jo et al., 2024; Boget et al., 2025). Despite these advances, existing methods either overlook or inadvertently disrupt higher-order structures during graph generation, or struggle to model topological properties, as denoising noisy samples does not explicitly preserve the intricate structural dependencies required to generate realistic graphs. This highlights the need for a graph-friendly diffusion framework that explicitly learns higher-order topology, preserves meaningful intermediate states and trajectories, and avoids inappropriate noise injection.

**Higher-order Generative Models**. Generative modeling uses higher-order information mostly in the form of motifs and hypergraphs. MiCaM (Geng et al., 2023) synthesizes molecules by iteratively merging motifs. HypeBoy (Kim et al., 2024) learns hypergraph representation through hyperedge filling, while Hygene (Gailhard et al., 2025) reduces hypergraph generation to bipartite graphs. To the best of our knowledge, we are the first to consider higher-order guides for graph generation.

## 6 CONCLUSION

We introduce HOG-Diff, a coarse-to-fine generation framework that exploits higher-order topology as explicit guidance. By employing cell complex filtering and diffusion bridges, HOG-Diff decomposes graph generation into structured, easier-to-learn, topology-aware substeps and preserves meaningful intermediate states. Our theoretical analysis supports its advantages over classical diffusion approaches, including improved training dynamics and a tighter reconstruction error bound. Extensive experiments on eight molecular and generic graph benchmarks further validate these gains. Overall, this work represents an important step toward topological diffusion models, highlighting the importance of higher-order features that prior methods overlook and leaving ample room for future work.

**Limitations and future work**. The performance of HOG-Diff depends on the presence of explicit higher-order structures, which certain graph types might lack. Nevertheless, higher-order motifs are often deducible in realistic graphs. Our future work will explore various filtering mechanisms beyond CCF. We will adaptively determine the order (dimension of the higher-order skeleton).

ACKNOWLEDGMENTS

T. Birdal acknowledges support from the Engineering and Physical Sciences Research Council [grant EP/X011364/1]. T. Birdal was supported by a UKRI Future Leaders Fellowship [grant number MR/Y018818/1] as well as a Royal Society Research Grant RG/R1/241402.

ETHICS AND REPRODUCIBILITY STATEMENT

This paper presents work whose goal is to advance the field of deep generative models. Positive applications include generating graph-structured data for scientific discovery and accelerating drug discovery by generating novel molecular structures. However, like other generative technologies, our work could potentially be misused to synthesize harmful molecules, counterfeit social interactions, or deceptive network structures.

Detailed theoretical derivations are provided in App. A. The complete architecture of HOG-Diff, along with the training and sampling procedures, is described in App. C. Details of the datasets, preprocessing steps, and experimental settings are provided in App. E. Additional experimental results, ablation studies, and visualizations can be found in App. F.

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

# Appendix

**Organization**. The appendix is structured as follows: We first present the derivations excluded from the main paper due to space limitations in Section A. Additional explanations of related work are provided in Section B. Section C details the generation process, including the spectral diffusion framework, the architecture of the proposed score network, and the training and sampling procedures. Computational efficiency is discussed in Section D. Section E outlines the experimental setup, and Section F reports additional experimental results, covering the impact of diffusion domain choice, scalability to large graphs, standard deviation analysis, and visualizations of the generated samples. Section G concludes with limitations.

## A  FORMAL STATEMENTS AND PROOFS

This section presents the formal statements of key theoretical results along with their detailed derivations. We will recall and more precisely state the theoretical claims before presenting the proof.

### A.1  DIFFUSION BRIDGE PROCESS

Recall that the generalized Ornstein-Uhlenbeck (GOU) process, also known as the time-varying OU process, is a stationary Gaussian-Markov process whose marginal distribution gradually tends towards a stable mean and variance over time. The GOU process is generally defined as follows (Ahmad, 1988; Luo et al., 2023b):

$$\mathrm{d}\boldsymbol{G}_t = \theta_t \left(\boldsymbol{\mu} - \boldsymbol{G}_t\right) \mathrm{d}t + g_t \mathrm{d}\boldsymbol{W}_t, \tag{11}$$

where $\boldsymbol{\mu}$ is a given state vector, $\theta_t$ denotes a scalar drift coefficient, and $g_t$ represents the diffusion coefficient. Additionally, we assume the relation $g_t^2/\theta_t = 2\sigma^2$, where $\sigma^2$ is a given constant scalar. As a result, its transition probability possesses a closed-form analytical solution:

$$\begin{aligned} p\left(\boldsymbol{G}_t \mid \boldsymbol{G}_s\right) &= \mathcal{N}(\mathbf{m}_{s:t}, v_{s:t}^2 \boldsymbol{I}), \\ \mathbf{m}_{s:t} &= \boldsymbol{\mu} + \left(\boldsymbol{G}_s - \boldsymbol{\mu}\right) e^{-\bar{\theta}_{s:t}}, \\ v_{s:t}^2 &= \sigma^2 \left(1 - e^{-2\bar{\theta}_{s:t}}\right). \end{aligned} \tag{12}$$

Here, $\bar{\theta}_{s:t} = \int_s^t \theta_z dz$. When $s = 0$, we write $\bar{\theta}_t := \bar{\theta}_{0:t}$ for notation simplicity.

Doob's $h$-transform can modify an SDE to pass through a specified endpoint (Doob & Doob, 1984). When applied to the GOU process, it eliminates variance in the terminal state by driving the diffusion process toward a Dirac distribution centered at $\boldsymbol{G}_{\tau_k}$ (Heng et al., 2021; Yue et al., 2024), making it well-suited for stochastic modelling with terminal constraints.

In the following, we derive the generalized Ornstein–Uhlenbeck (GOU) bridge process using Doob's $h$-transform (Doob & Doob, 1984), and subsequently examine its relationship with the Brownian bridge process.

**Generalized Ornstein–Uhlenbeck (GOU) bridge**. Let $\boldsymbol{G}_t$ evolve according to the generalized OU process in Eq. (5), subject to the terminal conditional $\boldsymbol{\mu} = \boldsymbol{G}_{\tau_k}$. Applying Doob's $h$-transform (Doob & Doob, 1984), we can derive the GOU bridge process as follows:

$$\mathrm{d}\boldsymbol{G}_t = \theta_t \left(1 + \frac{2}{e^{2\bar{\theta}_{t:\tau_k}} - 1}\right) (\boldsymbol{G}_{\tau_k} - \boldsymbol{G}_t)\mathrm{d}t + g_{k,t}\mathrm{d}\boldsymbol{W}_t. \tag{13}$$

The conditional transition probability $p(\boldsymbol{G}_t \mid \boldsymbol{G}_{\tau_{k-1}}, \boldsymbol{G}_{\tau_k})$ admits an analytical expression:

$$\begin{aligned} p(\boldsymbol{G}_t \mid \boldsymbol{G}_{\tau_{k-1}}, \boldsymbol{G}_{\tau_k}) &= \mathcal{N}(\bar{\mathbf{m}}_t, \bar{v}_t^2 \boldsymbol{I}), \\ \bar{\mathbf{m}}_t &= \boldsymbol{G}_{\tau_k} + (\boldsymbol{G}_{\tau_{k-1}} - \boldsymbol{G}_{\tau_k})e^{-\bar{\theta}_{\tau_{k-1}:t}} \frac{v_{t:\tau_k}^2}{v_{\tau_{k-1}:\tau_k}^2}, \\ \bar{v}_t^2 &= v_{\tau_{k-1}:t}^2 v_{t:\tau_k}^2 / v_{\tau_{k-1}:\tau_k}^2. \end{aligned} \tag{14}$$

Here, $\bar{\theta}_{a:b} = \int_a^b \theta_s \mathrm{d}s$, and $v_{a:b} = \sigma^2(1 - e^{-2\bar{\theta}_{a:b}})$.

*Proof.* Without loss of generality, consider one generation interval $[\tau_{k-1}, \tau_k]$ and denote $T = \tau_k$, $\mathbf{x}_t = \boldsymbol{G}_t^{(k)}$, $0 = \tau_{k-1}$, and endpoints $\mathbf{x}_0 = \boldsymbol{G}_{\tau_{k-1}}$, $\mathbf{x}_T = \boldsymbol{G}_{\tau_k}$.

From Eq. (6), we can derive the following conditional distribution

$$p(\mathbf{x}_T \mid \mathbf{x}_t) = \mathcal{N}(\mathbf{x}_T + (\mathbf{x}_t - \mathbf{x}_T)e^{\bar{\theta}_{t:T}}, v_{t:T}^2 \boldsymbol{I}). \tag{15}$$

Hence, the $h$-function can be directly computed as:

$$\begin{aligned}
\boldsymbol{h}(\mathbf{x}_t, t, \mathbf{x}_T, T) &= \nabla_{\mathbf{x}_t} \log p(\mathbf{x}_T \mid \mathbf{x}_t) \\
&= -\nabla_{\mathbf{x}_t} \left[ \frac{(\mathbf{x}_t - \mathbf{x}_T)^2 e^{-2\bar{\theta}_{t:T}}}{2v_{t:T}^2} + const \right] \\
&= (\mathbf{x}_T - \mathbf{x}_t) \frac{e^{-2\bar{\theta}_{t:T}}}{v_{t:T}^2} \\
&= (\mathbf{x}_T - \mathbf{x}_t)\sigma^{-2}/(e^{2\bar{\theta}_{t:T}} - 1).
\end{aligned} \tag{16}$$

Following the approach in Yue et al. (2024), applying the Doob's $h$-transform yields the representation of an endpoint $\mathbf{x}_T$ conditioned process defined by the following SDE:

$$\begin{aligned}
\mathrm{d}\mathbf{x}_t &= \left[ f(\mathbf{x}_t, t) + g_t^2 \boldsymbol{h}(\mathbf{x}_t, t, \mathbf{x}_T, T) \right] \mathrm{d}t + g_t \mathrm{d}\mathbf{w}_t \\
&= \left( \theta_t + \frac{g_t^2}{\sigma^2(e^{2\bar{\theta}_{t:T}} - 1)} \right) (\mathbf{x}_T - \mathbf{x}_t)\mathrm{d}t + g_t \mathrm{d}\mathbf{w}_t \\
&= \theta_t \left( 1 + \frac{2}{e^{2\bar{\theta}_{t:T}} - 1} \right) (\mathbf{x}_T - \mathbf{x}_t)\mathrm{d}t + g_t \mathrm{d}\mathbf{w}_t.
\end{aligned} \tag{17}$$

Given that the joint distribution of $[\mathbf{x}_0, \mathbf{x}_t, \mathbf{x}_T]$ is multivariate normal, the conditional distribution $p(\mathbf{x}_t \mid \mathbf{x}_0, \mathbf{x}_T)$ is also Gaussian:

$$p(\mathbf{x}_t \mid \mathbf{x}_0, \mathbf{x}_T) = \mathcal{N}(\bar{\mathbf{m}}_t, \bar{v}_t^2 \boldsymbol{I}), \tag{18}$$

where the mean $\bar{\mathbf{m}}_t$ and variance $\bar{v}_t^2$ are determined using the conditional formulas for multivariate normal variables:

$$\begin{aligned}
\bar{\mathbf{m}}_t &= \mathbb{E}[\mathbf{x}_t \mid \mathbf{x}_0, \mathbf{x}_T] = \mathbb{E}[\mathbf{x}_t \mid \mathbf{x}_0] + \mathrm{Cov}(\mathbf{x}_t, \mathbf{x}_T \mid \mathbf{x}_0)\mathrm{Var}(\mathbf{x}_T \mid \mathbf{x}_0)^{-1}(\mathbf{x}_T - \mathbb{E}[\mathbf{x}_T \mid \mathbf{x}_0]), \\
\bar{v}_t^2 &= \mathrm{Var}(\mathbf{x}_t \mid \mathbf{x}_0, \mathbf{x}_T) = \mathrm{Var}(\mathbf{x}_t \mid \mathbf{x}_0) - \mathrm{Cov}(\mathbf{x}_t, \mathbf{x}_T \mid \mathbf{x}_0)\mathrm{Var}(\mathbf{x}_T \mid \mathbf{x}_0)^{-1}\mathrm{Cov}(\mathbf{x}_T, \mathbf{x}_t \mid \mathbf{x}_0).
\end{aligned} \tag{19}$$

Notice that

$$\mathrm{Cov}(\mathbf{x}_t, \mathbf{x}_T \mid \mathbf{x}_0) = \mathrm{Cov}\left( \mathbf{x}_t, (\mathbf{x}_t - \mathbf{x}_T)e^{-\bar{\theta}_{t:T}} \mid \mathbf{x}_0 \right) = e^{-\bar{\theta}_{t:T}}\mathrm{Var}(\mathbf{x}_t \mid \mathbf{x}_0). \tag{20}$$

By substituting this and the results in Eq. (6) into Eq. (19), we can obtain

$$\begin{aligned}
\bar{\mathbf{m}}_t &= \left( \mathbf{x}_T + (\mathbf{x}_0 - \mathbf{x}_T)e^{-\bar{\theta}_t} \right) + \left( e^{-\bar{\theta}_{t:T}} v_t^2 \right)/v_T^2 \cdot \left( \mathbf{x}_T - \mathbf{x}_T - (\mathbf{x}_0 - \mathbf{x}_T)e^{-\bar{\theta}_T} \right) \\
&= \mathbf{x}_T + (\mathbf{x}_0 - \mathbf{x}_T) \left( e^{-\bar{\theta}_t} - e^{-\bar{\theta}_{t:T}} e^{-\bar{\theta}_T} v_t^2/v_T^2 \right) \\
&= \mathbf{x}_T + (\mathbf{x}_0 - \mathbf{x}_T)e^{-\bar{\theta}_t} \left( \frac{1 - e^{-2\bar{\theta}_T} - e^{-2\bar{\theta}_{t:T}}(1 - e^{-2\bar{\theta}_t})}{1 - e^{-2\bar{\theta}_T}} \right) \\
&= \mathbf{x}_T + (\mathbf{x}_0 - \mathbf{x}_T)e^{-\bar{\theta}_t} v_{t:T}^2/v_T^2,
\end{aligned} \tag{21}$$

and

$$\begin{aligned}
\bar{v}_t^2 &= v_t^2 - \left( e^{-\bar{\theta}_{t:T}} v_t^2 \right)^2 /v_T^2 \\
&= \frac{v_t^2}{v_T^2}(v_T^2 - e^{-2\bar{\theta}_{t:T}} v_t^2) \\
&= \frac{v_t^2}{v_T^2}\sigma^2 \left( 1 - e^{-2\bar{\theta}_T} - e^{-2\bar{\theta}_{t:T}}(1 - e^{-2\bar{\theta}_t}) \right) \\
&= v_t^2 v_{t:T}^2/v_T^2.
\end{aligned} \tag{22}$$

Finally, we conclude the proof by reverting to the original notations. $\qquad\square$

Note that the GOU bridge process, also referred to as the conditional GOU process, has been studied theoretically in previous works (Salminen, 1984; Heng et al., 2021; Yue et al., 2024). However, we are the first to demonstrate its effectiveness in explicitly learning higher-order structures within the graph generation process.

**Brownian Bridge Process**. In the following, we demonstrate that the Brownian bridge process is a particular case of the generalized OU bridge process when $\theta_t$ approaches zero.

Assuming $\theta_t = \theta$ is a constant that tends to zero, we obtain

$$\bar{\theta}_{a:b} = \int_a^b \theta_s \, \mathrm{d}s = \theta(b-a) \to 0. \tag{23}$$

Consider the term $e^{2\bar{\theta}_{t:\tau_k}} - 1$, we approximate the exponential function using a first-order Taylor expansion for small $\bar{\theta}_{t:\tau_k}$:

$$e^{2\bar{\theta}_{t:\tau_k}} - 1 \approx 2\bar{\theta}_{t:\tau_k} \to 2\theta(\tau_k - t). \tag{24}$$

Hence, the drift term in the generalized OU bridge simplifies to

$$\theta_t \left( 1 + \frac{2}{e^{2\bar{\theta}_{t:\tau_k}} - 1} \right) \approx \theta \left( 1 + \frac{2}{2\theta(\tau_k - t)} \right) \to \frac{1}{\tau_k - t}. \tag{25}$$

Consequently, in the limit $\theta_t \to 0$, the GOU bridge process described in Eq. (13) can be modelled by the following SDE:

$$\mathrm{d}\boldsymbol{G}_t = \frac{\boldsymbol{G}_{\tau_k} - \boldsymbol{G}_t}{\tau_k - t} \mathrm{d}t + g_{k,t} \mathrm{d}\boldsymbol{W}_t. \tag{26}$$

This equation precisely corresponds to the SDE representation of the classical Brownian bridge process.

In contrast to the GOU bridge process in Eq. (13), the evolution of the Brownian bridge is fully determined by the noise schedule $g_{k,t}$, resulting in a simpler SDE representation. However, this constraint in the Brownian bridge reduces the flexibility in designing the generative process.

Note that the Brownian bridge is an endpoint-conditioned process relative to a reference Brownian motion, which the SDE governs:

$$\mathrm{d}\boldsymbol{G}_t = g_t \mathrm{d}\boldsymbol{W}_t. \tag{27}$$

This equation describes a pure diffusion process without drift, making it a specific instance of the GOU process.

## A.2 PROOF OF THEOREM 3

To establish the proof, we begin by introducing essential definitions and assumptions.

**Definition 5** ($\beta$-smooth)**.** *A function $f : \mathbb{R}^m \to \mathbb{R}^n$ is said to be $\beta$-smooth if and only if*

$$\|f(\mathbf{w}) - f(\mathbf{v}) - \nabla f(\mathbf{v})(\mathbf{w} - \mathbf{v})\| \leq \frac{\beta}{2} \|\mathbf{w} - \mathbf{v}\|^2, \forall \mathbf{w}, \mathbf{v} \in \mathbb{R}^m. \tag{28}$$

**Theorem 3** (Formal)**.** *Let $\ell^{(k)}(\boldsymbol{\theta})$ be a loss function that is $\beta$-smooth and satisfies the $\mu$-PL (Polyak-Łojasiewicz) condition in the ball $B(\boldsymbol{\theta}_0, R)$ of radius $R = 2N\sqrt{2\beta\ell^{(k)}(\boldsymbol{\theta}_0)}/(\mu\delta)$, where $\delta > 0$. Then, with probability $1 - \delta$ over the choice of mini-batch of size $b$, stochastic gradient descent (SGD) with a learning rate $\eta^* = \frac{\mu N}{N\beta(N^2\beta + \mu(b-1))}$ converges to a global solution in the ball $B$ with exponential convergence rate:*

$$\mathbb{E}\left[\ell^{(k)}(\boldsymbol{\theta}_i)\right] \leq \left( 1 - \frac{b\mu^2}{\beta N (\beta N^2 + \mu(b-1))} \right)^i \ell^{(k)}(\boldsymbol{\theta}_0). \tag{29}$$

*Here, $N$ denotes the size of the training dataset. Furthermore, the proposed generative model yields a smaller smoothness constant $\beta_{HOG\text{-}Diff}$ compared to that of the classical model $\beta_{classical}$, i.e., $\beta_{HOG\text{-}Diff} \leq \beta_{classical}$, implying that the learned distribution in HOG-Diff converges to the target distribution faster than classical generative models.*

*Proof.* Assume that the loss function $\ell^{(k)}(\boldsymbol{\theta})$ in Eq. (9) is minimized using standard Stochastic Gradient Descent (SGD) on a training dataset $\mathcal{S} = \{\mathbf{x}^i\}_{i=1}^N$. At the $i$-th iteration, parameter $\boldsymbol{\theta}_i$ is updated using a mini-batch of size $b$ as follows:

$$\boldsymbol{\theta}_{i+1} \triangleq \boldsymbol{\theta}_i - \eta \nabla \ell^{(k)}(\boldsymbol{\theta}_i), \tag{30}$$

where $\eta$ is the learning rate.

Following Liu et al. (2020) and Luo et al. (2023a), we assume that $\ell^{(k)}(\boldsymbol{\theta})$ is $\beta$-smooth and satisfies the $\mu$-PL condition in the ball $B(\boldsymbol{\theta}_0, R)$ with $R = 2N\sqrt{2\beta\ell^{(k)}(\boldsymbol{\theta}_0)}/(\mu\delta)$ where $\delta > 0$. Then, with probability $1 - \delta$ over the choice of mini-batch of size $b$, SGD with a learning rate $\eta^* = \frac{\mu N}{N\beta(N^2\beta+\mu(b-1))}$ converges to a global solution in the ball $B(\boldsymbol{\theta}_0, R)$ with exponential convergence rate (Liu et al., 2020):

$$\mathbb{E}[\ell^{(k)}(\boldsymbol{\theta}_i)] \le \left(1 - \frac{b\mu\eta^*}{N}\right)^i \ell^{(k)}(\boldsymbol{\theta}_0) = \left(1 - \frac{b\mu^2}{\beta N(\beta N^2 + \mu(b-1))}\right)^i \ell^{(k)}(\boldsymbol{\theta}_0). \tag{31}$$

Next, we show that the proposed framework has a smaller smoothness constant than the classical one-step model. Therefore, we focus exclusively on the spectral component $||s_{\boldsymbol{\theta},\boldsymbol{\Lambda}}^{(k)} - \nabla_{\boldsymbol{\Lambda}} \log p_t(\boldsymbol{G}_t | \boldsymbol{G}_{\tau_k})||_2^2$ from the full loss function in Eq. (9), as the feature-related part of the loss function in HOG-Diff aligns with that of the classical framework. For simplicity, we use the notation $\bar{\ell}(\boldsymbol{\theta}) = ||s_{\boldsymbol{\theta},\boldsymbol{\Lambda}}^{(k)} - \nabla_{\boldsymbol{\Lambda}} \log p_t(\boldsymbol{G}_t | \boldsymbol{G}_{\tau_k})||^2 = ||s_{\boldsymbol{\theta}}(\mathbf{x}_t) - \nabla_{\mathbf{x}} \log p_t(\mathbf{x}_t)||^2$ as the feature-related part of the loss.

Next, we verify that $\bar{\ell}(\boldsymbol{\theta})$ is $\beta$-smooth under the assumptions given. Notice that the gradient of the loss function is given by:

$$\nabla\bar{\ell}(\boldsymbol{\theta}) = 2\mathbb{E}\left[(s_{\boldsymbol{\theta}}(\mathbf{x}) - \nabla \log p(\mathbf{x}))^\top \nabla_{\boldsymbol{\theta}} s_{\boldsymbol{\theta}}(\mathbf{x})\right] \tag{32}$$

Hence,

$$\|\nabla\bar{\ell}(\boldsymbol{\theta}_1) - \nabla\bar{\ell}(\boldsymbol{\theta}_2)\| = 2 \left\|\mathbb{E}\left[(s_{\boldsymbol{\theta}_1}(\mathbf{x}) - \nabla \log p(\mathbf{x}))^\top \nabla s_{\boldsymbol{\theta}_1}(\mathbf{x}) - (s_{\boldsymbol{\theta}_2}(\mathbf{x}) - \nabla \log p(\mathbf{x}))^\top \nabla s_{\boldsymbol{\theta}_2}(\mathbf{x})\right]\right\|$$
$$\le 2\mathbb{E}[\|s_{\boldsymbol{\theta}_1}(\mathbf{x}) - s_{\boldsymbol{\theta}_2}(\mathbf{x})\| \cdot \|\nabla s_{\boldsymbol{\theta}_1}(\mathbf{x})\| + \|s_{\boldsymbol{\theta}_2}(\mathbf{x}) - \nabla \log p(\mathbf{x})\| \cdot \|\nabla s_{\boldsymbol{\theta}_1}(\mathbf{x}) - \nabla s_{\boldsymbol{\theta}_2}(\mathbf{x})\|]. \tag{33}$$

Suppose $\|\nabla_{\boldsymbol{\theta}} s_{\boldsymbol{\theta}}(\mathbf{x})\| \le C_1$ and $\|s_{\boldsymbol{\theta}}(\mathbf{x}) - \nabla \log p(\mathbf{x})\| \le C_2$, then we can obtain

$$\|\nabla\bar{\ell}(\boldsymbol{\theta}_1) - \nabla\bar{\ell}(\boldsymbol{\theta}_2)\| \le 2\mathbb{E}\left[C_1\beta_{s_{\boldsymbol{\theta}}}\|\boldsymbol{\theta}_1 - \boldsymbol{\theta}_2\| + C_2\beta_{\nabla s_{\boldsymbol{\theta}}}\|\boldsymbol{\theta}_1 - \boldsymbol{\theta}_2\|\right]$$
$$= 2(\beta_{s_{\boldsymbol{\theta}}}C_1 + C_2\beta_{\nabla s_{\boldsymbol{\theta}}})\|\boldsymbol{\theta}_1 - \boldsymbol{\theta}_2\|. \tag{34}$$

To satisfy the $\beta$-smooth of $\bar{\ell}(\boldsymbol{\theta})$, we require that

$$2(C_1\beta_{s_{\boldsymbol{\theta}}} + C_2\beta_{\nabla s_{\boldsymbol{\theta}}}) \le \beta. \tag{35}$$

This implies that the distribution learned by the proposed framework can converge to the target distribution. Therefore, following Chung et al. (2022), we further assume that $s_{\boldsymbol{\theta}}$ is a sufficiently expressive parameterized score function so that $\beta_{s_{\boldsymbol{\theta}}} = \beta_{\nabla \log p_{t|\tau_{k-1}}}$ and $\beta_{\nabla^2 s_{\boldsymbol{\theta}}} = \beta_{\nabla^2 \log p_{t|\tau_{k-1}}}$.

The loss function used in classical diffusion generative models is given by $\bar{\ell}(\boldsymbol{\varphi}) = \mathbb{E}||s_{\boldsymbol{\varphi}}(\mathbf{x}_t) - \nabla_{\mathbf{x}_t} q_t(\mathbf{x}_t | \mathbf{x}_0)||^2$. To demonstrate that the proposed framework converges faster to the target distribution compared to the classical one-step generation framework, it suffices to show that: $\beta_{\nabla p_{t|\tau_{k-1}}} \le \beta_{\nabla q_{t|0}}$ and $\beta_{\nabla^2 p_{t|\tau_{k-1}}} \le \beta_{\nabla^2 q_{t|0}}$.

Let $\mathbf{x} \sim q_{t|0}$ and $\mathbf{x}' \sim p_{t|\tau_{k-1}}$. Since we inject topological information from $\mathbf{x}$ into $\mathbf{x}'$, $\mathbf{x}'$ can be viewed as being obtained by adding noise to $\mathbf{x}$. Hence, we can model $\mathbf{x}'$ as $\mathbf{x}' = \mathbf{x} + \epsilon$ where $\epsilon \sim \mathcal{N}(\mathbf{0}, \sigma^2 \boldsymbol{I})$. The variance of Gaussian noise $\sigma^2$ controls the information retained in $\mathbf{x}'$. Hence, its distribution can be expressed as $p(\mathbf{x}') = \int q(\mathbf{x}' - \epsilon)\pi(\epsilon) \, d\epsilon$.

Therefore, we can obtain

$$
\begin{aligned}
||\nabla_{\mathbf{x}'}^k p(\mathbf{x}_1') - \nabla_{\mathbf{x}'}^k p(\mathbf{x}_2')|| = ||\nabla_{\mathbf{x}'}^k \int \left( q(\mathbf{x}_1' - \epsilon) - q(\mathbf{x}_2' - \epsilon) \right) \pi(\epsilon) \, \mathrm{d}\epsilon|| \\
\le \int ||\nabla_{\mathbf{x}'}^k q(\mathbf{x}_1' - \epsilon) - \nabla_{\mathbf{x}'}^k q(\mathbf{x}_2' - \epsilon)|| \pi(\epsilon) \, \mathrm{d}\epsilon \\
\le ||\nabla_{\mathbf{x}'}^k q(\mathbf{x}')||_{\mathrm{lip}} (\mathbf{x}_1' - \mathbf{x}_2') \int \pi(\epsilon) \, \mathrm{d}\epsilon \\
\le ||\nabla_{\mathbf{x}'}^k q(\mathbf{x}')||_{\mathrm{lip}} (\mathbf{x}_1' - \mathbf{x}_2').
\end{aligned}
\tag{36}
$$

Hence, $||\nabla_{\mathbf{x}'}^k \log p(\mathbf{x}')||_{\mathrm{lip}} \le ||\nabla_{\mathbf{x}'}^k \log q(\mathbf{x}')||_{\mathrm{lip}}$.

By setting $k = 3$ and $k = 4$, we can obtain $\beta_{\nabla \log p_{t|\tau_{k-1}}} \le \beta_{\nabla \log q_{t|0}}$ and $\beta_{\nabla^2 \log p_{t|\tau_{k-1}}} \le \beta_{\nabla^2 \log q_{t|0}}$. Therefore $\beta_{\text{HOG-Diff}} \le \beta_{\text{classical}}$, implying that the training process of HOG-Diff ($s_{\boldsymbol{\theta}}$) will converge faster than the classical generative framework ($s_{\boldsymbol{\varphi}}$).

$\square$

## A.3 PROOF OF THEOREM 4

Here, we denote the expected reconstruction error at each generation process as $\mathcal{E}(t) = \mathbb{E} \left\| \bar{\boldsymbol{G}}_t - \widehat{\boldsymbol{G}}_t \right\|^2$.

Before comparing the reconstruction error bounds of HOG-Diff and the classical diffusion model, we first relate their optimal score-matching losses at the function-class level.

**Lemma 6.** *Let $\ell^{(k)}(\boldsymbol{\theta}^{(k)})$ denote the score-matching loss of the $k$-th HOG-Diff stage on $[\tau_{k-1}, \tau_k]$, and let $\ell_{\mathrm{cls}}(\boldsymbol{\varphi})$ be the classical single-stage loss on $[0, T]$. Define the corresponding optimal values*

$$
\ell_{\star}^{(k)} := \inf_{\boldsymbol{\theta}^{(k)}} \ell^{(k)}(\boldsymbol{\theta}^{(k)}), \qquad \ell_{\mathrm{cls},\star} := \inf_{\boldsymbol{\varphi}} \ell_{\mathrm{cls}}(\boldsymbol{\varphi}).
\tag{37}
$$

*Assume that, for each $k$, the score network used in HOG-Diff on $[\tau_{k-1}, \tau_k]$ is instantiated from the same backbone architecture as the classical score network on $[0, T]$ (with the coarse graph $\boldsymbol{G}_{\tau_k}$ provided as an additional input that can be ignored by a suitable parameter choice). Under this construction, the classical score class is contained in each stage-wise score class, and we have*

$$
\sum_{k=1}^K \ell_{\star}^{(k)} \le \ell_{\mathrm{cls},\star}.
\tag{38}
$$

*Proof.* Fix any parameter $\boldsymbol{\varphi}$ of the classical model and consider the associated score network $s_{\boldsymbol{\varphi}}(\boldsymbol{G}_t, t)$ on $[0, T]$. By the expressivity assumption, for each $k$ there exists a parameter vector $\boldsymbol{\theta}^{(k)}$ such that the HOG-Diff score network on $[\tau_{k-1}, \tau_k]$ can represent exactly the same score function while ignoring the coarse guide:

$$
s_{\boldsymbol{\theta}^{(k)}}^{(k)}(\boldsymbol{G}_t, \boldsymbol{G}_{\tau_k}, t) = s_{\boldsymbol{\varphi}}(\boldsymbol{G}_t, t), \qquad t \in [\tau_{k-1}, \tau_k].
\tag{39}
$$

By definition, the stage-wise loss $\ell^{(k)}(\boldsymbol{\theta}^{(k)})$ is obtained by restricting the classical objective $\ell_{\mathrm{cls}}(\boldsymbol{\varphi})$ to the time window $[\tau_{k-1}, \tau_k]$ and using the same score function on that interval. Therefore, under the above parameter choice we have

$$
\ell^{(k)}(\boldsymbol{\theta}^{(k)}) = \ell_{\mathrm{cls}}^{(k)}(\boldsymbol{\varphi}),
\tag{40}
$$

where $\ell_{\mathrm{cls}}^{(k)}(\boldsymbol{\varphi})$ denotes the contribution of the interval $[\tau_{k-1}, \tau_k]$ to the classical loss. Summing over all disjoint windows yields the exact decomposition

$$
\sum_{k=1}^K \ell^{(k)}(\boldsymbol{\theta}^{(k)}) = \sum_{k=1}^K \ell_{\mathrm{cls}}^{(k)}(\boldsymbol{\varphi}) = \ell_{\mathrm{cls}}(\boldsymbol{\varphi}).
\tag{41}
$$

Taking the infimum over $(\boldsymbol{\theta}^{(1)}, \ldots, \boldsymbol{\theta}^{(K)})$ on the left-hand side and over $\boldsymbol{\varphi}$ on the right-hand side gives

$$\sum_{k=1}^{K} \ell_{\star}^{(k)} = \sum_{k=1}^{K} \inf_{\boldsymbol{\theta}^{(k)}} \ell^{(k)}(\boldsymbol{\theta}^{(k)}) \leq \inf_{\boldsymbol{\varphi}} \ell_{\mathrm{cls}}(\boldsymbol{\varphi}) = \ell_{\mathrm{cls}, \star}. \tag{42}$$

$\square$

Lemma 6 reflects the structural fact that the classical model is a special case of the multi-stage architecture when all stages share the same backbone and the coarse graph input is ignored. Intuitively, each HOG-Diff stage only needs to approximate a smoother conditional score (given the coarse graph), whereas the classical model has to fit the full marginal score in a single stage. For a fixed network capacity, decomposing the problem into these easier conditional subproblems is therefore not expected to deteriorate the overall score-matching accuracy.

**Theorem 4.** *Under standard Lipschitz and boundedness assumptions, the reconstruction error of HOG-Diff satisfies the following bound:*

$$\mathcal{E}_{\mathrm{hog}}(0) \leq A_{\mathrm{hog}} \, e^{T_{\mathrm{hog}}}, \tag{43}$$

*where* $A_{\mathrm{hog}} = C^2 \sum_{k=1}^{K} \ell^{(k)}(\boldsymbol{\theta}) \int_{\tau_{k-1}}^{\tau_k} g_{k,s}^4 \mathrm{d}s$ *and* $T_{\mathrm{hog}} = \sum_{k=1}^{K} \int_{\tau_{k-1}}^{\tau_k} \gamma_k(s) \mathrm{d}s + C \sum_{k=1}^{K-1} \int_{\tau_{k-1}}^{\tau_k} h_{k,s}^2 \mathrm{d}s.$ *Moreover, under the same assumptions, HOG-Diff admits a tighter reconstruction error bound than classical diffusion models.*

*Proof.* Recall that $\mathcal{E}(t) = \mathbb{E} \left\| \bar{\boldsymbol{G}}_t - \widehat{\boldsymbol{G}}_t \right\|^2$ reflects the expected error between the data reconstructed with the ground truth score $\nabla \log p_t(\cdot)$ and the learned scores $\boldsymbol{s_\theta}(\cdot)$. In particular, $\bar{\boldsymbol{G}}$ is obtained by solving the following oracle reversed time SDE:

$$\mathrm{d}\bar{\boldsymbol{G}}_t = \left( \mathbf{f}_{k,t}(\bar{\boldsymbol{G}}_t) - g_{k,t}^2 \nabla_{\boldsymbol{G}} \log p_t(\bar{\boldsymbol{G}}_t) \right) \mathrm{d}\bar{t} + g_{k,t} \, \mathrm{d}\bar{\boldsymbol{W}}_t, t \in [\tau_{k-1}, \tau_k], \tag{44}$$

whereas $\widehat{\boldsymbol{G}}_t$ is governed based on the corresponding estimated reverse time SDE:

$$\mathrm{d}\widehat{\boldsymbol{G}}_t = \left( \mathbf{f}_{k,t}(\widehat{\boldsymbol{G}}_t) - g_{k,t}^2 \boldsymbol{s_\theta}(\widehat{\boldsymbol{G}}_t, t) \right) \mathrm{d}\bar{t} + g_{k,t} \, \mathrm{d}\bar{\boldsymbol{W}}_t, t \in [\tau_{k-1}, \tau_k]. \tag{45}$$

Here, $\mathbf{f}_{k,t}$ is the drift function of the Ornstein–Uhlenbeck bridge. For simplicity, we denote the Lipschitz norm by $|| \cdot ||_{\mathrm{lip}}$ and $\mathbf{f}_{k,s}(\boldsymbol{G}_s) = h_{k,s}(\boldsymbol{G}_{\tau_k} - \boldsymbol{G}_s)$, where $h_{k,s} = \theta_s \left( 1 + \frac{2}{e^{2\theta_s : \tau_k} - 1} \right).$

**Step 1: Single stage error bound.**

To bound the expected reconstruction error $\mathbb{E} \left\| \bar{\boldsymbol{G}}_{\tau_{k-1}} - \widehat{\boldsymbol{G}}_{\tau_{k-1}} \right\|^2$ at each generation process, we begin by analyzing how $\mathbb{E} \left\| \bar{\boldsymbol{G}}_t - \widehat{\boldsymbol{G}}_t \right\|^2$ evolves as time $t$ is reversed from $\tau_k$ to $\tau_{k-1}$. The reconstruction error goes as follows

$$\mathcal{E}(t) \leq \mathbb{E} \int_{\tau_k}^{t} \left\| \left( \mathbf{f}_{k,s}(\bar{\boldsymbol{G}}_s) - \mathbf{f}_{k,s}(\widehat{\boldsymbol{G}}_s) \right) + g_{k,s}^2 \left( \boldsymbol{s_\theta}(\widehat{\boldsymbol{G}}_s, s) - \nabla_{\boldsymbol{G}} \log p_s(\bar{\boldsymbol{G}}_s) \right) \right\|^2 \mathrm{d}\bar{s}$$

$$\leq C \mathbb{E} \int_{\tau_k}^{t} \left\| \mathbf{f}_{k,s}(\bar{\boldsymbol{G}}_s) - \mathbf{f}_{k,s}(\widehat{\boldsymbol{G}}_s) \right\|^2 \mathrm{d}\bar{s} + C \mathbb{E} \int_{\tau_k}^{t} g_{k,s}^4 \left\| \boldsymbol{s_\theta}(\widehat{\boldsymbol{G}}_s, s) - \nabla_{\boldsymbol{G}} \log p_s(\bar{\boldsymbol{G}}_s) \right\|^2 \mathrm{d}\bar{s}$$

$$\leq C \int_{\tau_k}^{t} \| h_{k,s} \|_{\mathrm{lip}}^2 \cdot \mathcal{E}(s) \mathrm{d}\bar{s} + C \mathcal{E}(\tau_k) \int_{\tau_k}^{t} h_{k,s}^2 \mathrm{d}\bar{s}$$

$$+ C^2 \int_{\tau_k}^{t} g_{k,s}^4 \cdot \mathbb{E} \left\| \boldsymbol{s_\theta}(\widehat{\boldsymbol{G}}_s, s) - \boldsymbol{s_\theta}(\bar{\boldsymbol{G}}_s, s) \right\|^2 + g_{k,s}^4 \cdot \mathbb{E} \left\| \boldsymbol{s_\theta}(\bar{\boldsymbol{G}}_s, s) - \nabla_{\boldsymbol{G}} \log p_s(\bar{\boldsymbol{G}}_s) \right\|^2 \mathrm{d}\bar{s}$$

$$\leq \underbrace{C^2 \ell^{(k)}(\boldsymbol{\theta}) \int_{\tau_k}^{t} g_{k,s}^4 \mathrm{d}\bar{s} + C \mathcal{E}(\tau_k) \int_{\tau_k}^{t} h_{k,s}^2 \mathrm{d}\bar{s}}_{\alpha_k(t)} + \int_{\tau_k}^{t} \underbrace{\left( C^2 g_{k,s}^4 \| \boldsymbol{s_\theta}(\cdot, s) \|_{\mathrm{lip}}^2 + C \| h_{k,s} \|_{\mathrm{lip}}^2 \right)}_{\gamma_k(s)} \mathcal{E}(s) \mathrm{d}\bar{s}$$

$$= \alpha_k(t) + \int_{\tau_k}^{t} \gamma_k(s) \mathcal{E}(s) \mathrm{d}\bar{s}. \tag{46}$$

Let $v(t) = \mathcal{E}(\tau_k - t)$ and $s' = \tau_k - s$, it can be derived that

$$v(t) = \mathcal{E}(\tau_k - t) \leq \alpha_k(\tau_k - t) + \int_0^t \gamma_k(\tau_k - s')v(s')\,\mathrm{d}s'. \tag{47}$$

Here, $\alpha_k(\tau_k - t)$ is a non-decreasing function. By applying Grönwall's inequality, we can derive that

$$v(t) \leq \alpha_k(\tau_k - t) \exp \int_0^t \gamma_k(\tau_k - s')\mathrm{d}s' \tag{48}$$

$$= \alpha_k(\tau_k - t) \exp \int_{\tau_k - t}^{\tau_k} \gamma_k(s)\mathrm{d}s. \tag{49}$$

Hence,

$$\mathcal{E}(t) \leq \alpha_k(t) \exp \int_t^{\tau_k} \gamma_k(s)\mathrm{d}s. \tag{50}$$

Evaluating the bound at the beginning of the $k$-th reverse stage, that is $t = \tau_{k-1}$, yields

$$\mathcal{E}(\tau_{k-1}) \leq \alpha_k(\tau_{k-1}) \exp \left( \int_{\tau_{k-1}}^{\tau_k} \gamma_k(s)\mathrm{d}s \right)$$

$$= \left[ C^2 \ell^{(k)}(\boldsymbol{\theta}) \int_{\tau_{k-1}}^{\tau_k} g_{k,s}^4 \mathrm{d}s + C\mathcal{E}(\tau_k) \int_{\tau_{k-1}}^{\tau_k} h_{k,s}^2 \mathrm{d}s \right] \exp \left( \int_{\tau_{k-1}}^{\tau_k} \gamma_k(s)\mathrm{d}s \right). \tag{51}$$

For later convenience, we introduce the nonnegative stage-wise coefficients

$$A_k = C^2 \ell^{(k)}(\boldsymbol{\theta}) \int_{\tau_{k-1}}^{\tau_k} g_{k,s}^4 \mathrm{d}s, \quad D_k = C \int_{\tau_{k-1}}^{\tau_k} h_{k,s}^2 \mathrm{d}s, \quad T_k = \int_{\tau_{k-1}}^{\tau_k} \gamma_k(s)\mathrm{d}s. \tag{52}$$

Then the error bound can be written compactly as

$$\mathcal{E}(\tau_{k-1}) \leq \left( A_k + D_k \mathcal{E}(\tau_k) \right) e^{T_k}, \qquad k = 1, \cdots, K. \tag{53}$$

The classical diffusion model corresponds to a single reverse stage on $[0, T]$, and its error bound can be formulated in the same functional form as

$$\mathcal{E}_{\mathrm{cls}}(0) \leq A_{\mathrm{cls}} e^{T_{\mathrm{cls}}}, \tag{54}$$

where $A_{\mathrm{cls}} := C^2 \ell_{\mathrm{cls}}(\boldsymbol{\varphi}) \int_0^T g_s^4 \, \mathrm{d}s$ and $T_{\mathrm{cls}} := \int_0^T \gamma_{\mathrm{cls}}(s)\mathrm{d}s + C \int_0^T h_s^2 \mathrm{d}s$.

**Step 2: Multi-stage accumulated error bound.**

The reverse process starts from the same prior for both oracle and learned SDEs, hence $\mathcal{E}(\tau_K) = 0$. Unrolling the recursion Eq. (53) backward from $k = K$ yields

$$\mathcal{E}(\tau_{K-1}) \leq A_K e^{T_K}, \tag{55}$$

and

$$\mathcal{E}(\tau_{K-2}) \leq \left( A_{K-1} + D_{K-1} \mathcal{E}(\tau_{K-1}) \right) e^{T_{K-1}} \leq \left( A_{K-1} + D_{K-1} A_K e^{T_K} \right) e^{T_{K-1}}. \tag{56}$$

Continuing inductively, one obtains for the initial time $t = 0 = \tau_0$ the general bound

$$\mathcal{E}_{\mathrm{hog}}(0) \leq \sum_{k=1}^{K} A_k \exp \left( \sum_{m=1}^{k} T_m \right) \prod_{j=1}^{k-1} D_j, \tag{57}$$

where the empty product is defined as $\prod_{j=1}^{0} D_j := 1$.

Since each $D_j$ is nonnegative, the elementary inequality $D_j \leq e^{D_j}$ for all $x \geq 0$ implies

$$\mathcal{E}_{\mathrm{hog}}(0) \leq \sum_{k=1}^{K} A_k \exp \left( \sum_{m=1}^{k} T_m + \sum_{j=1}^{k-1} D_j \right) \leq \left( \sum_{k=1}^{K} A_k \right) \exp \left( \sum_{m=1}^{K} T_m + \sum_{j=1}^{K-1} D_j \right). \tag{58}$$

The latter inequality holds as all quantities in the exponent are nonnegative, so enlarging the summation ranges can only increase the exponent.

Therefore, we obtain the compact multi-stage error bound

$$\mathcal{E}_{\mathrm{hog}}(0) \leq A_{\mathrm{hog}}\, e^{T_{\mathrm{hog}}}, \tag{59}$$

where $A_{\mathrm{hog}} := \sum_{k=1}^{K} A_k$, and $T_{\mathrm{hog}} := \sum_{m=1}^{K} T_m + \sum_{j=1}^{K-1} D_j$. This bound admits the same functional form as in Eq. (54) but accumulates the contributions of all $K$ reverse stages from $T = \tau_K$ to $0 = \tau_0$.

**Step 3: Comparing the two error bounds.**

We now compare $(A_{\mathrm{cls}}, T_{\mathrm{cls}})$ in Eq. (54) with $(A_{\mathrm{hog}}, T_{\mathrm{hog}})$ in Eq. (59).

First, by construction each subinterval $[\tau_{k-1}, \tau_k]$ lies inside $[0, T]$ and the OU bridge horizon of the $k$-th HOG-Diff stage satisfies $\tau_k \leq T$. The corresponding drift and diffusion coefficients are therefore pointwise dominated by those of the classical bridge:

$$g_{k,s}^4 \leq g_s^4, \qquad h_{k,s}^2 \leq h_s^2, \qquad \gamma_k(s) \leq \gamma_{\mathrm{cls}}(s), \qquad s \in [\tau_{k-1}, \tau_k]. \tag{60}$$

Since all these quantities are nonnegative, their integrals over a subinterval are bounded by the integrals over the full horizon:

$$\sum_{k=1}^{K} \int_{\tau_{k-1}}^{\tau_k} g_{k,s}^4 \mathrm{d}s \leq \int_0^T g_s^4 \mathrm{d}s, \quad \sum_{k=1}^{K} \int_{\tau_{k-1}}^{\tau_k} \gamma_k(s) \mathrm{d}s \leq \int_0^T \gamma_{\mathrm{cls}}(s) \mathrm{d}s, \tag{61}$$

and

$$\sum_{k=1}^{K} \int_{\tau_{k-1}}^{\tau_k} h_{k,s}^2 \mathrm{d}s \leq \int_0^T h_s^2 \, \mathrm{d}s. \tag{62}$$

For the coefficient $A_{\mathrm{hog}}$, using the definition in Eq. (52) we have

$$A_{\mathrm{hog}} = C^2 \sum_{k=1}^{K} \ell^{(k)}(\boldsymbol{\theta}) \int_{\tau_{k-1}}^{\tau_k} g_{k,s}^4 \, \mathrm{d}s \leq C^2 \left( \sum_{k=1}^{K} \ell^{(k)}(\boldsymbol{\theta}) \right) \int_0^T g_s^4 \, \mathrm{d}s. \tag{63}$$

At the level of optimal score-matching losses, Lemma 6 shows that the coarse-to-fine curriculum of HOG-Diff cannot be intrinsically worse than the single-stage formulation: $\sum_{k=1}^{K} \ell_\star^{(k)} \leq \ell_{\mathrm{cls},\star}$. In the regime where the optimization error is negligible and the loss is dominated by approximation error, it is natural to focus on solutions for which the curriculum does not increase the total score-matching loss, namely

$$\sum_{k=1}^{K} \ell^{(k)}(\boldsymbol{\theta}) \leq \ell_{\mathrm{cls}}(\boldsymbol{\varphi}). \tag{64}$$

This condition is clearly satisfied in our experiments. For $K = 2$, the combined spectrum loss of the Coarse and Fine stages, denoted as Coarse+Fine in Fig. 4, is lower than that of the classical model (Classical (Noise)), providing empirical support for the above comparison.

Hence, we can bound $A_{\mathrm{hog}}$ as

$$A_{\mathrm{hog}} \leq C^2 \ell_{\mathrm{cls}}(\boldsymbol{\varphi}) \int_0^T g_s^4 \mathrm{d}s = A_{\mathrm{cls}}. \tag{65}$$

For the exponent $T_{\mathrm{hog}}$ in Eq. (59), we use the same domination to obtain

$$\begin{aligned} T_{\mathrm{hog}} &= \sum_{k=1}^{K} T_k + \sum_{k=1}^{K-1} D_k = \sum_{k=1}^{K} \int_{\tau_{k-1}}^{\tau_k} \gamma_k(s) \mathrm{d}s + C \sum_{k=1}^{K-1} \int_{\tau_{k-1}}^{\tau_k} h_{k,s}^2 \mathrm{d}s \\ &\leq \int_0^T \gamma_{\mathrm{cls}}(s) \mathrm{d}s + C \int_0^T h_s^2 \mathrm{d}s \\ &= T_{\mathrm{cls}}. \end{aligned} \tag{66}$$

Together with the bounds in Eq. (54) and Eq. (59), we obtain

$$A_{\text{hog}} e^{T_{\text{hog}}} \leq A_{\text{cls}} e^{T_{\text{cls}}}. \tag{67}$$

Combining these inequalities, we can finally conclude $\mathcal{E}_{\text{hog}}(0) \leq \mathcal{E}_{\text{cls}}(0)$, which shows that the reconstruction error bound of HOG-Diff over the full trajectory $T \to 0$ is no worse, and can be strictly sharper, than the corresponding bound for the classical diffusion model.

$\square$

## B  ADDITIONAL EXPLANATION ON RELATED WORKS

### B.1  HIGHER-ORDER NETWORKS

Graphs provide elegant and practical abstractions for modeling irregular relationships in empirical systems, transforming unstructured data into analyzable representations. However, their inherent restriction to pairwise interactions limits their ability to represent group dynamics (Battiston et al., 2020). For example, cyclic structures like benzene rings and functional groups play a holistic role in molecular networks; densely interconnected structures, like simplices, often have a collective influence on social networks; and functional brain networks exhibit higher-order dependencies. To address this, various topological formalisms have been employed to describe data in terms of its higher-order relations, including simplicial complexes, cell complexes, and combinatorial complexes (Hajij et al., 2023). As such, the study of higher-order networks has gained increasing attention for their capacity to capture higher-order interactions, with broad applications across domains such as influence maximization (Zeng et al., 2024), financial system analysis (Mai et al., 2024; Xiao et al., 2024), graph signal processing (Sardellitti et al., 2021; Roddenberry et al., 2022), and graph representation learning (Bodnar et al., 2021; Huang et al., 2024; Papamarkou et al., 2024).

Given their broad applicability and theoretical richness, we next focus on two core frameworks for modelling higher-order interactions: simplicial complexes and cell complexes.

**Simplicial Complexes**. Simplicial complexes (SCs) are fundamental concepts in algebraic topology that flexibly subsume pairwise graphs (Hatcher, 2001). Specifically, simplices generalize fundamental geometric structures such as points, lines, triangles, and tetrahedra, enabling the modelling of higher-order interactions in networks. They offer a robust framework for capturing multi-way relationships that extend beyond pairwise connections typically represented in classical networks.

A simplicial complex $\mathcal{X}$ consists of a set of simplices of varying dimensions, including vertices (dimension 0), edges (dimension 1), and triangles (dimension 2).

A $d$-dimensional simplex is formed by a set of $(d+1)$ interacting nodes and includes all the subsets of $\delta + 1$ nodes (with $\delta < d$), which are called the $\delta$-dimensional faces of the simplex. A simplicial complex of dimension $d$ is formed by simplices of dimension at most $d$ glued along their faces.

**Definition 7** (Simplicial complexes). *A simplicial complex $\mathcal{X}$ is a finite collection of node subsets closed under the operation of taking nonempty subsets, and such a node subset $\sigma \in \mathcal{X}$ is called a simplex.*

We can obtain a clique complex, a particular kind of SCs, by extracting all cliques from a given graph and regarding them as simplices. This implies that an empty triangle (owning $[v_1, v_2], [v_1, v_3], [v_2, v_3]$ but without $[v_1, v_2, v_3]$) cannot occur in clique complexes.

**Cell Complexes**. Cell complexes (CCs) generalize simplicial complexes by incorporating generalized building blocks called cells instead of relying solely on simplices (Hatcher, 2001). This broader approach allows for the representation of many-body interactions that do not adhere to the strict requirements of simplicial complexes. For example, a square can be interpreted as a cell of four-body interactions whose faces are just four links. This flexibility is advantageous in scenarios such as social networks, where, for instance, a discussion group might not involve all-to-all pairwise interactions, or in protein interaction networks, where proteins in a complex may not bind pairwise.

Formally, a cell complex is termed regular if each attaching map is a homeomorphism onto the closure of the associated cell's image. Regular cell complexes generalize graphs, simplicial complexes, and polyhedral complexes while retaining many desirable combinatorial and intuitive properties of

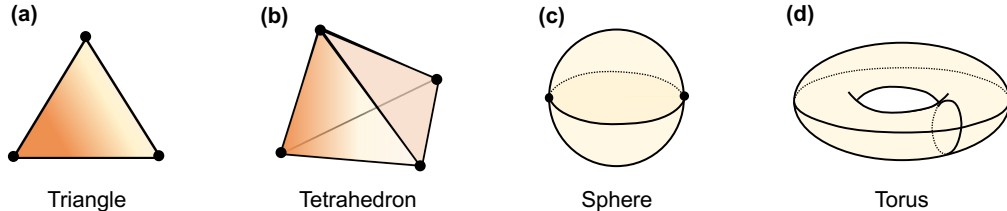

Figure 5: Visual illustration of cell complexes. (**a**) Triangle. (**b**) Tetrahedron. (**c**) Sphere. (**d**) Torus.

these simpler structures. In this paper, all cell complexes will be regular and consist of finitely many cells.

As shown in Fig. 5 (**a**) and (**b**), triangles and tetrahedra are two particular types of cell complexes called simplicial complexes (SCs). The only 2-cells they allow are triangle-shaped. The sphere shown in Fig. 5 (**c**) is a 2-dimensional cell complex. It is constructed using two 0-cells (*i.e.*, nodes), connected by two 1-cells (*i.e.*, the edges forming the equator). The equator serves as the boundary for two 2-dimensional disks (the hemispheres), which are glued together along the equator to form the sphere. The torus in Fig. 5 (**d**) is a 2-dimensional cell complex formed by attaching a single 1-cell to itself in two directions to form the loops of the torus. The resulting structure is then completed by attaching a 2-dimensional disk, forming the surface of the torus. Note that this is just one way to represent the torus as a cell complex, and other decompositions might lead to different numbers of cells and faces.

These topological frameworks provide the mathematical foundation for capturing multi-way interactions beyond pairwise graphs. Building upon this background, we next review advances in graph generative models, highlighting how existing approaches attempt to learn graph distributions and where they fall short in preserving such higher-order structures.

## B.2 GRAPH GENERATIVE MODELS

The study of graph generation seeks to synthesize graphs that align with the observed distribution. Graph generation has been extensively studied, which dates back to the early works of the random network models, such as the Erdős–Rényi (ER) model (Erdős et al., 1960) and the Barabási-Albert (BA) model (Barabási & Albert, 1999). While these models offer foundational insights, they are often too simplistic to capture the complexity of graph distributions we encounter in practice.

Recent graph generative models have made great progress in graph distribution learning by exploiting the capacity of deep neural networks. GraphRNN (You et al., 2018) and GraphVAE (Simonovsky & Komodakis, 2018) adopt sequential strategies to generate nodes and edges. MolGAN (De Cao & Kipf, 2018) integrates generative adversarial networks (GANs) with reinforcement learning objectives to synthesize molecules with desired chemical properties. Shi et al. (2020) generates molecular graphs using a flow-based approach, while GraphDF (Luo et al., 2021) adopts an autoregressive flow-based model with discrete latent variables. Additionally, GraphEBM (Liu et al., 2021) employs an energy-based model for molecular graph generation. However, the end-to-end structure of these methods often makes them more challenging to train compared to diffusion-based generative models.

**Diffusion-based Generative Models**. A leap in graph generative models has been marked by the recent progress in diffusion-based generative models (Song et al., 2021). EDP-GNN (Niu et al., 2020) generates the adjacency matrix by learning the score function of the denoising diffusion process, while GDSS (Jo et al., 2022) extends this framework by simultaneously generating node features and an adjacency matrix with a joint score function capturing the node-edge dependency. DiGress (Vignac et al., 2023) addresses the discretization challenge due to Gaussian noise, while CDGS (Huang et al., 2023) designs a conditional diffusion model based on discrete graph structures. GSDM (Luo et al., 2023a) introduces an efficient graph diffusion model driven by low-rank diffusion SDEs on the spectrum of adjacency matrices. HypDiff (Fu et al., 2024) introduces a geometrically latent diffusion on hyperbolic space to preserve the anisotropy of the graph. Despite these advancements, current methods are ineffective at modeling the topological properties of higher-order systems since

learning to denoise the noisy samples does not explicitly lead to preserving the intricate structural dependencies required for generating realistic graphs.

**Diffusion Bridge**. Diffusion bridge processes, *i.e.*, processes conditioned to the endpoints, have been widely adopted in image-related domains, including image generation (De Bortoli et al., 2021), image translation (Zhou et al., 2024), and image restoration (Luo et al., 2023b; Yue et al., 2024). Recently, several studies have improved the graph generative framework of diffusion models by leveraging the diffusion bridge processes. Wu et al. (2022) inject physical information into the process by incorporating informative prior to the drift. GruM (Jo et al., 2024) utilizes the OU bridge to condition the diffusion endpoint as the weighted mean of all possible final graphs. GLAD (Boget et al., 2025) employs the Brownian bridge on a discrete latent space with endpoints conditioned on data samples. However, existing methods often overlook or inadvertently disrupt the higher-order topological structures in the graph generation process.

**Hierarchical and Fragment-based Generation**. Several recent studies have also explored hierarchical and fragment-based generative frameworks. HiGen (Karami, 2024) decomposes graph generation into multiple layers of abstraction, using separate neural networks to model intra-community structures and inter-community connections at each level. GPrinFlowNet (Mo et al., 2024) proposes a semantic-preserving framework based on a low-to-high frequency generation curriculum, where the $k$-th intermediate generation state corresponds to the $k$ smallest principal components of the adjacency matrices. Dymond (Zeno et al., 2021) focuses on temporal motifs in dynamic graph generation. HierDiff (Qiang et al., 2023) progressively generates fragment-level 3D geometries, refines them into fine-grained fragments, and then assembles these fragments into complete molecules. MiCaM (Geng et al., 2023) synthesizes molecules by iteratively merging motifs.

**Higher-order Generative Models**. Since higher-order structures are intrinsic to many real-world systems, incorporating them into generative models could yield more faithful representations of complex phenomena. Existing efforts have primarily explored higher-order information through hypergraphs. HypeBoy (Kim et al., 2024) introduces a self-supervised hypergraph representation framework based on a hyperedge filling task, which enhances embeddings rather than performing direct generation. Hygene (Gailhard et al., 2025) reduces hypergraph generation to standard graph generation via a bipartite representation. However, no prior approach has explicitly integrated higher-order topology due to the stricter challenges of modeling multi-way rather than pairwise dependencies.

## C    DETAILS OF HOG-DIFF

This section elucidates our spectral diffusion methodology, the parameterization of the score network, and the associated training and sampling procedures.

### C.1    OVERVIEW

As shown in Fig. 6, HOG-Diff employs a hierarchical, coarse-to-fine generation curriculum, where both forward diffusion and reverse denoising processes are decomposed into $K$ easy-to-learn subprocesses. Each subprocess is realized using the GOU bridge process.

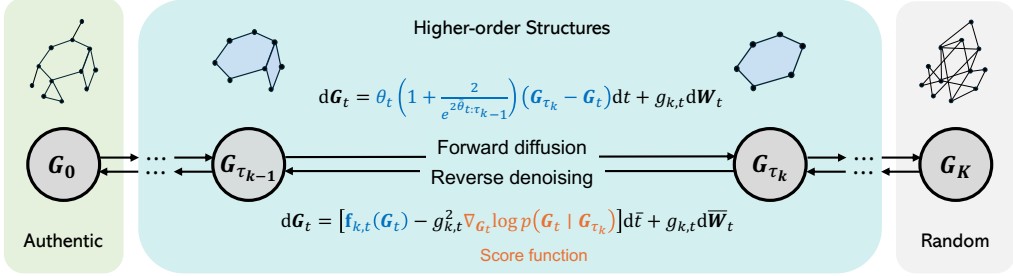

Figure 6: Illustration of the coarse-to-fine generation process in HOG-Diff using the generalized OU bridge.

## C.2 SPECTRAL DIFFUSION

Classical graph diffusion approaches typically inject isotropic Gaussian noise directly into adjacency matrices $\boldsymbol{A}$, leading to various fundamental challenges. Firstly, the inherent non-uniqueness of graph representations implies that a graph with $n$ vertices can be equivalently modelled by up to $n!$ distinct adjacency matrices. This ambiguity requires a generative model that assigns probabilities uniformly across all equivalent adjacencies to accurately capture the graph's inherent symmetry. Additionally, unlike densely distributed image data, graphs typically follow a Pareto distribution and exhibit sparsity (Ghavasieh & De Domenico, 2024), so that adjacency score functions lie on a low-dimensional manifold. Consequently, noise injected into out-of-support regions of the full adjacency space severely degrades the signal-to-noise ratio, impairing the training of the score-matching process. Even for densely connected graphs, isotropic noise distorts global message-passing patterns by encouraging message-passing on sparsely connected regions. Moreover, the adjacency matrix scales quadratically with the number of nodes, making the direct generation of adjacency matrices computationally prohibitive for large-scale graphs.

To address these challenges, inspired by Martinkus et al. (2022) and Luo et al. (2023a), we introduce noise in the eigenvalue domain of the graph Laplacian matrix $\boldsymbol{L} = \boldsymbol{D} - \boldsymbol{A}$, instead of the adjacency matrix $\boldsymbol{A}$, where $\boldsymbol{D}$ denotes the diagonal degree matrix. As a symmetric positive semi-definite matrix, the graph Laplacian can be diagonalized as $\boldsymbol{L} = \boldsymbol{U}\boldsymbol{\Lambda}\boldsymbol{U}^\top$. Here, the orthogonal matrix $\boldsymbol{U} = [\boldsymbol{u}_1, \cdots, \boldsymbol{u}_n]$ comprises the eigenvectors, and the diagonal matrix $\boldsymbol{\Lambda} = \mathrm{diag}(\lambda_1, \cdots, \lambda_n)$ holds the corresponding eigenvalues. The relationship between the Laplacian spectrum and the graph's topology has been extensively explored (Chung, 1997). For instance, the low-frequency components of the spectrum capture the global structural properties such as connectivity and clustering, whereas the high-frequency components are crucial for reconstructing local connectivity patterns. Therefore, the target graph distribution $p(\boldsymbol{G}_0)$ represents a joint distribution of $\boldsymbol{X}_0$ and $\boldsymbol{\Lambda}_0$, exploiting the permutation invariance and structural robustness of the Laplacian spectrum.

Consequently, we split the reverse-time SDE into two parts that share drift and diffusion coefficients as

$$\begin{cases} \mathrm{d}\boldsymbol{X}_t = \left[\mathbf{f}_{k,t}(\boldsymbol{X}_t) - g_{k,t}^2 \nabla_{\boldsymbol{X}} \log p_t(\boldsymbol{G}_t|\boldsymbol{G}_{\tau_k})\right] \mathrm{d}\bar{t} + g_{k,t}\mathrm{d}\bar{\boldsymbol{W}}_t^1 \\ \mathrm{d}\boldsymbol{\Lambda}_t = \left[\mathbf{f}_{k,t}(\boldsymbol{\Lambda}_t) - g_{k,t}^2 \nabla_{\boldsymbol{\Lambda}} \log p_t(\boldsymbol{G}_t|\boldsymbol{G}_{\tau_k})\right] \mathrm{d}\bar{t} + g_{k,t}\mathrm{d}\bar{\boldsymbol{W}}_t^2 \end{cases}. \tag{68}$$

Here, the superscript of $\boldsymbol{X}_t^{(k)}$ and $\boldsymbol{\Lambda}_t^{(k)}$ are dropped for simplicity, and $\mathbf{f}_{k,t}$ is determined according to Eq. (7). Note that, for molecular datasets, the intermediate endpoint is obtained by cell complex filtering: it preserves coarse higher-order connectivity, but the resulting node features can carry only an approximate atom-type signal. To stabilize training, we therefore add a small masked Gaussian perturbation to its node features during training

In addition, we conduct a comparative evaluation of HOG-Diff under two generative settings: one operating directly in the adjacency matrix domain, and the other in the Laplacian spectral domain. Using a consistent hyperparameter search space, the results summarized in Tab. 12 show that generation in the spectral domain generally achieves comparable performance across most evaluation metrics. We adopt the Laplacian spectral domain as the default diffusion space in HOG-Diff, as the spectral approach is more efficient and better aligned with theoretical principles such as permutation invariance and signal concentration on low-dimensional manifolds.

## C.3 SCORE NETWORK PARAMETRIZATION

The score network in HOG-Diff is a critical component responsible for estimating the score functions required to reverse the diffusion process effectively. The architecture of the proposed score network is depicted in Fig. 7. The input $\boldsymbol{A}_t$ is computed from $\boldsymbol{U}_0$ and $\boldsymbol{\Lambda}_t^{(k)}$ using the relation $\boldsymbol{A}_t = \boldsymbol{D}_t^{(k)} - \boldsymbol{L}_t^{(k)}$, where the Laplacian matrix is given by $\boldsymbol{L}_t^{(k)} = \boldsymbol{U}_0 \boldsymbol{\Lambda}_t^{(k)} \boldsymbol{U}_0^\top$ and the diagonal degree matrix is given by $\boldsymbol{D}_t^{(k)} = \mathrm{diag}\left(\boldsymbol{L}_t^{(k)}\right)$. To enhance the input to the Attention module, we derive enriched node and edge features using the $l$-step random walk matrix obtained from the binarized $\boldsymbol{A}_t$. Specifically, the arrival probability vector is incorporated as additional node features, while the truncated shortest path distance derived from the same matrix is employed as edge features. Temporal information is integrated into the outputs of the Attention and GCN modules using

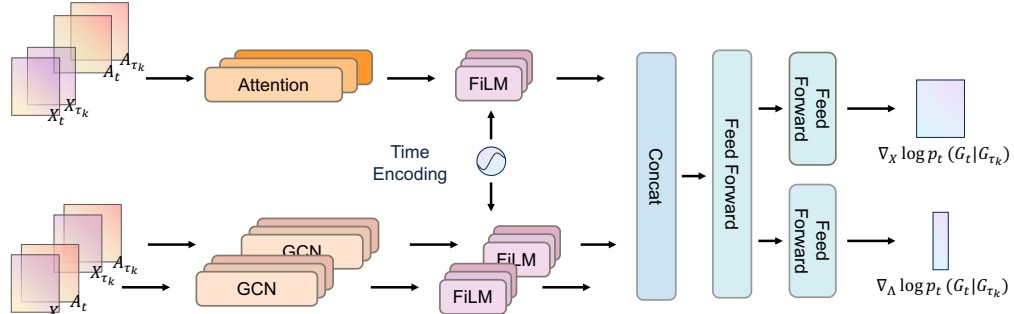

Figure 7: Score Network Architecture of HOG-Diff. The score network integrates GCN and Attention blocks to capture both local and global features, and further incorporates time information through FiLM layers. These enriched outputs are subsequently concatenated and processed by separate feed-forward networks to produce predictions for $\nabla_{\boldsymbol{X}_t} \log p_t(\boldsymbol{G}_t | \boldsymbol{G}_{\tau_k})$ and $\nabla_{\boldsymbol{\Lambda}_t} \log p_t(\boldsymbol{G}_t | \boldsymbol{G}_{\tau_k})$, respectively.

Feature-wise Linear Modulation (FiLM) (Perez et al., 2018) layers, following sinusoidal position embeddings (Waswani et al., 2017).

A graph processing module is considered permutation invariant if its output remains unchanged under any permutation of its input, formally expressed as $f(\boldsymbol{G}) = x \iff f(\pi(\boldsymbol{G})) = x$, where $\pi(\boldsymbol{G})$ represents a permutation of the input graph $\boldsymbol{G}$. It is permutation equivariant when the output undergoes the same permutation as the input, formally defined as $f(\pi(\boldsymbol{G})) = \pi(f(\boldsymbol{G}))$. It is worth noting that our score network model is permutation equivalent, as each model component avoids any node ordering-dependent operations.

## C.4 Training and Sampling Procedure

The diffusion process in HOG-Diff is divided into $K$ hierarchical intervals, denoted by $\{[\tau_{k-1}, \tau_k]\}_{k=1}^{K}$, where $0 = \tau_0 < \cdots < \tau_{k-1} < \tau_k < \cdots < \tau_K = T$. Within each interval $[\tau_{k-1}, \tau_k]$, we employ the GOU bridge process to ensure smooth transitions between intermediate states $\boldsymbol{G}_{\tau_{k-1}}$ and $\boldsymbol{G}_{\tau_k}$. We apply the cell complex filtering (CCF) operation at each interval to obtain structured, topologically meaningful intermediate states $\boldsymbol{G}_{\tau_k} := \text{CCF}(\boldsymbol{G}, \mathcal{S}, p)$. Specifically, CCF prunes nodes and edges that are not contained in the closure of any $p$-cell within a given cell complex $\mathcal{S}$. At the initial state, the filtering operation is defined as $\text{CCF}(\boldsymbol{G}, \mathcal{S}, 0) = \boldsymbol{G}$, *i.e.*, the filtering operation leaves the input unchanged. A special case arises at the final step, where the intermediate state is initialized from Gaussian noise, *i.e.*, $\text{CCF}(\boldsymbol{G}, \mathcal{S}, K) \sim \mathcal{N}(\boldsymbol{0}, \boldsymbol{I})$. Since the GOU bridge process naturally reduces to a standard diffusion process when the terminal distribution is Gaussian noise, we omit the GOU bridge in the final segment $[\tau_{K-1}, \tau_K]$, and instead use the Variance Preserving (VP) SDE (Ho et al., 2020; Song et al., 2021). In our experiments, we adopt a two-stage generation process, *i.e.*, $K = 2$. The intermediate state $\boldsymbol{G}_{\tau_1}$ is obtained via 2-cell complex filtering for molecule generation tasks, or via the 3-simplicial complex filtering for generic graph generation tasks. The rationale for this choice of filtering strategy is discussed in App. F.4.

To approximate the score functions $\nabla_{\boldsymbol{X}_t} \log p_t(\boldsymbol{G}_t | \boldsymbol{G}_{\tau_k})$ and $\nabla_{\boldsymbol{\Lambda}_t} \log p_t(\boldsymbol{G}_t | \boldsymbol{G}_{\tau_k})$, we employ a neural network $\boldsymbol{s}_{\boldsymbol{\theta}}^{(k)}(\boldsymbol{G}_t, \boldsymbol{G}_{\tau_k}, t)$, as introduced in App. C.3. This model consists of a node $(\boldsymbol{s}_{\boldsymbol{\theta}, \boldsymbol{X}}^{(k)}(\boldsymbol{G}_t, \boldsymbol{G}_{\tau_k}, t))$ and a spectrum $(\boldsymbol{s}_{\boldsymbol{\theta}, \boldsymbol{\Lambda}}^{(k)}(\boldsymbol{G}_t, \boldsymbol{G}_{\tau_k}, t))$ output. The network is trained by minimizing the following score-matching loss:

$$
\begin{aligned}
\ell^{(k)}(\boldsymbol{\theta}) = \mathbb{E}_{t, \boldsymbol{G}_t, \boldsymbol{G}_{\tau_{k-1}}, \boldsymbol{G}_{\tau_k}} \{ \omega(t) [c_1 \| \boldsymbol{s}_{\boldsymbol{\theta}, \boldsymbol{X}}^{(k)} - \nabla_{\boldsymbol{X}} \log p_t(\boldsymbol{G}_t | \boldsymbol{G}_{\tau_k}) \|_2^2 \\
+ c_2 \| \boldsymbol{s}_{\boldsymbol{\theta}, \boldsymbol{\Lambda}}^{(k)} - \nabla_{\boldsymbol{\Lambda}} \log p_t(\boldsymbol{G}_t | \boldsymbol{G}_{\tau_k}) \|_2^2 ] \},
\end{aligned}
\tag{69}
$$

where $\omega(t)$ is a positive weighting function, and $c_1, c_2$ control the relative importance of vertices and spectrum.

The overall generation procedure is as follows. We sample $(\hat{\boldsymbol{X}}_{\tau_K}, \hat{\boldsymbol{\Lambda}}_{\tau_K})$ from the prior distribution and select $\hat{\boldsymbol{U}}_0$ as an eigenbasis drawn from the training set. Reverse diffusion is then ap-

plied across multiple stages, sequentially generating $(\hat{\boldsymbol{X}}_{\tau_{K-1}}, \hat{\boldsymbol{\Lambda}}_{\tau_{K-1}}), \ldots, (\hat{\boldsymbol{X}}_1, \hat{\boldsymbol{\Lambda}}_1), (\hat{\boldsymbol{X}}_0, \hat{\boldsymbol{\Lambda}}_0)$, where each stage is implemented via the diffusion bridge and initialized from the output of the previous step. Finally, plausible samples with higher-order structures can be reconstructed as $\hat{\boldsymbol{G}}_0 = (\hat{\boldsymbol{X}}_0, \hat{\boldsymbol{L}}_0 = \hat{\boldsymbol{U}}_0 \hat{\boldsymbol{\Lambda}}_0 \hat{\boldsymbol{U}}_0^\top)$.

We provide the pseudo-code of the training and sampling process in Alg. 1 and Alg. 2, respectively.

---

**Algorithm 1** Training Algorithm of HOG-Diff

**Input:** Score network $\boldsymbol{s}_{\boldsymbol{\theta}}^{(k)}$, training graph dataset $\mathcal{G}$, training epochs $M_k$.
**For the $k$-th step:**
1: **for** $m = 1$ **to** $M_k$ **do**
2:      Sample $\boldsymbol{G}_0 = (\boldsymbol{X}_0, \boldsymbol{A}_0) \sim \mathcal{G}$
3:      $\mathcal{S} \leftarrow \text{lifting}(\boldsymbol{G}_0)$
4:      $\boldsymbol{G}_{\tau_k} \leftarrow \text{CCF}(\boldsymbol{G}_0, \mathcal{S}, k)$, and $\boldsymbol{G}_{\tau_{k-1}} \leftarrow \text{CCF}(\boldsymbol{G}_0, \mathcal{S}, k-1)$          $\triangleright$ Cell complex filtering
5:      $\boldsymbol{U}_0 \leftarrow \text{EigenVectors}(\boldsymbol{D}_0 - \boldsymbol{A}_0)$
6:      $\boldsymbol{\Lambda}_{\tau_k} \leftarrow \text{EigenDecomposition}(\boldsymbol{D}_{\tau_k} - \boldsymbol{A}_{\tau_k})$
7:      $\boldsymbol{\Lambda}_{\tau_{k-1}} \leftarrow \text{EigenDecomposition}(\boldsymbol{D}_{\tau_{k-1}} - \boldsymbol{A}_{\tau_{k-1}})$
8:      Sample $t \sim \text{Unif}([0, \tau_k - \tau_{k-1}])$
9:      $\boldsymbol{X}_t^{(k)} \sim p(\boldsymbol{X}_t \mid \boldsymbol{X}_{\tau_{k-1}}, \boldsymbol{X}_{\tau_k})$          $\triangleright$ Eq. (6)
10:      $\boldsymbol{\Lambda}_t^{(k)} \sim p(\boldsymbol{\Lambda}_t \mid \boldsymbol{\Lambda}_{\tau_{k-1}}, \boldsymbol{\Lambda}_{\tau_k})$          $\triangleright$ Eq. (6)
11:      $\boldsymbol{L}_t^{(k)} \leftarrow \boldsymbol{U}_0 \boldsymbol{\Lambda}_t^{(k)} \boldsymbol{U}_0^\top$
12:      $\boldsymbol{A}_t^{(k)} \leftarrow \boldsymbol{D}_t^{(k)} - \boldsymbol{L}_t^{(k)}$
13:      $\ell^{(k)}(\boldsymbol{\theta}) \leftarrow c_1 \|\boldsymbol{s}_{\boldsymbol{\theta}, \boldsymbol{X}}^{(k)} - \nabla_{\boldsymbol{X}} \log p_t(\boldsymbol{G}_t | \boldsymbol{G}_{\tau_k})\|^2 + c_2 \|\boldsymbol{s}_{\boldsymbol{\theta}, \boldsymbol{\Lambda}}^{(k)} - \nabla_{\boldsymbol{\Lambda}} \log p_t(\boldsymbol{G}_t | \boldsymbol{G}_{\tau_k})\|^2$
14:      $\boldsymbol{\theta} \leftarrow \text{optimizer}(\ell^{(k)}(\boldsymbol{\theta}))$
15: **end for**
16: **Return:** $\boldsymbol{s}_{\boldsymbol{\theta}}^{(k)}$

---

**Algorithm 2** Sampling Algorithm of HOG-Diff

**Input:** Trained score network $\boldsymbol{s}_{\theta}^{(k)}$, diffusion time split $\{\tau_0, \cdots, \tau_K\}$, number of sampling steps $M_k$
1: $t \leftarrow \tau_K$
2: $\widehat{\boldsymbol{X}}_{\tau_K} \sim \mathcal{N}(\boldsymbol{0}, \boldsymbol{I})$ and $\widehat{\boldsymbol{\Lambda}}_{\tau_K} \sim \mathcal{N}(\boldsymbol{0}, \boldsymbol{I})$
3: $\widehat{\boldsymbol{U}}_0 \sim \text{Unif}\left(\{\boldsymbol{U}_0 \triangleq \text{EigenVectors}(\boldsymbol{L}_0)\}\right)$
4: $\widehat{\boldsymbol{G}}_{\tau_K} \leftarrow (\widehat{\boldsymbol{X}}_{\tau_K}, \widehat{\boldsymbol{\Lambda}}_{\tau_K}, \widehat{\boldsymbol{D}}_{\tau_K} - \widehat{\boldsymbol{U}}_0 \widehat{\boldsymbol{\Lambda}}_{\tau_K} \widehat{\boldsymbol{U}}_0^\top)$
5: **for** $k = K$ **to** $1$ **do**
6:      **for** $m = M_k - 1$ **to** $0$ **do**
7:          $\boldsymbol{S}_{\boldsymbol{X}}, \boldsymbol{S}_{\boldsymbol{\Lambda}} \leftarrow \boldsymbol{s}_{\boldsymbol{\theta}}^{(k)}(\widehat{\boldsymbol{G}}_t, \widehat{\boldsymbol{G}}_{\tau_k}, t)$
8:          $\widehat{\boldsymbol{X}}_t \leftarrow \widehat{\boldsymbol{X}}_t - \left[\theta_t \left(1 + \frac{2}{e^{2\bar{\theta}t:\tau_{k-1}} - 1}\right)(\widehat{\boldsymbol{X}}_{\tau_k} - \widehat{\boldsymbol{X}}_t) - g_{k,t}^2 \boldsymbol{S}_{\boldsymbol{X}}\right]\delta t + g_{k,t}\sqrt{\delta t}\boldsymbol{w}_{\boldsymbol{X}}, \boldsymbol{w}_{\boldsymbol{X}} \sim \mathcal{N}(\boldsymbol{0}, \boldsymbol{I})$
9:          $\widehat{\boldsymbol{\Lambda}}_t \leftarrow \widehat{\boldsymbol{\Lambda}}_t - \left[\theta_t \left(1 + \frac{2}{e^{2\bar{\theta}t:\tau_{k-1}} - 1}\right)(\widehat{\boldsymbol{\Lambda}}_{\tau_k} - \widehat{\boldsymbol{\Lambda}}_t) - g_{k,t}^2 \boldsymbol{S}_{\boldsymbol{\Lambda}}\right]\delta t + g_{k,t}\sqrt{\delta t}\boldsymbol{w}_{\boldsymbol{\Lambda}}, \boldsymbol{w}_{\boldsymbol{\Lambda}} \sim \mathcal{N}(\boldsymbol{0}, \boldsymbol{I})$
10:          $\widehat{\boldsymbol{L}}_t \leftarrow \widehat{\boldsymbol{U}}_0 \widehat{\boldsymbol{\Lambda}}_t \widehat{\boldsymbol{U}}_0^\top$
11:          $\widehat{\boldsymbol{A}}_t \leftarrow \widehat{\boldsymbol{D}}_t - \widehat{\boldsymbol{L}}_t$
12:          $t \leftarrow t - \delta t$
13:      **end for**
14:      $\widehat{\boldsymbol{A}}_{\tau_{k-1}} = \text{quantize}(\widehat{\boldsymbol{A}}_t)$          $\triangleright$ Quantize if necessary
15:      $\widehat{\boldsymbol{G}}_{\tau_{k-1}} \leftarrow (\widehat{\boldsymbol{X}}_t, \widehat{\boldsymbol{\Lambda}}_t, \widehat{\boldsymbol{A}}_t)$
16: **end for**
17: **Return:** $\widehat{\boldsymbol{X}}_0, \widehat{\boldsymbol{A}}_0$          $\triangleright$ $\tau_0 = 0$

# D  EFFICIENCY AND COMPLEXITY ANALYSIS

## D.1  ONE-TIME PREPROCESSING COMPLEXITY

When the targeted graph is not in the desired higher-order forms, one should also consider the one-time preprocessing procedure for cell filtering. Unlike *cell lifting*, which enumerates all cell structures and can incur substantial computational overhead, cell filtering can be performed much more efficiently. This is because filtering does not require the explicit enumeration of all cells; instead, it only checks whether individual nodes and edges participate in a cell. For instance, the 2-cell filter requires only checking whether each edge belongs to some cycle.

One method to achieve the 2-cell filter is to use a depth-first search (DFS) strategy. Starting from the adjacency matrix, we temporarily remove the edge $(i, j)$ and initiate a DFS from node $i$, keeping track of the path length. If the target node $j$ is visited within a path length of $l$, the edge $(i, j)$ is marked as belonging to a 2-cell of length at most $l$. In sparse graphs with $n$ nodes and $m$ edges, the time complexity of a single DFS is $\mathcal{O}(m + n)$. With the path length limited to $l$, the DFS may traverse up to $l$ layers of recursion in the worst case. Therefore, the complexity of a single DFS is $\mathcal{O}(\min(m + n, l \cdot k_{max}))$, where $k_{max}$ is the maximum degree of the graph. For all $m$ edges, the total complexity is $\mathcal{O}\left(m \cdot \min(m + n, l \cdot k_{max})\right)$.

Alternatively, matrix operations can be utilized to accelerate this process. By removing the edge $(i, j)$ from the adjacency matrix $A$ to obtain $\bar{A}$, the presence of a path of length $l$ between $i$ and $j$ can be determined by checking whether $\bar{A}^l_{i,j} > 0$. This indicates that the edge $(i, j)$ belongs to a 2-cell with a maximum length of $l + 1$. Assuming the graph has $n$ nodes and $m$ edges, the complexity of sparse matrix multiplication is $\mathcal{O}(mn)$. Since $l$ matrix multiplications are required, the total complexity is: $\mathcal{O}(l \cdot m^2 \cdot n)$. While this complexity is theoretically higher than the DFS approach, matrix methods can benefit from significant parallel acceleration on modern hardware, such as GPUs and TPUs. In practice, this makes the matrix-based method competitive, especially for large-scale graphs or cases where $l$ is large.

For simplicial complexes, the number of $p$-simplices in a graph with $n$ nodes and $m$ edges is upper-bounded by $\mathcal{O}(n^{p-1})$, and they can be enumerated in $\mathcal{O}(a\,(\mathcal{G})^{p-3} m)$ time (Chiba & Nishizeki, 1985), where $a\,(\mathcal{G})$ is the arboricity of the graph $\mathcal{G}$, a measure of graph sparsity. Since arboricity is demonstrated to be at most $\mathcal{O}(m^{1/2})$ and $m \leq n^2$, all $p$-simplices can thus be listed in $\mathcal{O}\left(n^{p-3}m\right)$. Besides, the complexity of finding 2-simplex is estimated to be $\mathcal{O}(\langle k \rangle\, m)$ with the Bron–Kerbosch algorithm (Bron & Kerbosch, 1973), where $\langle k \rangle$ denotes the average node degree, typically a small value for empirical networks.

## D.2  EMPIRICAL EFFICIENCY: PREPROCESSING, TRAINING, AND SAMPLING

To complement the theoretical analysis, we empirically evaluate the scalability of our approach on standard benchmarks of varying scales. As shown in Fig. 8, we record the wall-clock time for the filtering procedure on datasets ranging from small community graphs to large-scale molecular datasets such as ZINC250k and MOSES. The results indicate that the filtering time scales reasonably with data size. Given that this is a one-time preprocessing cost, it confirms that HOG-Diff is computationally feasible for large-scale graph generation tasks.

After preprocessing, the training loop is identical to that of standard diffusion-based generative models. The proposed coarse-to-fine strategy splits a long diffusion trajectory into shorter segments, each trained with a smaller smoothness constant, which by Theorem 3 provably leads to faster convergence than classical models in idealized settings. Furthermore, since the sub-processes are independent, they can be trained in parallel for additional efficiency gains. In practice, we observe no measurable slowdown relative to GDSS or DiGress.

Table 5: Training time comparison between GDSS and HOG-Diff.

| Method | Community-small | Ego-small | Enzymes | QM9 | ZINC250k |
|---|---|---|---|---|---|
| GDSS | 13 min | 19 min | 1.9 h | 1.2 h | 14.0 h |
| HOG-Diff | 20 min | 12 min | 0.5 h | 3.1 h | 6.5 h |

For inference, Tab. 6 shows that the extra guidance logic does not slow sampling. HOG-Diff matches DiGress on smaller graphs and achieves up to 12×speedup over both baselines on the protein-scale Enzymes dataset.

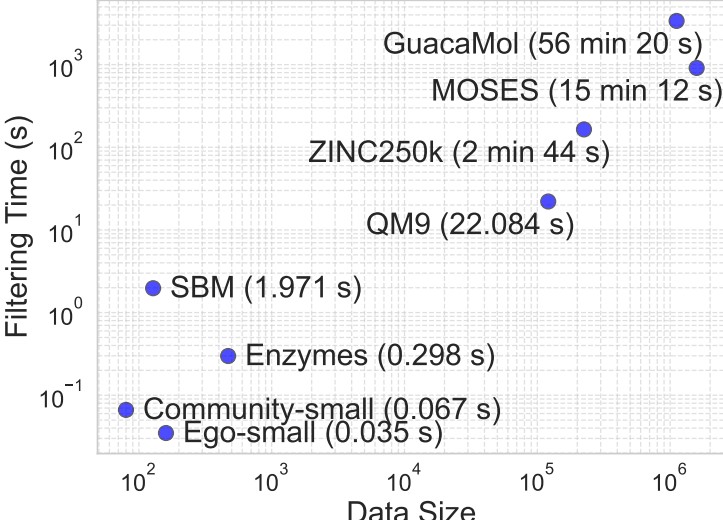

Figure 8: Empirical scalability of the filtering procedure. We report the wall-clock time required for the filtering (preprocessing) operation across datasets of varying sizes. The results show that the one-time preprocessing cost remains tractable even for large-scale datasets.

Table 6: Sampling time (s).

| Method | Community-small | Enzymes | Ego-small |
|---|---|---|---|
| GDSS | 41 | 110 | 28 |
| DiGress | 8 | 301 | 13 |
| HOG-Diff | 11 | 26 | 15 |

**Scalability on Large Benchmarks..** As shown in Fig. 8, the one-time filtering cost remains tractable across large-scale benchmarks, including datasets with substantially more training samples (MOSES) and larger graphs (SBM and GuacaMol). Importantly, this preprocessing is incurred only once; after that, the training loop is identical to standard diffusion models and the sampling overhead remains negligible. Detailed generation results on MOSES, GuacaMol, and SBM are reported in App. F.2.

In short, HOG-Diff scales gracefully: the extra cost is confined to a one-time preprocessing step, while the end-to-end runtime is dominated by standard GPU diffusion training and sampling.

# E EXPERIMENTAL SETUP

## E.1 COMPUTING RESOURCES

In this work, all experiments are conducted using PyTorch (Paszke, 2019) on a single NVIDIA L40S GPU with 46 GB memory and AMD EPYC 9374F 32-Core Processor.

## E.2 OVERVIEW OF COMPARED METHODS

For benchmarking, we include various representative molecular generation models. Autoregressive models include GraphAF (Shi et al., 2020), GraphDF (Luo et al., 2021), and GraphArm (Kong et al., 2023); the fragment-based MiCaM (Geng et al., 2023) creatively leverages motif information. In contrast, the remaining methods adopt a one-shot generation paradigm: GraphEBM serves as an energy-based model, SPECTRE (Martinkus et al., 2022) and GSDM (Luo et al., 2023a) incorporate spectral conditioning within GAN and diffusion frameworks, respectively, while EDP-GNN (Niu et al., 2020), GDSS (Jo et al., 2022), DiGress (Vignac et al., 2023), and Cometh (Siraudin et al., 2025) represent diffusion-based generation models. We also compare against advanced flow-based models, including MoFlow (Zang & Wang, 2020), CatFlow (Eijkelboom et al., 2024), and De-FoG (Qin et al., 2025).

## E.3 MOLECULE GENERATION

Early efforts in molecule generation introduce sequence-based generative models and represent molecules as SMILES strings (Kusner et al., 2017). Nevertheless, this representation frequently encounters challenges related to long dependency modelling and low validity issues, as the SMILES string fails to ensure absolute validity. Therefore, in recent studies, graph representations are more commonly employed for molecule structures where atoms are represented as nodes and chemical bonds as connecting edges (Jo et al., 2022). Consequently, this shift has driven the development of graph-based methodologies for molecule generation, which aim to produce valid, meaningful, and diverse molecules.

**QM9 and ZINC250k**. We evaluate the quality of generated molecules on two well-known molecular datasets: QM9 (Ramakrishnan et al., 2014) and ZINC250k (Irwin et al., 2012), and obtain the intermediate higher-order skeletons using the 2-cell complex filtering. In experiments, each molecule is preprocessed into a graph comprising adjacency matrix $A \in \{0, 1, 2, 3\}^{n \times n}$ and node feature matrix $X \in \{0, 1\}^{n \times d}$, where $n$ denotes the maximum number of atoms in a molecule of the dataset, and $d$ is the number of possible atom types. The entries of $A$ indicate the bond types: 0 for no bond, 1 for the single bond, 2 for the double bond, and 3 for the triple bond. Further, we scale $A$ with a constant scale of 3 in order to bound the input of the model in the interval [0, 1], and rescale the final sample of the generation process to recover the bond types. Following the standard procedure (Shi et al., 2020; Luo et al., 2021), all molecules are kekulized by the RDKit library (Landrum et al., 2016) with hydrogen atoms removed. In addition, we make use of the valency correction proposed by Zang & Wang (2020). After generating samples by simulating the reverse diffusion process, the adjacency matrix entries are quantized to discrete values $0, 1, 2, 3$ by applying value clipping. Specifically, values in $(-\infty, 0.5)$ are mapped to 0, $[0.5, 1.5)$ to 1, $[1.5, 2.5)$ to 2, and $[2.5, +\infty)$ to 3, ensuring the bond types align with their respective categories.

We report the baseline results taken from Jo et al. (2022) and Kong et al. (2023) or as reproduced using publicly available code.

To comprehensively assess the quality of the generated molecules across datasets, we evaluate 10,000 generated samples using several key metrics: validity, validity with correction (Val. w/ corr.), uniqueness, novelty (Jo et al., 2022), Frechet ChemNet Distance (FCD) (Preuer et al., 2018), and Neighborhood Subgraph Pairwise Distance Kernel (NSPDK) MMD (Costa & Grave, 2010).

- **Validity** computes the fraction of valid molecules before any corrections Jo et al. (2022), providing insight into the intrinsic quality of the generative process. While **validity with correction (Val. w/ corr.)** is measured as the fraction of valid molecules to all generated molecules after applying post-processing corrections such as valency adjustments or edge

resampling. Whether molecules are valid is generally determined by compliance with the valence rules in RDKit (Landrum et al., 2016).

- **Novelty** assesses the model's ability to generalize by calculating the percentage of generated graphs that are not subgraphs of the training set, with two graphs considered identical if they are isomorphic.

- **Uniqueness** quantifies the diversity of generated molecules as the ratio of unique samples to valid samples, removing duplicates that are subgraph-isomorphic, ensuring variety in the output.

- **FCD** quantifies the similarity between generated and test molecules by leveraging the activations of ChemNet's penultimate layer, assessing the generation quality within the chemical space.

- **NSPDK**-MMD evaluates the generation quality from the graph topology perspective by computing the MMD between the generated and test sets while considering both node and edge features.

**MOSES**. We conducted experiments on the MOSES dataset (Polykovskiy et al., 2020), a refinement of the ZINC database for molecular generation containing approximately 1.9 million lead-like molecules. This dataset serves as a rigorous testbed for assessing whether the proposed higher-order guidance remains computationally feasible and effective at scale. Following Vignac et al. (2023), the reported scores for FCD, SNN, and Scaffold similarity are computed on the dataset made of separate scaffolds, which measures the ability of the networks to predict new ring structures.

**GuacaMol**. GuacaMol (Brown et al., 2019) is derived from the ChEMBL database and contains 1.4M molecules, from which 1.1M are used for training. We apply a preprocessing step similar to Vignac et al. (2023), filtering out molecules that cannot be mapped from SMILES to a graph and back to SMILES. In Fig. 3(left), we report the scores of the FCD and KL metrics following Brown et al. (2019). Note that, unlike other settings, the FCD score is normalized using $FCDscore = \exp(-0.2 \cdot FCD)$ to obtain a final value between 0 and 1, meaning that higher values indicate better performance.

For the KL score, we compute the following descriptors for both the generated and reference molecules: BertzCT, MolLogP, MolWt, TPSA, NumHAcceptors, NumHDonors, NumRotatableBonds, NumAliphaticRings, NumAromaticRings, and the ECFP4 fingerprint-based similarity to the nearest neighbor. The KL divergence $D_{KL,i}$ is computed for each descriptor to measure the difference between the two molecular sets. These divergences are then combined to produce a final normalized score $S = \frac{1}{k} \sum_{i=1}^{k} \exp(-D_{KL,i})$.

### E.4 GENERIC GRAPH GENERATION

To display the topology distribution learning ability, we assess HOG-Diff over four common generic graph datasets: (**1**) Community-small, containing 100 randomly generated community graphs; (**2**) Ego-small, comprising 200 small ego graphs derived from the Citeseer network dataset; (**3**) Enzymes, featuring 587 protein graphs representing tertiary structures of enzymes from the BRENDA database; and (**4**) Stochastic Block Model (SBM), a larger-scale dataset.

**Community-small, Ego-small, and Enzymes**. We follow the standard experimental and evaluation settings from Jo et al. (2022), including the same train/test splits and the use of the Gaussian Earth Mover's Distance (EMD) kernel for MMD computation, to ensure fair comparisons with baseline models — except for the Stochastic Block Model (SBM) dataset. We use node degree and spectral features of the graph Laplacian decomposition as hand-crafted input features. Baseline results are sourced from Jo et al. (2022); Kong et al. (2023) or reproduced using the corresponding publicly available code.

**SBMs**. To further evaluate the scalability and robustness of HOG-Diff, we conduct experiments on the **Stochastic Block Model (SBM)** dataset, which comprises graphs of larger scale and is evaluated under a distinct experimental protocol from that used in Sec. 4.2. The dataset consists of 200 synthetic graphs generated using the stochastic block model. The number of communities is uniformly sampled between 2 and 5, and the number of nodes within each community is uniformly sampled

between 20 and 40. Edges are created with probabilities of 0.3 for intra-community connections and 0.05 for inter-community connections.

We adopt the evaluation setting introduced by Martinkus et al. (2022), including the same data splits, feature initialization, and the use of Total Variation (TV) distance to compute the MMDs. The TV kernel is adopted since it offers higher computational efficiency compared to the Earth Mover's Distance (EMD) kernel, especially for large graphs. We also report the percentage of valid, unique, and novel (V.U.N.) samples among the generated graphs to further assess the ability of our model to capture the properties of the targeted distributions correctly. Baseline results are sourced from Vignac et al. (2023); Jo et al. (2024) or reproduced using the corresponding publicly available code.

### E.5 DATA STATISTICS

Tab. 7 summarizes the key characteristics of the datasets utilized in this study. The table outlines the type of dataset, the total number of graphs, and the range of graph sizes ($|V|$). Additionally, it also provides the number of distinct node types and edge types for each dataset. Notably, the Community-small and Ego-small datasets contain relatively small graphs, whereas Enzymes, SBM and the molecular datasets exhibit greater diversity in terms of graph size and complexity.

Table 7: Dataset summary.

| Dataset | Graph type | #Graphs | #Nodes | Node types | Edge types |
|---|---|---|---|---|---|
| Community-small | Synthetic | 100 | $12 \leq |V| \leq 20$ | 1 | 1 |
| Ego-small | Citation | 200 | $4 \leq |V| \leq 18$ | 1 | 1 |
| Enzymes | Protein | 587 | $10 \leq |V| \leq 125$ | 1 | 1 |
| SBM | Synthetic | 200 | $44 \leq |V| \leq 187$ | 1 | 1 |
| QM9 | Molecule | 133,885 | $1 \leq |V| \leq 9$ | 4 | 3 |
| ZINC250k | Molecule | 249,455 | $6 \leq |V| \leq 38$ | 9 | 3 |
| GuacaMol | Molecule | 1,398,223 | $2 \leq |V| \leq 88$ | 12 | 3 |
| MOSES | Molecule | 1,936,962 | $8 \leq |V| \leq 27$ | 7 | 3 |

## F ADDITIONAL EXPERIMENTAL RESULTS

### F.1 EXTENDED RESULTS ON STANDARD MOLECULAR BENCHMARKS: QM9 AND ZINC250K

To visually assess the capability of the proposed method in molecular generation, Fig. 9 compares distributional results on the molecular datasets. Specifically, we first calculate the Morgan fingerprints (Rogers & Hahn, 2010) of all molecules, which have been widely utilized in drug discovery for capturing structural information. Subsequently, we apply Uniform Manifold Approximation and Projection (UMAP), a nonlinear dimensionality-reduction method that preserves local similarities, to embed the fingerprints into two dimensions and plot the resulting distributions.

On the QM9 dataset, the distributions generated by our model closely align with those of the reference set. On the more complex ZINC250k dataset, the generated distributions show slight deviations but remain well aligned. These results provide an intuitive demonstration of strong generative performance and complement the findings in Tab. 1.

We provide the standard deviation results and the additional metric validity with correction (Val. w/ corr.) in Tabs. 8 and 9. Baseline results are sourced from Jo et al. (2022); Kong et al. (2023) or reproduced using the corresponding publicly available code.

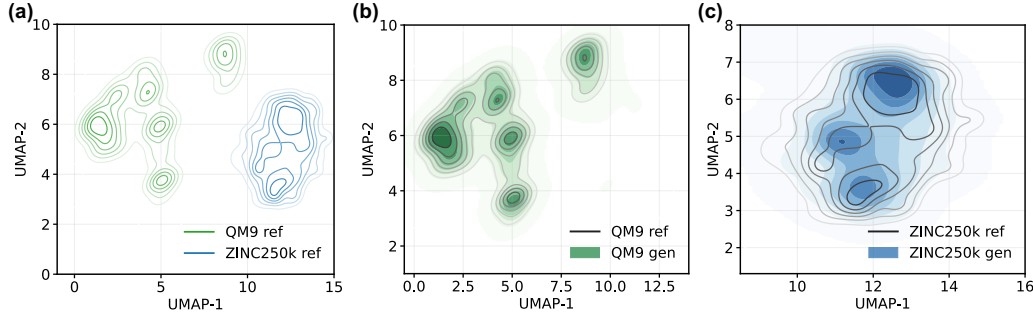

Figure 9: Visualization of chemical space distributions. Molecular representations are obtained using Morgan fingerprints and subsequently visualized through dimensionality reduction with Uniform Manifold Approximation and Projection (UMAP). Contour lines denote the probability distributions of reference (ref) and generated (gen) molecules. (**a**) illustrates the distributional shift across QM9 (green) and ZINC250k (blue), while (**b**) and (**c**) show that our generative model faithfully captures the distinct distributions of each dataset.

Table 8: Comparison of different methods on QM9. We report the means and standard deviations of 3 runs. The **best** results are highlighted in bold. Asterisks (*) indicate that the source did not report standard deviations.

| Method | Val.↑ | Val. w/ corr.↑ | Uni.↑ | Nov.↑ | FCD↓ | NSPDK↓ |
|---|---|---|---|---|---|---|
| GraphAF (Shi et al., 2020) | 74.43±2.55 | 100.00±0.00 | 88.64±2.37 | 86.59±1.95 | 5.625±0.259 | 0.021±0.003 |
| GraphDF (Luo et al., 2021) | 93.88±4.76 | 100.00±0.00 | 98.58±0.25 | **98.54**±0.48 | 10.928±0.038 | 0.064±0.000 |
| GraphArm* (Kong et al., 2023) | 90.25 | 100.00 | 95.62 | 70.39 | 1.220 | 0.002 |
| MiCaM* (Geng et al., 2023) | **99.93** | 100.00 | 93.89 | 83.25 | 1.045 | 0.001 |
| GraphEBM (Liu et al., 2021) | 8.22±2.24 | 100.00±0.00 | 97.90±0.05 | 97.01±0.17 | 6.143±0.411 | 0.030±0.004 |
| SPECTRE* (Martinkus et al., 2022) | 87.30 | 100.00 | 35.70 | 97.28 | 47.960 | 0.163 |
| GSDM* (Luo et al., 2023a) | 99.90 | 100.00 | - | - | 2.650 | 0.003 |
| EDP-GNN (Niu et al., 2020) | 47.52±3.60 | 100.00±0.00 | 99.25±0.05 | 86.58±1.85 | 2.680±0.221 | 0.005±0.001 |
| GDSS (Jo et al., 2022) | 95.72±1.94 | 100.00±0.00 | 98.46±0.61 | 86.27±2.29 | 2.900±0.282 | 0.003±0.000 |
| DiGress* (Vignac et al., 2023) | 99.00 | 100.00 | 96.66 | 33.40 | 0.360 | 0.0005 |
| MoFlow (Zang & Wang, 2020) | 91.36±1.23 | 100.00±0.00 | 98.65±0.57 | 94.72±0.77 | 4.467±0.595 | 0.017±0.003 |
| CatFlow* (Eijkelboom et al., 2024) | 99.81 | 100.00 | **99.95** | - | 0.441 | - |
| Cometh (Siraudin et al., 2025) | 99.59±0.09 | 100.0±0.00 | 96.75±0.21 | 72.06±0.19 | 0.248±0.012 | 0.0005±0.0003 |
| DeFoG (Qin et al., 2025) | 99.26±0.10 | 100.00±0.00 | 96.61±0.30 | 72.57±1.89 | 0.268±0.006 | 0.0005±0.0001 |
| **HOG-Diff (Ours)** | 98.74±0.22 | 100.00±0.00 | 97.14±0.11 | 75.12±0.39 | **0.172**±0.002 | **0.0003**±0.0000 |

Table 9: Comparison of different methods on ZINC250k. We report the means and standard deviations of 3 runs. The **best** results are highlighted in bold. Asterisks (*) indicate that the source did not report standard deviations.

| Method | Val.↑ | Val. w/ corr.↑ | Uni.↑ | Nov.↑ | FCD↓ | NSPDK↓ |
|---|---|---|---|---|---|---|
| GraphAF (Shi et al., 2020) | 68.47±0.99 | 100.00±0.00 | 98.64±0.69 | 99.99±0.01 | 16.023±0.451 | 0.044±0.005 |
| GraphDF (Luo et al., 2021) | 90.61±4.30 | 100.00±0.00 | 99.63±0.01 | 100.00±0.00 | 33.546±0.150 | 0.177±0.001 |
| GraphArm* (Kong et al., 2023) | 88.23 | 100.00 | 99.46 | 100.00 | 16.260 | 0.055 |
| MiCaM* (Geng et al., 2023) | **100.00** | 100.00 | 88.48 | 99.98 | 31.495 | 0.166 |
| GraphEBM (Liu et al., 2021) | 5.29±3.83 | 99.96±0.02 | 98.79±0.15 | 100.00±0.00 | 35.471±5.331 | 0.212±0.075 |
| SPECTRE* (Martinkus et al., 2022) | 90.20 | 100.00 | 67.05 | 100.00 | 18.440 | 0.109 |
| GSDM* (Luo et al., 2023a) | 92.70 | 100.00 | - | - | 12.956 | 0.017 |
| EDP-GNN (Niu et al., 2020) | 82.97±2.73 | 100.00±0.00 | 99.79±0.08 | 100.00±0.00 | 16.737±1.300 | 0.049±0.006 |
| GDSS (Jo et al., 2022) | 97.01±0.77 | 100.00±0.00 | 99.64±0.13 | 100.00±0.00 | 14.656±0.680 | 0.019±0.001 |
| DiGress* (Vignac et al., 2023) | 91.02 | 100.00 | 81.23 | 100.00 | 23.060 | 0.082 |
| MoFlow (Zang & Wang, 2020) | 63.11±5.17 | 100.00±0.00 | 99.99±0.01 | 100.00±0.00 | 20.931±0.184 | 0.046±0.002 |
| CatFlow* (Eijkelboom et al., 2024) | 99.95 | 99.99 | **100.00** | - | 13.211 | - |
| DeFoG (Qin et al., 2025) | 94.97±0.026 | 99.99±0.01 | 99.98±0.02 | 100.00±0.00 | 2.030±0.031 | 0.002±0.001 |
| **HOG-Diff (Ours)** | 98.56±0.12 | 100.00±0.00 | 99.96±0.02 | 99.53±0.07 | **1.633**±0.012 | **0.001**±0.001 |

## F.2 SCALABILITY EVALUATION ON LARGE BENCHMARKS: MOSES AND SBM

To address concerns regarding scalability and performance on larger benchmarks, we extended our evaluation to encompass both large-scale molecular datasets and generic graph benchmarks.

**MOSES**. We conducted experiments on the MOSES dataset (Polykovskiy et al., 2020). The results are presented in Tab. 10. HOG-Diff demonstrates superior performance in terms of distribution learning, achieving the lowest FCD score. This indicates that our method more accurately captures the underlying chemical and topological distribution of the large-scale training set.

Table 10: Generation performance on the large-scale MOSES dataset. Top results are highlighted in **bold**.

| Method | Val. ↑ | Uni. ↑ | Nov. ↑ | Filters ↑ | FCD ↓ | SNN | Scaf |
|---|---|---|---|---|---|---|---|
| GraphINVENT (Mercado et al., 2021) | 96.4 | 99.8 | - | 95.0 | 1.22 | 0.54 | 12.7 |
| DiGress (Vignac et al., 2023) | 85.7 | **100.0** | 95.0 | 97.1 | 1.19 | 0.52 | 14.8 |
| DisCo (Xu et al., 2024) | 88.3 | **100.0** | **97.7** | 95.6 | 1.44 | 0.50 | 15.1 |
| Cometh (Siraudin et al., 2025) | 90.5 | 99.9 | 92.6 | **99.1** | 1.27 | 0.54 | 16.0 |
| DeFoG (Qin et al., 2025) | 92.8 | 99.9 | 92.1 | 98.9 | 1.95 | 0.55 | 14.4 |
| **HOG-Diff (Ours)** | **99.7**±0.2 | **100.0** | 89.1 | 96.7 | **0.94** | 0.52 | 8.6 |

**SBMs**. To further evaluate the scalability and robustness of HOG-Diff, we report results on the **Stochastic Block Model (SBM)** dataset in Tab. 11, which comprises graphs of larger scale and is evaluated under a distinct experimental protocol from that used in Sec. 4.2. As shown in Tab. 11, HOG-Diff achieves competitive performance compared to the state-of-the-art.

Table 11: Generation results on the SBM dataset. The **best** and second-best results are highlighted in bold and underlined, respectively.

| Method | Deg.↓ | Clus.↓ | Orb.↓ | Spec.↓ | Avg.↓ | V.U.N.↑ |
|---|---|---|---|---|---|---|
| GraphRNN (You et al., 2018) | 0.0055 | 0.0584 | 0.0785 | 0.0065 | 0.0372 | 0.05 |
| GRAN (Liao et al., 2019) | 0.0113 | 0.0553 | 0.0540 | 0.0054 | 0.0315 | 0.25 |
| EDP-GNN (Niu et al., 2020) | 0.0011 | 0.0552 | 0.0520 | 0.0070 | 0.0288 | - |
| SPECTRE (Martinkus et al., 2022) | 0.0015 | 0.0521 | **0.0412** | 0.0056 | 0.0251 | 0.53 |
| GDSS (Jo et al., 2022) | 0.0212 | 0.0646 | 0.0894 | 0.0128 | 0.0470 | 0.05 |
| RefineGen (Bergmeister et al., 2024) | 0.0141 | 0.0528 | 0.0809 | 0.0071 | 0.0387 | 0.75 |
| GruM (Jo et al., 2024) | 0.0015 | 0.0589 | 0.0450 | 0.0077 | 0.0283 | **0.85** |
| DeFoG (Qin et al., 2025) | **0.0006** | 0.0517 | 0.0556 | 0.0054 | 0.0283 | 0.80 |
| **HOG-Diff (Ours)** | 0.0028 | 0.0500 | 0.0428 | **0.0043** | **0.0249** | 0.83 |

### F.3 ANALYSIS OF DIFFUSION DOMAIN CHOICE

To investigate the impact of diffusion domain choice, we perform ablation experiments comparing two variants of HOG-Diff: one that operates directly in the adjacency matrix domain, and another in the Laplacian spectral domain. Both variants are trained under the same hyperparameter search space for a fair comparison.

As shown in Tab. 12, the spectral variant achieves comparable performance to the adjacency-based approach across most evaluation metrics on QM9 and ZINC250k. Despite similar results, we adopt the Laplacian spectral domain as the default diffusion space in HOG-Diff due to its theoretical and practical advantages. Specifically, the spectral domain offers greater efficiency and aligns naturally with core graph principles such as permutation invariance and signal concentration on low-dimensional manifolds.

Table 12: Comparison of diffusion domains in HOG-Diff.

| Dataset | Domain | NSPDK↓ | FCD↓ | Val. ↑ | Val. w/ corr.↑ | Uni.↑ | Nov.↑ |
|---------|--------|--------|------|--------|----------------|-------|-------|
| QM9 | Adjacency matrix | 0.0004 | 0.264 | **99.08** | 100.00 | 95.90 | 67.78 |
| | Laplacian spectrum | **0.0003** | **0.172** | 98.74 | 100.00 | **97.10** | **75.12** |
| ZINC250k | Adjacency matrix | 0.006 | 4.259 | 96.75 | 100.00 | 99.78 | **99.98** |
| | Laplacian spectrum | **0.001** | **1.633** | **98.56** | 100.00 | **99.96** | 99.53 |

### F.4 RATIONALE FOR SELECTING TOPOLOGICAL FILTERS

The filtering strategy in HOG-Diff is primarily driven by data statistics and complemented by domain knowledge to ensure meaningful choices.

In molecular generation experiments, we employ 2-cells as guides because they are ubiquitous in molecular graphs and capture critical chemical information, *e.g.*, functional groups. In particular, 2-cells play a crucial role in determining the three-dimensional conformation, electron distribution, and target binding mode of compounds, making them highly informative. Consequently, Tab. 1 shows that conditioning on 2-cells alone provides strong guidance and achieves competitive results across all metrics.

In contrast, cells with dimension $> 2$ are extremely sparse in the standard benchmarks (see Tab. 13). Identifying them requires increasingly complex preprocessing, such as detecting candidate higher-order topological structures and verifying their validity, which introduces non-trivial computational overhead. Moreover, given that these benchmarks contain only a few hundred nodes, guides with higher-dimensional cells would cover less than 0.05% of possible structures, introducing both unnecessary computation and a risk of overfitting without evident benefit. In practice, 2-cells already capture virtually all of the higher-order structure present in these datasets.

We further tested guiding molecular generation with 2-simplices but observed worse performance. This is expected, as summarized in Tab. 13, higher-dimensional simplices are extremely rare in molecular datasets (*i.e.*, QM9 and ZINC250k), while 2-cells are comparatively more abundant. Moreover, most functional structures in molecules do not satisfy the requirements of simplices. These findings confirm that aligning the guide with the natural higher-order structures of the data improves both training efficiency and sample quality, whereas mismatched guides provide limited benefit.

In contrast, generic graphs (particularly those in social or biological domains) often exhibit diverse simplicial structures, making simplicial filtering more suitable. This demonstrates the adaptability of HOG-Diff across domains.

Overall, the proposed framework is most effective when the intermediate skeletons reflect structures naturally present in the data. This aligns with prior work in graph representation learning, which shows that the benefits of higher-order representations are most pronounced in datasets that are rich in such structures (Huang et al., 2024). Indeed, many real-world networks, including social, biological, and citation networks, exhibit abundant higher-order cells and simplices. Thus, many real-world

tasks naturally fall into the "topologically rich" regime in which our method is particularly advantageous.

Table 13: Average counts of simplices and cells across datasets. "-" indicates computation exceeded 10h.

| Dataset | 0-simplices | 1-simplices | 2-simplices | 3-simplices | 4-simplices | 2-cells | 3-cells |
|---|---|---|---|---|---|---|---|
| Community-small | 15.28 | 35.15 | 27.45 | 8.24 | 0.76 | 40.92 | 0.00 |
| Ego-small | 6.41 | 8.70 | 4.61 | 1.64 | 0.35 | 5.66 | 0.00 |
| Enzymes | 33.03 | 63.27 | 25.93 | 3.05 | 0.01 | – | – |
| SBM | 104.01 | 500.41 | 441.25 | 90.88 | 4.42 | – | – |
| QM9 | 8.80 | 9.40 | 0.47 | 0.00 | 0.00 | 1.84 | 0.10 |
| ZINC250k | 23.15 | 24.90 | 0.06 | 0.00 | 0.00 | 2.77 | 0.00 |

## F.5    ADDITIONAL VISUALIZATIONS

Fig. 10 provides a qualitative view of the generative trajectory of HOG-Diff for the model trained on ZINC250k. It shows several intermediate graph states from the initial noisy prior to the final molecule, illustrating the coarse-to-fine refinement process.

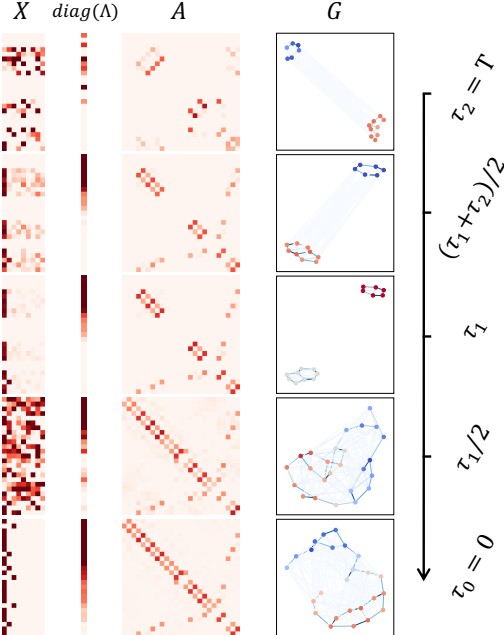

Figure 10: Visualization of molecular graphs at different stages of the reverse generative process. Model trained on the ZINC250k dataset.

In this section, we additionally provide the visualizations of the generated graphs for both molecule generation tasks and generic graph generation tasks. Figs. 11-16 illustrate non-curated generated samples. HOG-Diff demonstrates the capability to generate high-quality samples that closely resemble the topological properties of empirical data while preserving essential structural details.

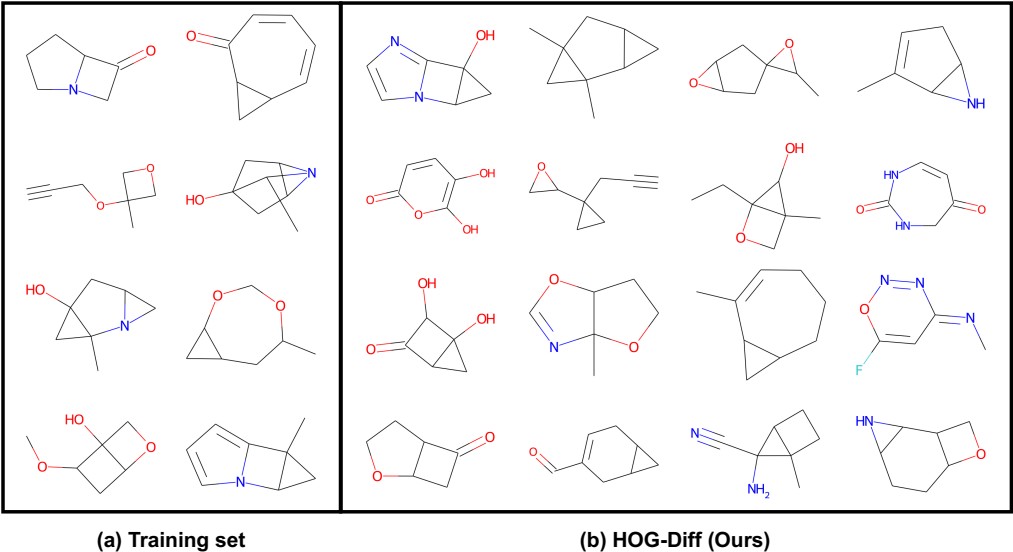

Figure 11: Visualization of random samples taken from the HOG-Diff trained on the QM9 dataset.

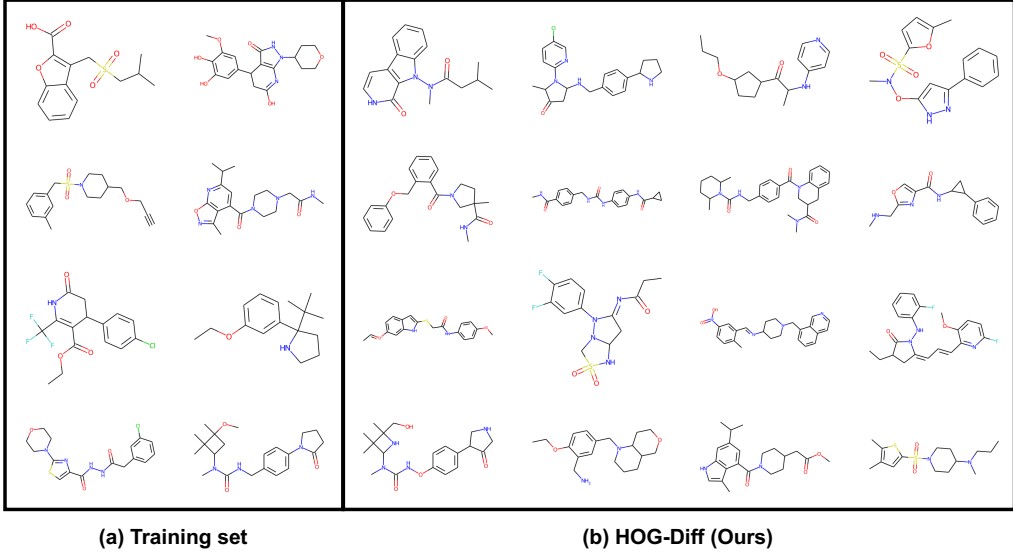

Figure 12: Visualization of random samples taken from the HOG-Diff trained on the ZINC250k dataset.

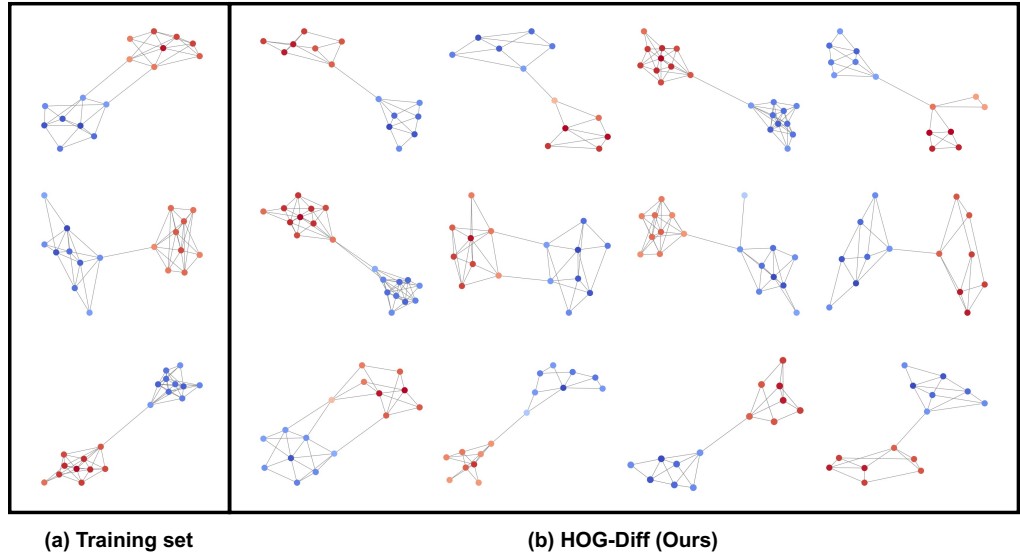

(a) Training set                                    (b) HOG-Diff (Ours)

Figure 13: Visual comparison between training set graph samples and generated graph samples produced by HOG-Diff on the Community-small dataset.

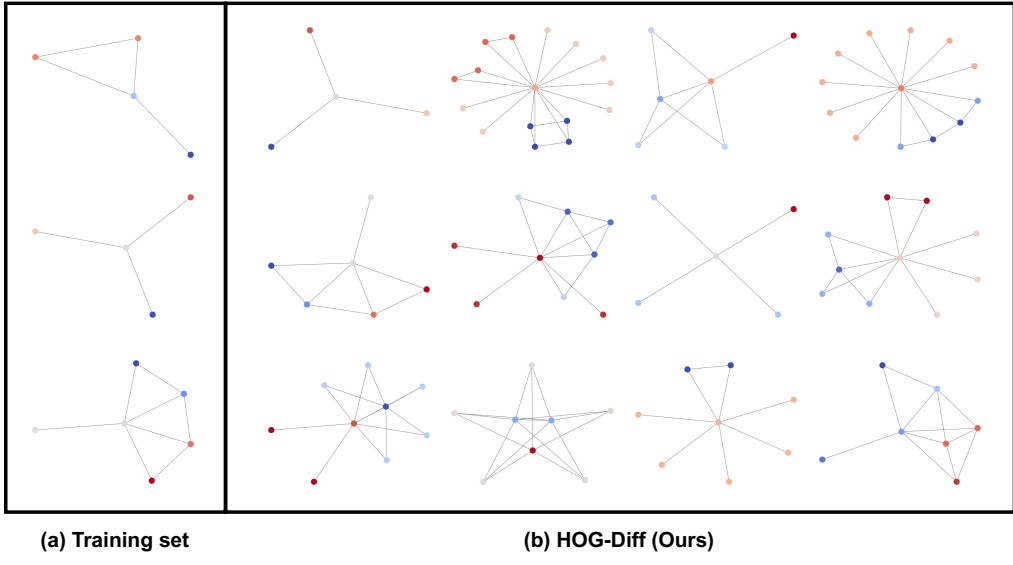

(a) Training set                                    (b) HOG-Diff (Ours)

Figure 14: Visual comparison between training set graph samples and generated graph samples produced by HOG-Diff on the Ego-small dataset.

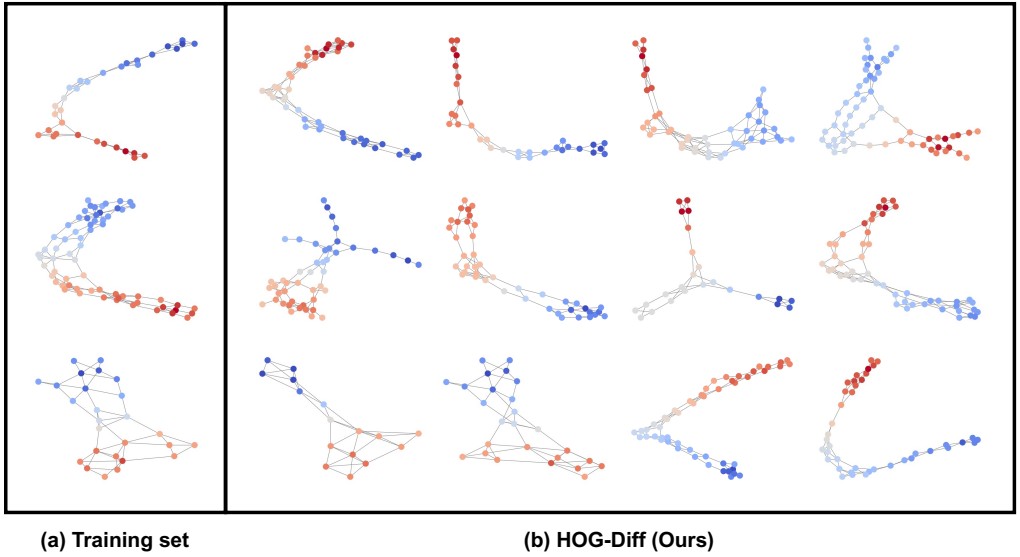

(a) Training set    (b) HOG-Diff (Ours)

Figure 15: Visual comparison between training set graph samples and generated graph samples produced by HOG-Diff on the Enzymes dataset.

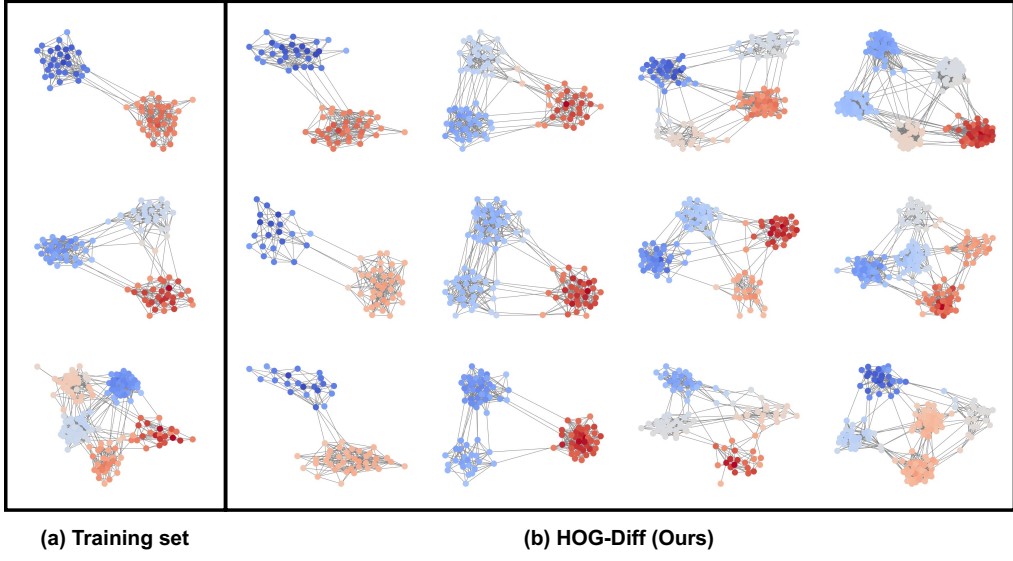

(a) Training set    (b) HOG-Diff (Ours)

Figure 16: Visual comparison between training set graph samples and generated graph samples produced by HOG-Diff on the SBM dataset.

## G    LIMITATIONS AND FURTHER WORK

We propose a principled graph generation framework that explicitly exploits higher-order topological cues to guide the generative process. This design enables HOG-Diff to achieve strong empirical performance across various tasks, including molecule and generic graph generation. While HOG-Diff shows superior performance, future work would benefit from improving our framework.

As discussed in Sec. 4.2, the performance of the proposed framework depends on the presence of explicit higher-order structures. Although previous studies have shown that such structures are prevalent in many empirical systems, certain types of graphs, such as the ego-small dataset, lack this topological richness. In these cases, the benefits of higher-order diffusion guidance diminish, and the performance advantage becomes less pronounced.

In addition, our framework is built around the use of higher-order structures as diffusion guides, enabled by the Cell Complex Filtering (CCF) mechanism. As detailed in App. D, we introduce a simplified computational formulation of the CCF that facilitates efficient implementation of both cell complex and simplicial complex filters. However, extending this framework to capture more intricate topological elements, such as motifs, higher-order cells (beyond second order), and topological cavities, poses significant computational challenges. We leave this scalability bottleneck as future work.

