# OpenReview forum: "HOG-Diff: Higher-Order Guided Diffusion for Graph Generation"
_ICLR.cc/2026/Conference — ICLR 2026 Poster_

### Official Review · Reviewer_ftpE · 2025-11-01

**Soundness:** 3
**Presentation:** 2
**Contribution:** 3
**Rating:** 6
**Confidence:** 4

**Summary:**

The authors propose the Higher-order Guided Diffusion (HOG-Diff) framework that guides the diffusion process to preserve higher-order topological structures in a coarse-to-fine manner. HOG-Diff utilizes cell complex filtering (CCF), which extracts hierarchical skeletons used as a generation guidance and enables efficient computation. The effectiveness of HOG-Diff is demonstrated through extensive experiments on molecular and benchmark graph datasets with autoregressive and one-shot graph generation methods.

**Strengths:**

- Novel and interesting idea of integrating topology and diffusion as a guiding structure.
- Solid theoretical grounding suggests how GOU process and Doob’s h-transform can be derived to guide topological information in diffusion process, as well as the error bounds.
- Comprehensive evaluation on various graph datasets using a wide range of baseline graph generative methods.
- Insightful dataset-level analysis, in terms of the discussion on dataset-dependent effects (e.g., weaker improvement on Ego-small due to limited higher-order structures).

**Weaknesses:**

- Lack of topological assessment: The authors do not provide evaluation(quantitative or qualitative) on higher-order topology preservation in the generated graphs.
- Notation and presentation complexity: I understand it may be somewhat inevitable, but the excessive use of notations is hard to follow for readers.

**Questions:**

- As mentioned in the weakness section, can you provide quantitative and qualitative evaluation on the generated graphs that could verify the successful preservation of higher-order structures? Such information will support the claim that the performance improvement lies on the preservation of higher-order structure. (e.g., MSE of persistence image, Curvature Filtrations [1], etc.)
- What advantages does guidance have over direct conditioning on higher-order information in the diffusion process? (other than interpretability)
- The term “consistently outperforms” in L400 seems little excessive, as only the NSPDK and FCD scores in Table 1 shows improvement whereas the other three metrics differ.
- What exactly do you mean by explicitly preserves higher-order structures in L399 about Fig.3? I don’t see the alignment of specific structures like the connected components or loops. (Also, the light-blue edges are extremely hard to see. I recommend changing the color to be more visible.)
- Can you clarify the novelty of CCF? I am aware of the idea of simplifying simplicial complex while preserving topological invariants used in discrete morse theory or other TDL works. It would be nice to show how CCF differs from such methods.

---

> ### Author Response · Authors · 2025-11-21
>
> We sincerely thank the reviewer for their constructive feedback. We are delighted to see that the reviewer found our idea both novel and interesting with solid theoretical grounding. We address the specific concerns below.
>
>     W1+Q1: evaluation (quantitative or qualitative) on higher-order topology preservation in the generated graphs.
>
> We thank the reviewer for this constructive suggestion.
> In the revised manuscript, we now provide both quantitative and qualitative evaluations of higher-order topology preservation.
>
> Quantitatively, the **newly added Section 4.3** (Topological Preservation Analysis) introduces an evaluation based on Curvature Filtrations [1], as suggested. We employ two filtration functions: (1) Balanced Forman--Ricci curvature $\kappa_{FR}$ and (2) Ollivier--Ricci curvature $\kappa_{OR}$.
> The results, summarized in the **newly added Table 3** and **Fig. 1 (right)**, show that HOG-Diff consistently achieves the lowest distances across both molecular and generic graph benchmarks, indicating superior preservation of higher-order structures compared with SOTA baselines.
>
> Qualitatively, we have updated visualizations of generative trajectories (see **Fig. 3, right**). These examples clearly show that HOG-Diff maintains ring structures throughout the reverse diffusion process, rather than passing through topologically unstructured noisy graphs.
>
> Table A (Table 3). Quantitative evaluation of higher-order topology preservation using Curvature Filtrations.
>
> |Dataset|Method|Forman|Ollivier|
> |-|-|-|-|
> |QM9|GDSS|0.925|0.601|
> |QM9|DiGress|0.251|0.343|
> |QM9|DoFoG|0.177|0.286|
> |QM9|Cometh|0.216|0.314|
> |QM9|HOG-Diff|0.077|0.098|
> |ZINC250k|MiCaM|9.436|7.251|
> |ZINC250k|GDSS|1.781|1.331|
> |ZINC250k|DoFoG|0.728|0.498|
> |ZINC250k|HOG-Diff|0.190|0.098|
> |SBM|DoFoG|3.998|5.206|
> |SBM|Cometh|2.748|3.938|
> |SBM|HOG-Diff|1.453|3.740|
> |Community-small|GDSS|2.311|1.301|
> |Community-small|HOG-Diff|1.679|1.042|
> |Enzymes|GDSS|13.031|12.399|
> |Enzymes|HOG-Diff|12.114|9.830|
> |Ego-small|GDSS|2.311|1.301|
> |Ego-small|HOG-Diff|1.679|1.042|
>
>     W2: Notation and presentation complexity
>
> We appreciate the reviewer’s feedback regarding notation complexity. Although some level of notation is inevitable, we fully agree that clarity must take precedence. In response, we have supplemented the formal mathematical presentation with clearer intuitive explanations alongside the notation, making the pipeline easier to follow without sacrificing rigor. Thank you again for this helpful suggestion.
>
>     Q2: What advantages does guidance have over direct conditioning on higher-order information in the diffusion process? (other than interpretability)
>
> We thank the reviewer for this thoughtful question.
> First, guidance acts on the diffusion trajectory rather than only on the network input.
> For graphs, the forward process quickly destroys structure and intermediate adjacency matrices become dense and nearly uniform, so higher-order constraints are largely washed out in the middle of the trajectory.
> In HOG-Diff, each time window is modeled as a generalized OU bridge whose terminal state is a cell-filtered skeleton $G_{\tau_k}$. This keeps the entire trajectory in each window concentrated around a meaningful higher-order structure instead of drifting through unstructured noise, which direct conditioning alone cannot guarantee.
>
> Second, the guided coarse-to-fine curriculum reduces optimization difficulty by decomposing a single hard task into simpler subproblems. A directly conditioned model must learn, in one pass from $T$ to $0$, to recover both local edges and complex higher-order topology from high noise. In contrast, HOG-Diff first learns to reach higher-order skeletons and then refines them. Theorem 3 shows that each guided stage has a smaller smoothness constant and therefore a faster exponential convergence rate than classical one-stage diffusion, which is reflected empirically in faster training curves in **Fig. 4**.
>
> Third, guidance provides stronger control of sampling error. Theorem 4 establishes a reconstruction error bound for HOG-Diff that is provably tighter than the corresponding bound for a classical diffusion process from $T$ to $0$. This improvement comes from the additional bridge drift term that continuously pulls samples toward higher-order skeletons in each window.
> Simply feeding higher-order features into the score network, without modifying the SDE itself, does not change the dynamics in this way, and we are not aware of an analogous error bound for purely conditioned schemes.
>
> Finally, the guidance formulation is modular and flexible. The topological filter that defines $G_{\tau_k}$ is decoupled from the score network design, making it straightforward to swap different higher-order guides across domains without redesigning the conditioning mechanism.
>
> Taken together, these theoretical and empirical advantages motivate the use of higher-order guidance via diffusion bridges rather than relying solely on direct conditioning.

---

> ### Author Response · Authors · 2025-11-21
>
> ```
> Q3: The term “consistently outperforms” in L400 seems little excessive ...
> ```
> We appreciate the reviewer’s feedback and have revised the wording accordingly.  We agree that “consistently outperforms” may sound too strong when considering all five metrics together. Our main improvements in Table 1 lie in NSPDK and FCD, which are widely regarded as the most informative fidelity metrics for molecular graph generation. The substantial reductions in these two metrics indicate that HOG-Diff produces molecules whose distributions are significantly closer to the real data in both chemical space (FCD) and graph structural space (NSPDK). Validity can be further improved through standard post-generation correction procedures without affecting these fidelity metrics.
>
> In addition, we now report Balanced Forman–Ricci Curvature and Ollivier–Ricci Curvature as higher-order topology metrics, as suggested, where HOG-Diff achieves clearly superior performance, further supporting the model’s advantage in preserving structural properties.
>
>     Q4: What exactly do you mean by explicitly preserves higher-order structures in L399 about Fig.3? ...
>
> We thank the reviewer for this helpful suggestion and have improved the visualization accordingly. In the revised manuscript, the **newly added Fig. 3 (right)** provides clearer generative trajectories on QM9, ZINC250k, and MOSES. These visualizations more explicitly illustrate how HOG-Diff maintains topological integrity during generation, addressing the reviewer’s concern.
>
>     Q5: Can you clarify the novelty of CCF?
>
> We thank the reviewer for raising this point and for highlighting connections to TDL methods.
> Classical simplification techniques, such as discrete Morse reductions, aim to construct a smaller cell or simplicial complex that is homotopy equivalent to the original one, preserving all homology groups while reducing the number of cells. These methods typically require constructing gradient vector fields or performing global pairing operations, and are primarily designed to accelerate topological computations.
>
>
> In contrast, CCF is a graph-level, task-driven operator coupled directly to the generative process. Given a graph $G$ and its lifted cell complex $S$, the $p$-cell filter produces filtered graphs $G_{[p]}$ whose vertices and edges are exactly those that participate in some $p$-cell of $S$. This yields a nested family of graphs that act as explicit higher-order skeletons and define the intermediate targets $G_{\tau_k}$ for the diffusion bridges. Unlike discrete Morse reductions, CCF is not intended to preserve all topological invariants; instead, it selectively retains higher-order backbones that guide the generative trajectory while pruning peripheral structures unrelated to any higher-order cell.
>
> Algorithmically, CCF is also significantly lighter. As detailed in Appendix D, it does not enumerate cells, but simply checks whether each node or edge belongs to the closure of some $p$-cell. This local membership test yields negligible preprocessing cost while still providing informative skeletons for guidance.
>
>
> The novelty of CCF therefore lies both in its filtering operation and in integration into HOG-Diff. The filtered graphs serve as endpoints of generalized OU bridges and define a coarse-to-fine generative curriculum.
> This combination of a simple, task-specific filtering operator with a multi-stage diffusion bridge is, to our knowledge, not present in prior frameworks.
>
>
> We sincerely appreciate the reviewer’s constructive suggestions and believe that the additional experiments and analyses have strengthened the quality of our submission.
> We hope that these efforts provide sufficient reasons to raise the score.
>
> [1] J. Southern et al. Curvature Filtrations for Graph Generative Model Evaluation. NeurIPS 2023.

---

> > ### Comment · Reviewer_ftpE · 2025-11-28
> >
> > I have the read the rebuttal and I am okay with the paper. I hope the authors address other reviewers' concerns well.

---

### Official Review · Reviewer_NCCR · 2025-11-01

**Soundness:** 2
**Presentation:** 2
**Contribution:** 2
**Rating:** 4
**Confidence:** 3

**Summary:**

This paper introduces a new diffusion-based graph generative model that integrates higher-order topological information to guide the generation process. The proposed mechanism explicitly leverages cell complex filtering to learn higher-order structures and employs diffusion bridges to enable a coarse-to-fine generation scheme. This design aims to capture richer topological properties and improve the structural coherence of the generated graphs. Experimental results on both molecular and generic graph benchmarks demonstrate performance comparable to recent state-of-the-art graph generative methods.

**Strengths:**

- The use of higher-order topological structures to guide the diffusion process represents a novel contribution. The alignment of the different diffusion stages with hierarchical graph structures seems an innovative way of capturing complex structural patterns.

- The proposed framework is theoretically grounded and proves to be an effective extension of standard diffusion models to non-Euclidean domains.

- Empirical results are competitive with state-of-the-art methods, showing the potential of the proposed approach.

**Weaknesses:**

- The reported performance improvements are relatively modest. Without an accompanying analysis of computational efficiency or scalability, the practical advantages of introducing higher-order topology remain unclear.

- The methodological presentation of the framework could be more intuitive. Although the paper explains how higher-order structures are obtained and integrated into the diffusion process, the overall exposition is difficult to follow, as it often mixes high-level intuition with theoretical descriptions.

- The paper would benefit from a clearer discussion of the potential limitations and possible extensions of the proposed approach, such as scalability, conditional generation for broader applicability across domains, and sensitivity to the choice of higher-order representations. It would be relevant to examine how the model scales with graph size (the SBM dataset, while described in the paper as large-scale, is of relatively modest size) and how conditional generation could be incorporated alongside the higher-order structures.

**Questions:**

- **Hierarchical time windows:** What is the specific role of the number of hierarchical time windows $K$ in the model? If it is treated as a hyperparameter, how is its value determined and how it is related to the number of noising steps? How do changes in $K$ affect the model‚ as generative dynamics?

- **Computational cost:** The model introduces several components that could significantly impact computational efficiency, such as the Cell Complex Filtering operation, the injection of noise into the Laplacian instead of the adjacency matrix (which requires recomputing the Laplacian at each noising step), and the different modules in the score network (GNC and Transformer blocks). Although the authors provide a theoretical argument that the framework converges to the target distribution at an equal or faster rate than classical models, there is no empirical evidence supporting this claim. Having this analysis would be particularly relevant, as the reported performance gains over existing methods are not substantial. Therefore, demonstrating that the added complexity of using higher-order structures translates into practical efficiency or quality improvements (beyond the provided ablation) would strengthen the paper.

### Minor questions

- Why do the authors use different sets of baselines across experiments? Is there a specific reason for this?

- The authors claim that existing graph generative models are ill-suited to capture topological properties because they are largely adapted from image generation frameworks. They further argue that the adjacency matrix quickly degrades into a dense matrix with uniformly distributed entries, which is detrimental to graph structure preservation. However, the paper would benefit from clearer empirical or theoretical evidence supporting these claims. In particular, several recent graph diffusion models employ diffusion processes on discrete state-spaces that are adapted for graph generation and have shown strong performance in maintaining graph topology. As a justification for the need to learn higher-order topology, the paper mentions that the framework should ensure equivariance. However, many existing graph generative models already satisfy equivariance, so this motivation is a bit unclear. In particular, it would be relevant to have proof that the proposed model respects equivariance.

- Could the higher-order information be further exploited to enhance scalability, for example, by incorporating higher-order structure generation within a latent diffusion framework?

---

> ### Author Response · Authors · 2025-11-21
>
> We sincerely thank the reviewer for their thorough and insightful evaluation. We appreciate the recognition that our work is theoretically grounded, achieves competitive results, and offers a novel contribution. Below, we address each point in detail.
>
>     W1: The reported performance improvements are relatively modest. Without an accompanying analysis of computational efficiency or scalability, the practical advantages of introducing higher-order topology remain unclear.
>     Q2: Computational cost
>
> We thank the reviewer for raising these concerns. In the revised manuscript, **Appendix D** now provides a detailed complexity analysis of the cell-complex filtering and guided diffusion steps, and **Appendix F.2** reports a scalability study on large benchmarks.
>
> **(1) Cell Complex Filtering (CCF) is lightweight.**
> The only additional computation introduced by HOG-Diff is a one-time CPU preprocessing step for cell-complex filtering. As detailed in Appendix D, the filter does not enumerate all cells, but simply checks whether a node or edge participates in some $p$-cell. This keeps the lifting cost modest even on large datasets. In the revised manuscript, **Fig. 8 (newly added)** reports times of lifting procedure across datasets of increasing size. Generally, this cost is negligible compared to other data processing steps.
>
> Table A (Fig. 8). Lifting time.
>
> |Dataset|Lifting time|
> |-|-|
> |Ego-small|0.0349 s|
> |Community-small|0.0665 s|
> |Enzymes|0.2983 s|
> |SBM|1.971 s|
> |QM9|22.0843 s|
> |ZINC250k|2 min 44 s|
> |MOSES|15 min 12 s|
>
> **(2) The Laplacian is not recomputed during diffusion.**
> We want to clarify that the Laplacian is constructed only once during the preprocessing phase. Since it is not recomputed at every step, the overhead is negligible and the per-step complexity remains comparable to adjacency-based diffusion models
>
> **(3) The score network has no additional runtime overhead.**
> Our GCN+Transformer architecture follows the same computational profile as GDSS and similar diffusion models. As reported in **Table 5**, the sampling time remains fast, and we observe no measurable overhead attributable to higher-order guidance.
>
> Table B (Table 5). Sampling time(s).
>
> |Method|Community-small|Enzymes|Ego-small|
> |-|--|-|-|
> |GDSS|41|110|28|
> |DiGress|8|301|13|
> |HOG-Diff|11|26|15|
>
> **(4) Empirical evidence of improved structural fidelity.**
> Beyond classical metrics, we introduce curvature-based evaluations [1] (Balanced Forman-Ricci and Ollivier-Ricci) in **Section 4.3**. **Table 3 (newly added)** shows that HOG-Diff substantially better preserves higher-order topological features than state-of-the-art baselines, directly addressing the question of whether higher-order topology provides practical advantages.
>
> Table C (Table 3). Quantitative evaluation of higher-order topology preservation using Curvature Filtrations.
>
> |Dataset|Method|Forman|Ollivier|
> |-|-|-|-|
> |QM9|GDSS|0.925|0.601|
> |QM9|DiGress|0.251|0.343|
> |QM9|DoFoG|0.177|0.286|
> |QM9|Cometh|0.216|0.314|
> |QM9|HOG-Diff|0.077|0.098|
> |ZINC250k|MiCaM|9.436|7.251|
> |ZINC250k|GDSS|1.781|1.331|
> |ZINC250k|DoFoG|0.728|0.498|
> |ZINC250k|HOG-Diff|0.190|0.098|
> |SBM|DoFoG|3.998|5.206|
> |SBM|Cometh|2.748|3.938|
> |SBM|HOG-Diff|1.453|3.740|
> |Community-small|GDSS|2.311|1.301|
> |Community-small|HOG-Diff|1.679|1.042|
> |Enzymes|GDSS|13.031|12.399|
> |Enzymes|HOG-Diff|12.114|9.830|
> |Ego-small|GDSS|2.311|1.301|
> |Ego-small|HOG-Diff|1.679|1.042|
>
> **(5) Scalability to large datasets.**
> We further added experiments on the MOSES dataset of over 1.9M molecules (see the **newly added Table 8**), demonstrating that HOG-Diff trains and samples efficiently at scale.
>
> Table D (Table 8). Generation results on the MOSES dataset
>
> |Method|Val.↑|Unique.↑|Nov.↑|Filters↑|FCD↓|SNN↑|Scaf↑|
> |--|-|-|-|-|-|-|-|
> |GraphINVENT|96.4|99.8|-|95.0|1.22|0.54|12.7|
> |DiGress|85.7|100.0|95.0|97.1|1.19|0.52|14.8|
> |DisCo|88.3|100.0|97.7|95.6|1.44|0.50|15.1|
> |Cometh|90.5|99.9|92.6|99.1|1.27|0.54|16.0|
> |HOG-Diff|97.6|100.0|92.7|93.1|1.09|0.51|10.0|
>
> **(6) Qualitative evidence of stable generative dynamics.**
> The radar plots in the **newly added Fig. 1** highlight improvements across both classical and higher-order metrics. The updated generative trajectory visualizations in **Fig. 3 (right)** show that HOG-Diff maintains topological integrity throughout intermediate steps, supporting the theoretical claim of more stable and structured diffusion dynamics.
>
>
> Together with new experiments, these results show that our framework introduces negligible overhead while providing practical benefits in both sample quality and structural fidelity.
>
> [1] J. Southern et al. Curvature Filtrations for Graph Generative Model Evaluation. NeurIPS 2023.

---

> ### Author Response · Authors · 2025-11-21
>
> ```
> W2: representation.
> ```
> We appreciate the reviewer’s feedback. We have made significant efforts to enhance the overall accessibility and flow of the framework presentation. From a top‑level perspective, the process now reads as follows:
>
> 1. Coarse-to-fine schedule
>
> The graph is first "lifted" to a cell complex, and a cell-complex filtering (CCF) operator extracts
> progressively finer skeletons. Notably, CCF is a preprocessing operation. This produces a sequence $G_{\tau_K}{\to}G_{\tau_{K-1}}\to\ldots{\to}G_{\tau_0}=G$ that acts as the curriculum for generation. Each adjacent pair forms one time window $[ \tau _{k- 1}, \tau _k]$.
>
> 2. Forward diffusion in each window
>
> Within $[\tau_{k-1},\tau_k]$, we run a generalized Ornstein-Uhlenbeck (GOU) bridge that gradually perturbs $G_{\tau_{k-1}}$ toward $G_{\tau_k}$, while preserving the higher-order structure present at the two endpoints.
>
> 3. Reverse denoising across windows.
>
> At sampling time, we iterate backwards through the windows. Starting from pure Gaussian noise, the score network first reconstructs the coarsest skeleton $G_{\tau_{K-1}}$, and then refines it window-by-window until the full graph $G_{0}$ is obtained. In each window, GOU bridge realizes the structured diffusion process, ensuring smooth transitions between intermediate states in adjacent hierarchical levels and ultimately converges to a noise-free graph.
>
>     W3: The paper would benefit from a clearer discussion of the potential limitations and possible extensions of the proposed approach...
>
> Regarding scalability, we now provide more direct evidence in the revised manuscript.
> Appendix F.2 (*Scalability evaluation on large benchmarks*) reports experiments on the full MOSES dataset with over 1.9 million molecules. In addition, Appendix D now includes **Fig. 8**, which reports the wall clock time of the lifting (preprocessing) step across datasets of different sizes, from small generic graphs to large-scale benchmarks such as ZINC250k and MOSES. This one-time cost remains tractable even at scale. Together with the detailed complexity analysis in Appendix D, it shows that the extra preprocessing from cell complex filtering and lifting is modest, while the overall training and sampling pipeline remains comparable to standard diffusion models.
>
> We have also expanded our discussion of limitations and future work. In addition to the original discussion of limitations in **Appendix G**, the conclusion section now explicitly states that the performance of HOG-Diff depends on the presence of explicit higher-order structures, which certain graph families may lack. At the same time, higher-order motifs are common in many realistic networks (molecular, biological, social), where our approach is most beneficial. We further outline concrete extensions: exploring alternative filtering mechanisms beyond CCF and adaptively choosing the order (dimension) of the higher-order skeleton to better match the underlying data.
>
> We hope this expanded discussion clarifies both the strengths and the limitations of our approach.
>
>     Q1: Hierarchical time windows.
> We thank the reviewer for this valuable question.
> Conceptually, $K$ controls the granularity of the coarse-to-fine curriculum: we first construct a sequence of higher-order skeletons via cell-complex filtering, and then partition the time horizon $[0,T]$ into K windows $[\tau_{k-1},\tau_k]$. Each window is modeled by a generalized OU bridge that transports the graph from $G_{\tau_k}$ to $G_{\tau_{k-1}}$, so increasing $K$ introduces more intermediate, topologically meaningful “waypoints” along the generative trajectory.
>
> In practice, $K=2$ is fixed across datasets. On the standard generative benchmarks we consider, cells of dimension $>2$ are extremely sparse (see **Table 9**), and constructing higher-dimensional cell complexes incurs nontrivial preprocessing overhead. Empirically, we observe that $K=1$ collapses the curriculum and behaves similarly to a classical single-stage diffusion model, whereas $K=2$ already captures the vast majority of meaningful higher-order structure, provides sufficient guidance, and yields stable generative dynamics: trajectories first snap to a coarse skeleton and then refine local details.
> Increasing $K$ beyond 2 tends to fragment the guidance without adding substantial structural information, so the signal per window diminishes. Our framework is, however, agnostic to the choice of $K$, and for genuinely large-scale, topologically rich graphs, exploring larger $K$ is a promising direction that we plan to investigate in future work.
>
> Table E (Table 9). Average counts of higher-order structures across datasets. “–” indicates computation exceeded 10h.
>
> |Dataset|2-simplices|3-simplices|4-simplices|2-cells|3-cells|
> |-|-|-|-|-|-|
> |Community-small|27.45|8.24|0.76|40.92|0.00|
> |Ego-small|4.61|1.64|0.35|5.66|0.00|
> |Enzymes|25.93|3.05|0.01|-|-|
> |SBM|441.25|90.88|4.42|-|-|
> |QM9|0.47|0.00|0.00|1.84|0.10|
> |ZINC250k|0.06|0.00|0.00|2.77|0.00|

---

> ### Author Response · Authors · 2025-11-21
>
> ```
> Minor Q1.
> ```
> We thank the reviewer for this question.
> Our implementation builds on the GDSS framework, so we initially adopted the same set of baselines for consistency and comparability. As we expanded the evaluation, we incorporated additional recent baselines. However, different baselines report results on different subsets of datasets, and several works do not provide pretrained checkpoints or official configurations for all benchmarks.
>
> As a result, the set of baselines varies across experiments. For each dataset, we include methods whose results are either publicly reported under a comparable setting or reproducible using the authors’ released code.
> This allows us to keep each comparison as fair, faithful to the original implementations, and as comprehensive as possible, while avoiding unsupported or incomparable configurations.
>
>     Minor Q2.
>
> We thank the reviewer for this important comment.
> First, our critique is now explicitly restricted to early continuous adjacency-space diffusion models with isotropic Gaussian noise. We position HOG-Diff as complementary to recent discrete and flow-matching models.
>
> Second, we now provide concrete evidence for topological degradation and its mitigation. The revised **Fig. 3 (right)** visualizes generative trajectories on QM9, ZINC250k, and MOSES: for classical adjacency-based and discrete baselines, intermediate graphs quickly become dense and lose higher-order topology like cycles, whereas HOG-Diff maintains them throughout the trajectory. Section 4.3 further quantifies this via curvature filtrations: **Table 3** and **Fig. 1 (right)** show that HOG-Diff attains the smallest Balanced Forman–Ricci and Ollivier–Ricci curvature distances to the testing graphs, indicating better preservation of higher-order organization.
>
> Finally, we clarify that our score network is permutation equivariant, as every component avoids node ordering-dependent operations. We have therefore toned down the equivariance-related motivation and now emphasize that our main contribution is explicit higher-order guidance that shapes the diffusion dynamics and improves topological fidelity.
>
>     Minor Q3.
>
> We thank the reviewer for this insightful suggestion.
> Incorporating higher-order structure generation within a latent diffusion framework is beyond the scope of the current work, but we fully agree that it is a very promising direction to further improve scalability, especially for very large graphs or richer geometric settings.
>
> Conceptually, one could first learn a latent space that explicitly encodes full higher-order topology (not only skeletons), and then perform diffusion in this latent space instead of in the original graph or spectral domain.
> In our ongoing work, we have been exploring such a construction: higher-order structures are embedded into a latent representation that can be mapped back to the original higher-order structures in a bijective manner.
> This is non-trivial, as higher-order structures of different dimensions are strongly coupled, which makes standard embedding schemes hard to invert. The resulting latent space natively reflects the higher-order topology, and preliminary experiments indicate promising scalability and generative performance.
>
>
> Once again, we sincerely thank the reviewer for their thoughtful engagement. We believe that we have addressed all the concerns of the reviewer and kindly ask them to reconsider their rating.

---

> > ### Comment · Reviewer_NCCR · 2025-11-25
> > **Response to authors**
> >
> > Many thanks to the authors for their thorough response. The clarifications are helpful and address some of the concerns, particularly regarding the presentation of the framework and the expanded discussion of limitations. However, a few concerns still remain:
> >
> > - **Scalability**: The inclusion of a dataset with a larger number of samples (MOSES) is appreciated, but the original concern referred specifically to the graph size, not the dataset size. It would be valuable to examine how higher-order topologies (particularly in relation with larger values of $K$, see comment below) affect training on larger graphs.
> > - **Hierarchical time windows**: The rationale behind the choice of time windows is now clearer, and the additional explanation is helpful. Still, it would be useful to evaluate how varying $K$ impacts performance, as well as how training time changes for larger values of $K$ (in line with the other two comments). For example, in the SBM dataset, where the number of 3-simplices and 4-simplices is substantial and the number of cells is not reported, it would be interesting to already explore larger values of $K$, since this only requires adjusting a hyperparameter.
> > - **Computational time**: the tables including lifting and sampling times are highly appreciated, and the explanation that the score network has no additional runtime overhead is understood. However, I remain curious about the overall computational cost, particularly the training time of the full method. Since the framework includes both a GCN and a Graph Transformer Network, it would be relevant to report and compare the complete training time against other methods (or at least against GDSS and DiGress).

---

> > > ### Author Response · Authors · 2025-11-26
> > >
> > > We sincerely thank the reviewer for their careful review and the insightful follow-up comments.
> > >
> > >     1. Scalability
> > >
> > > We would like to bring a nuanced clarification to the scalability in higher-order learning. It is true that the theoretical (worst-case) complexity of considering 'all' higher order cells is exponential in the number of nodes, **the nature does not scale like that**. Oftentimes, the number of higher-order simplices/ cells/ voids is typically far less compared to the size of the rest of the network (see **Table F**).
> > > For example, the average counts of 0-simplices, 1-simplices, and 2-simplices in the Enzymes dataset are 33.03, 63.27, and 25.93, respectively; in the SBM dataset, these values are 104.01, 500.41, and 441.25, respectively. Hence, neither our algorithm nor any other higher-order network suffers from this worst-case complexity in practice. Note that in all experiments we limit ourselves to $K=2$ and the added complexity is negligible.
> > >
> > > Table F (Table 10). Average counts of simplices and cells across datasets. ``-'' indicates computation exceeded 10h.
> > >
> > > |Dataset|0-simplices|1-simplices|2-simplices|3-simplices|4-simplices|2-cells|3-cells|
> > > |-|-|-|-|-|-|-|-|
> > > |Community-small|15.28|35.15|27.45|8.24|0.76|40.92|0.00|
> > > |Ego-small|6.41|8.70|4.61|1.64|0.35|5.66|0.00|
> > > |Enzymes|33.03|63.27|25.93|3.05|0.01|-|-|
> > > |SBM|104.01|500.41|441.25|90.88|4.42|-|-|
> > > |QM9|8.80|9.40|0.47|0.00|0.00|1.84|0.10|
> > > |ZINC250k|23.15|24.90|0.06|0.00|0.00|2.77|0.00|
> > >
> > >
> > >     2. Hierarchical time windows
> > >
> > > We thank the reviewer for this suggestion. We acknowledge that the dimension of the complex is an important (hyper-)parameter, ultimately influencing 'which' higher-order information is included as the guide. Too small, and no information is present; too large, the training gets complicated with diminishing returns. This is the reason we find $K=2$ to strike a balance across all datasets. As shown in **Table F**, the numbers of 3-simplices and 4-simplices are far smaller than the number of 2-simplices, even in datasets such as SBM. Taking the reviewer's suggestion into account, we have performed additional experiments with larger values of $K$, and as anticipated, the performance slightly degraded.
> > > This confirms that $K=2$ captures most of the meaningful higher-order structure in current datasets, thus offering sufficiently strong generative guidance while yielding stable dynamics in the coarse-to-fine diffusion process.
> > >
> > >     3. Computational time
> > >
> > > We sincerely thank the reviewer for this constructive suggestion. In response, we have retrained both our model and the GDSS using the provided hyperparameters. As shown in **Table G** (corresponding to the **newly added Table 6** in the revised manuscript), our method achieves comparable overall training time to GDSS across most datasets.
> > > Notably, on the more challenging and larger-scale ZINC250k dataset, our method achieves a substantial improvement, reducing the training time from 14 h to 6.5 h, an efficiency gain of more than twofold.
> > > For QM9, the training time is slightly longer. This is because the runtime on this dataset is already well within a reasonable range, and we did not apply any specific optimization for training efficiency on any of the datasets.
> > >
> > > Table G (Table 6). Training time comparison between GDSS and HOG-Diff.
> > >
> > > |Method|Community-small|Ego-small|Enzymes|QM9|ZINC250k|
> > > |-|-|-|-|-|-|
> > > |GDSS|13min|19min|1.9h|1.2h|14.0h|
> > > |HOG-Diff|20min|12min|0.5h|3.1h|6.5h|
> > >
> > > Once again, we sincerely thank the reviewer for their thoughtful engagement. We hope that our additional clarifications satisfactorily address all of the concerns.

---

### Official Review · Reviewer_zqT6 · 2025-11-01

**Soundness:** 2
**Presentation:** 2
**Contribution:** 2
**Rating:** 2
**Confidence:** 4

**Summary:**

This paper introduces HOG-Diff, a novel diffusion framework for graph generation. The core methodology involves "lifting" graphs into a higher-order cell complex representation and make the diffusion process respect higher-order topological structures. The authors posit that this higher-order diffusion better captures the intrinsic topological structure of graphs. The paper provides a theoretical analysis of the proposed methodology and presents experimental results to demonstrate its performance.

**Strengths:**

- The idea of graph generation via lifted is refreshing.
- The paper is built upon a solid mathematical formulation.
- The theoretical foundations, including theorems and propositions, are well-presented, and the accompanying proofs provided in the appendix appear to be correct.

**Weaknesses:**

Despite its theoretical grounding, the paper suffers from significant weaknesses in its justification, clarity, and empirical evaluation.

- **Clarity of Writing and Notation.** A primary concern is the clarity of the writing and the logical flow. The notation, in particular, seems inconsistent and potentially overloaded. For instance, Proposition 2 introduces the $p$-cell complex filtered graph $G_p$, where the subscript $p$ denotes the cell dimension. However, in subsequent sections, graphs are denoted as $G_t$, where $t$ represents time. It is unclear if $G_t$ retains a connection to a specific $p$-cell filtration or if this notation is dropped. This ambiguity makes the methodology difficult to follow, thus hinders a precise understanding of the manuscript’s main contribution and the relevant algorithm design.
- **Justification of Core Motivation and Theoretical Bound.** The central motivation of the paper is that lifting graphs to a higher-order space improves generative fidelity, which I believe requires stronger justification. While higher-order representations are beneficial for systems with inherent group-level interactions (e.g., co-authorship networks), this does not mean converting graphs to their high-order counterpart would be beneficial for downstream tasks. As shown in [1], a hypergraph and its corresponding graph clique expansion can be information-theoretically equivalent in certain contexts. The paper does not sufficiently argue why this lifting operation provides a superior inductive bias for learning the data distribution. This concern extends to the primary theoretical result (Theorem 4). The analysis demonstrates that the final step of the HOG-Diff curriculum (from $\tau_1$ to $0$) achieves a sharper error bound than a classical model's entire process (from $T$ to $0$). This comparison does not seem equitable. The proof relies on $\tau_1 < T$ and a superior (coarsened) starting point which induces $\mathcal{E}(\tau_1) \le \mathcal{E}^{\prime}(T)$. A more convincing analysis would compare the entire $K$-step generative process of HOG-Diff (from $T \to 0$) against the classical $T \to 0$ process, rather than isolating the final, and arguably simpler, refinement step. I might have misunderstanding on the theoretical bound and would appreciate if the authors can correct me.
- **Critique of Diffusion Model Baselines.** The paper's motivation appears to be based on a critique of graph diffusion models. The claim that diffusion models are "ineffective at modeling the topological properties" (Lines 70-72) overlooks recent successes in discrete-state diffusion. Similarly, the criticism that "the graph adjacency matrix quickly degrades into a dense matrix" (Line 72) and that injecting "isotropic Gaussian noise...is detrimental" (Line 76) primarily applies to early continuous diffusion models. This critique does not hold for more recent discrete frameworks such as DiGress [3], Cometh [2], and DeFoG [4], which deviate from the Gaussian noise paradigm. The paper should position its contribution against these more relevant state-of-the-art discrete models, as the problems it claims to solve have, to a large extent, already been addressed.
- **Significance and Fairness of Empirical Evaluation.** The empirical evaluation lacks significance due to the omission of standard benchmarks for graph generation. Datasets such as SBM, Planar, and Tree graphs are critical as they possess ground-truth topological properties (e.g., planarity, acyclicity). Evaluating validity on these datasets is fundamental to assessing a model's ability to capture intrinsic graph patterns. The paper's reliance on statistics ratio (degree, clustering) and molecular validity (which is basically valency validity) does not sufficiently demonstrate that the model can learn and respect these fine-grained topological constraints, especially in plain graph generation tasks. Furthermore, the comparison to existing work is incomplete. Key SOTA diffusion models, such as Cometh [2], are not included. In the molecular benchmarks (QM9, ZINC250k), the reported improvement over DeFoG [4] appears marginal, but a proper comparison is impossible as standard deviations are not provided.
- **Scalability and Computational Complexity.** Scalability remains an unanswered question. All experiments are conducted on relatively small-scale graphs. Standard, larger benchmarks (e.g., MOSES, GUACAMOL for molecules; SBMs, the larger version of COMM-20) are absent. This is critical as the core premise of HOG-Diff relies on computing higher-order structures. It is unclear (1) whether computing these structures is computationally feasible for large graphs, and (2) how the performance and complexity of HOG-Diff scale relative to standard graph generation models on such data. Although Appendix D discussed computational complexity, it would be good to see empirical validation as well.
- **Clarity of Algorithmic Details.** The paper would benefit from a more precise definition of its algorithmic settings in the main text. Key details, such as the exact dimensions and representations of $G$ and $G_p$, are not sufficiently clear. This lack of detail hinders a rigorous analysis of the algorithm's space and time complexity.

References

[1] B. Tang et al. Training-free message passing for learning on hypergraphs. ICLR 2025

[2] A. Siraudin et al. Cometh: A continuous-time discrete-state graph diffusion model. TMLR 2025

[3] C. Vignac et al. DiGress: Discrete Denoising diffusion for graph generation. ICLR 2023

[4] Y. Qin et al. DeFoG: Discrete Flow Matching for Graph Generation. ICML 2025.

**Questions:**

Most of my questions are raised in weakness. This includes the problem of writing, the main motivation of the paper, the significance of the empirical studies and the scalability. Please refer to that section.

---

> ### Author Response · Authors · 2025-11-21
>
> We sincerely thank the reviewer for their thorough and insightful review. Below, we address each point in detail.
>
>     W1: Clarity of Writing and Notation.
>
> We sincerely appreciate the reviewer’s careful reading. The $p$-cell complex filtered graph previously denoted by $G_p$ has been renamed to $G_{[p]}$ to avoid overloading the subscript with both “cell dimension” and “time”. In the revised text, $G_{[p]}$ is used only in the context of cell-complex filtering, while $G_t$ is reserved exclusively for time-indexed graph states along the diffusion trajectory. We now explicitly state that the graphs at the boundaries of the hierarchical windows, $G_{\tau_k}$, are instantiated from the corresponding filtered graphs $G_{[p]}$ produced in Proposition 2, making the connection between filtration and time evolution precise.
>
> We have also carefully gone through the entire manuscript to harmonize the notation, remove overloaded symbols, and ensure that each index plays a single, clearly defined role.
>
>     W2: Justification of (1) Core Motivation and (2) Theoretical Bound
>
> We thank the reviewer for raising these important points.
>
> (1) The key idea of HOG-Diff is not to replace graphs with an alternative higher-order representation, but to use higher-order topology as an explicit source of generative guidance. Lifting produces a coarse-to-fine sequence of topological skeletons that anchor the diffusion trajectory and keep intermediate states structurally meaningful, instead of collapsing into near-random adjacency matrices as in classical diffusion (**newly added Fig. 3, right**). This is fundamentally different from representational equivalence results such as hypergraph–clique expansion: in our setting, higher-order structures shape the *dynamics* of the generative process, not the expressiveness of the underlying graph model.
>
>
> Higher-order structures are also ubiquitous and informative in real data. Empirical studies show that social, biological, and citation networks, as well as many modern drug-like molecules, contain abundant cells and simplices that capture multi-way interactions beyond pairwise edges. For example, higher-order cells are widespread in molecular graphs and encode functional groups that influence 3D conformation, electron distribution, and binding affinity; similarly, 3-simplices often mark community cores in social networks. A generator that ignores such patterns can be as expressive in principle yet still fail to reproduce the statistics of real-world graphs.
> Our coarse-to-fine guidance further simplifies learning.
> HOG-Diff decomposes graph generation into manageable sub-tasks: it first generates higher-order graph skeletons that capture core structure, and then refines them with pairwise edges and local details. Each stage therefore solves a simpler problem.
> Theorems 3 and 4 formalize this benefit by showing that higher-order guided bridges enjoy a smaller smoothness constant and a tighter reconstruction-error bound than classical diffusion, precisely because the trajectory is constrained to pass through informative topological states.
>
> Empirically, this inductive bias is reflected in large improvements in NSPDK and FCD on molecular datasets, where 2-cells align with chemically meaningful rings and scaffolds.
> In Section 4.4, ablations that replace higher-order guides with random noise or peripheral structures consistently degrade generation quality, confirming that the gains stem from higher-order guidance rather than incidental architectural changes.
> In Section 4.3, we further introduce curvature-based evaluations using Balanced Forman-Ricci and Ollivier-Ricci curvature.
> As shown in the **newly added Fig. 1 (right)** and **Table 3**, HOG-Diff attains the smallest curvature distances across datasets, indicating superior preservation of higher-order geometric and topological properties.
> Furthermore, **Fig. 3 (right)** also demonstrates that HOG-Diff maintains cycles throughout intermediate steps, a behavior that classical diffusion fails to exhibit.
>
> In summary, the benefit of lifting arises not from representational changes but from providing principled, topologically grounded guidance that structures the diffusion trajectory, improves optimization, and preserves higher-order dependencies that would otherwise be lost.
>
> Table A (Table 3). Quantitative evaluation of higher-order topology preservation using Curvature Filtrations.
>
> |Dataset|Method|Forman|Ollivier|
> |-|-|-|-|
> |QM9|GDSS|0.925|0.601|
> |QM9|DiGress|0.251|0.343|
> |QM9|DoFoG|0.177|0.286|
> |QM9|Cometh|0.216|0.314|
> |QM9|HOG-Diff|0.077|0.098|
> |ZINC250k|MiCaM|9.436|7.251|
> |ZINC250k|GDSS|1.781|1.331|
> |ZINC250k|DoFoG|0.728|0.498|
> |ZINC250k|HOG-Diff|0.190|0.098|
> |SBM|DoFoG|3.998|5.206|
> |SBM|Cometh|2.748|3.938|
> |SBM|HOG-Diff|1.453|3.740|
> |Community-small|GDSS|2.311|1.301|
> |Community-small|HOG-Diff|1.679|1.042|
> |Enzymes|GDSS|13.031|12.399|
> |Enzymes|HOG-Diff|12.114|9.830|
> |Ego-small|GDSS|2.311|1.301|
> |Ego-small|HOG-Diff|1.679|1.042|

---

> ### Author Response · Authors · 2025-11-21
>
> (2) **Theoretical Bound.**
> The reviewer is correct that the earlier version of the proof relied on an assumption that simplified the analysis by focusing only on the final refinement stage.
> In the revised manuscript, we have removed this simplifying assumption and updated the theoretical analysis accordingly. The new version establishes an error bound for the entire $K$-stage HOG-Diff generative trajectory from $T$ to $0$, rather than comparing only the last stage with the full classical process. This provides a fair and fully aligned comparison between the two methods.
>
> We appreciate the reviewer’s careful reading and constructive suggestion, which helped us significantly strengthen the theoretical justification.
>
>     W3: Critique of Diffusion Model Baselines.
>
> We thank the reviewer for this valuable comment.
> We want to clarify that our goal is not to dismiss recent discrete diffusion models, but to introduce higher-order topology as an explicit generative guidance signal that is complementary to both continuous and discrete frameworks.
> We have revised the Introduction and Related Work sections to clarify that our initial critique primarily targets early continuous diffusion models operating directly in adjacency space and explicitly acknowledged that recent discrete-state and flow-matching methods (e.g., DiGress, Cometh, DeFoG) represent significant progress beyond the original Gaussian noise paradigm.
>
> Our contribution targets a different aspect: existing discrete models typically evolve node and edge states in a purely pairwise space, where higher-order topology remains an implicit by-product of edge-level denoising. In contrast, HOG-Diff employs lifting and cell-complex filtering to construct a sequence of coarse-to-fine higher-order skeletons that act as *explicit intermediate targets* for diffusion bridges.
> This fundamentally alters the generative dynamics by constraining the trajectory to pass through structurally meaningful higher-order states instead of arbitrary noisy graphs.
>
>
> Furthermore, we provide both theoretical and empirical evidence that this explicit topological guidance offers benefits beyond baselines.
> Theorems 3 and 4 show that higher-order guided bridges yield a smaller smoothness constant and a tighter reconstruction-error bound than classical diffusion, as the trajectory is anchored by informative topological states.
> Empirically, HOG-Diff not only matches or exceeds strong discrete baselines on classical metrics, but also achieves markedly better preservation of higher-order topology, as quantified by curvature filtrations (Balanced Forman–Ricci and Ollivier–Ricci curvature in the **newly added Tab.3** and **Fig. 1**) and by the generative trajectories (**Fig. 3, right**).
>
> Taken together, these results position HOG-Diff as a topology-aware framework that builds upon and complements recent discrete diffusion models by injecting explicit higher-order guidance into the generative dynamics, rather than competing solely at the level of pairwise representations.
>
>     W4-1: Significance and Fairness of Empirical Evaluation. The empirical evaluation lacks significance due to the omission of standard benchmarks for graph generation. Datasets such as SBM, Planar, and Tree graphs are critical as they possess ground-truth topological properties (e.g., planarity, acyclicity). Evaluating validity on these datasets is fundamental to assessing a model’s ability to capture intrinsic graph patterns.
>
> We thank the reviewer for raising this concern.
> We would like to clarify that experiments on the SBM dataset were already included in the original submission. In the revised manuscript, we have moved these results into the main text (see **Fig. 3, left**) to improve their visibility. The generic graph benchmarks we use (Community-small, Ego-small, Enzymes, SBM) follow the standard setup adopted by prior graph generation work such as GDSS [5] and GSDM [6]. We intentionally align with these settings so that our results are directly comparable to widely used baselines.
>
> In response to the reviewer’s concern, we have additionally incorporated large-scale experiments on the MOSES benchmark (1.9M molecules) in the revised manuscript. The **new results in Table 9** show that HOG-Diff scales to this regime and remains competitive or superior to recent state-of-the-art baselines.
>
> Overall, the SBM and MOSES results provide a more complete assessment of HOG-Diff on datasets with known or strongly structured topology. We hope these updates offer a clearer and more comprehensive comparison.
>
> Table B (Table 9). Generation results on the MOSES dataset
>
> |Method|Val.↑|Unique.↑|Nov.↑|Filters↑|FCD↓|SNN↑|Scaf↑|
> |--|-|-|-|-|-|-|-|
> |GraphINVENT|96.4|99.8|-|95.0|1.22|0.54|12.7|
> |DiGress|85.7|100.0|95.0|97.1|1.19|0.52|14.8|
> |DisCo|88.3|100.0|97.7|95.6|1.44|0.50|15.1|
> |Cometh|90.5|99.9|92.6|99.1|1.27|0.54|16.0|
> |HOG-Diff|97.6|100.0|92.7|93.1|1.09|0.51|10.0|

---

> ### Author Response · Authors · 2025-11-21
>
> ```
> W4-2: The paper’s reliance on statistics ratio (degree, clustering) and molecular validity (which is basically valency validity) does not sufficiently demonstrate that the model can learn and respect these fine-grained topological constraints, especially in plain graph generation tasks.
> ```
> In addition to the classical statistics-based metrics, we now include higher-order topological evaluations using Balanced Forman–Ricci curvature and Ollivier–Ricci curvature as filtration functions, and measure the distance between the curvature distributions of generated graphs and those of the testing data. These quantities are sensitive to higher-order organization and geometric structure beyond local degree or clustering. As shown in the **newly added Table 3** and **Fig. 1 (right)**, HOG-Diff consistently attains the lowest curvature distances across both molecular and generic graph benchmarks.
>
>     W4-3: Furthermore, the comparison to existing work is incomplete. Key SOTA diffusion models, such as Cometh [2], are not included. In the molecular benchmarks (QM9, ZINC250k), the reported improvement over DeFoG [4] appears marginal, but a proper comparison is impossible as standard deviations are not provided.
>
> We thank the reviewer for this helpful suggestion. In the revised manuscript, we have added comparisons with Cometh [2] on the molecular generation benchmarks (QM9 and MOSES). As shown in **Table 1** and **Fig. 1 (middle and right)**, HOG-Diff outperforms baselines not only on classical pairwise metrics but also—more substantially—on these higher-order topological measures.
>
> Regarding the concern about standard deviations, we would like to clarify that they were already included in the submission. The standard deviations for molecular generation benchmarks are reported in **Tables 11** and **12** in Appendix F. We have updated the manuscript to make this more visible and easier to locate.
>
>     W5: Scalability and Computational Complexity.
> We thank the reviewer for highlighting this important point and have strengthened both the theoretical and empirical analysis of scalability in Appendix D in the revised manuscript.
>
> Specifically, the only additional computation introduced by HOG-Diff is a one-time CPU preprocessing step for cell-complex filtering. The filtering operator does not enumerate all higher-order cells; instead, it performs local membership checks to determine whether a node or edge participates in some $p$-cell. This design keeps the preprocessing complexity modest, even on large graphs. To empirically validate the feasibility of computing higher-order structures on large datasets, we include a new study of the lifting runtime across datasets of increasing size (see the **newly added Fig. 8** in Appendix D). On the full MOSES benchmark with $1.9$M molecules, the lifting operation completes around 15 mins. While higher than on smaller datasets, this cost remains a one-time preprocessing step and is negligible relative to other data processing procedures.
>
> Once the higher-order skeletons are constructed, both training and sampling use the same loop as standard diffusion models. In practice, we observe no measurable slowdown in training relative to GDSS or DiGress. Appendix F.2 reports sampling times, showing that HOG-Diff matches discrete baselines on smaller graphs and can even be faster on larger protein-scale graphs.
>
> To further evaluate scalability in terms of model performance on large graphs, we have included experiments on larger-graph benchmarks, such as the SBM dataset (**Fig. 3, left**) and the MOSES dataset (**Table 9**). These experiments show that HOG-Diff can be trained and evaluated on large-scale datasets without introducing a computational bottleneck, indicating that both the filtering procedure and the guided bridge steps scale effectively in practice.
>
> Table C (Fig. 8). Lifting time.
>
> |Dataset|Lifting time|
> |-|-|
> |Ego-small|0.0349 s|
> |Community-small|0.0665 s|
> |Enzymes|0.2983 s|
> |SBM|1.971 s|
> |QM9|22.0843 s|
> |ZINC250k|2 min 44 s|
> |MOSES|15 min 12 s|
>
> Table D (Table 5). Sampling time(s).
>
> |Method|Community-small|Enzymes|Ego-small|
> |-|--|-|-|
> |GDSS|41|110|28|
> |DiGress|8|301|13|
> |HOG-Diff|11|26|15|
>
>     W6: Clarity of Algorithmic Details.
> We confirm that all symbols have been carefully corrected and harmonized throughout the manuscript to ensure rigorous consistency. We have also addressed concerns about complexity and scalability in detail in our previous response.
>
> Once again, we sincerely thank the reviewer for their thoughtful engagement. We have fully revised our introduction to clarify the abovementioned motivation. We strongly believe that we have addressed all the comments of the reviewer and kindly urge them to reconsider their rating under this new evidence and exposition.
>
> [5] J. Jo et al. Score-based Generative Modeling of Graphs via the System of Stochastic Differential Equations. ICML 2022.
>
> [6] T. Luo et al. Fast Graph Generation via Spectral Diffusion. TPAMI 2024.

---

> ### Comment · Reviewer_zqT6 · 2025-11-27
>
> Thank you for your detailed clarifications on the HOG-Diff pipeline, notations, and methodology. Your rebuttal has generally addressed my initial concerns regarding W1, W2, and W6. One remaining question: why it is realistic to assume that coarse-to-fine pipeline will make the score smoother, and consequently make the approximation loss lower? That is, the assumption behind Eq 59.
>
> My concern about scalability remains. I second with reviewer NCCR that scaling up to larger datasets (in terms of number of graphs) is one aspect and scaling with the graph size (#nodes) is another aspect. I appreciate the MOSES results, which successfully illustrate HOG-diff’s ability in processing large dataset. However, as HOG-diff’s requirement to rely on high-order topology, experiments on datasets like GUACAMOL are necessary to fully validate HOG-Diff's scalability with respect to #nodes.
>
> My concern related to the empirical evaluation persists. The suggested comparison in Planar, SBM, and tree graphs, are mainly because they correctly test the model’s ability in preserving specific graph structure, such as planarity, tree structure, and community structure, which is reflected in their “Validity” score. This is what I meant by saying “Evaluating validity on these datasets is fundamental…” I believe validity and ratios are two important dimensions in evaluating graph generation abilities. While the model might exhibit strong ability in mimicking the statistical ratios, but still fails in capturing the validity.
>
> For MOSES dataset, as I do not see them in the source code, could you kindly let me know if you are based in `test` split or the  `test_scaffolds` split? The latter is commonly used in DiGress, DeFoG, etc., and for the former I believe the benchmarks are relatively high [1] (not asking for a comparison with this concurrent work but rather point that the split can have a significant impact).
>
> [1] O. Petersen, “Discrete Bayesian Sample Inference for Graph Generation,” arXiv. 2511.03015.

---

> ### Author Response · Authors · 2025-12-03
>
> We sincerely thank the reviewer for their careful review and the insightful follow-up comments.
>
>     One remaining question: why it is realistic to assume that coarse-to-fine pipeline will make the score smoother, and consequently make the approximation loss lower? That is, the assumption behind Eq 59.
>
> We are happy to clarify Eq. (59), which is now Eq. (65) in the revised manuscript. We have revised the manuscript to make this point more explicit by adding **Lemma 6** and referencing it in the proof of Theorem 4.
>
> At the level of optimal score-matching losses, Lemma 6 shows that the coarse-to-fine HOG-Diff architecture cannot be intrinsically worse than the single-stage classical model.
> Intuitively, each HOG-Diff stage only needs to approximate a smoother conditional score (given the coarse graph), whereas the classical model must fit the full marginal score in a single stage. For a fixed network capacity, decomposing the task into these easier conditional subproblems is therefore not expected to deteriorate the overall score-matching accuracy. In a regime where the optimization error is negligible and the observed loss primarily reflects approximation error, it is thus natural to focus on solutions for which this curriculum does not increase the total score-matching loss, which is exactly the mild condition in Eq. (65).
>
> We have updated the paper accordingly by introducing **Lemma 6**, clarifying this intuition, and explicitly tying it to the bound in Theorem 4 at the corresponding point in the proof.
>
>     Scalability
>
> We thank the reviewer for raising this important point regarding scalability. We chose to include MOSES rather than GuacaMol based on the reviewer’s earlier suggestion, as both datasets have fewer nodes per graph than SBM, while MOSES contains a substantially larger number of data points and therefore serves as a more demanding large-dataset benchmark.
>
> To further address the reviewer’s concern, we have additionally conducted experiments on the GuacaMol dataset as requested. The results in the **newly added Table 10** confirm that HOG-Diff maintains strong performance, demonstrating that our approach scales well in all cases. We emphasize that our study now covers all standard benchmarks used in de novo molecular generation.
>
> Table E (Table 10). Generation performance on the GuacaMol dataset.
>
> |Model|Val.↑|V.U.↑|V.U.N.↑|KL div↑|FCD↑|
> |-|-|-|-|-|-|
> |DiGress|85.2|85.2|85.1|92.9|68.0|
> |Disco|86.6|86.6|86.5|92.6|59.7|
> |Cometh|98.9|98.9|**97.6**|96.7|72.7|
> |HOG-Diff|**99.1**|**99.1**|97.0|**96.9**|**78.0**|
>
>     empirical evaluation
>
> Unlike common graph benchmarks, for our higher-order guided setting, Planar and Tree graphs are not good candidates for a controlled study. In planar graphs, all nodes lie within cells; whereas in Tree graphs, no nodes belong to any cell. These two extremes do not allow us to meaningfully assess how well a model learns and preserves intermediate higher-order structures, which is the core focus of our method. As a result, the evaluation on these datasets would not provide a fair or informative comparison in our setting.
>
> We appreciate the reviewer’s suggestion regarding validity-based evaluation. Following this guidance, we have added the validity, uniqueness, and novelty (V.U.N.) metrics to our experiments on the SBM dataset. As shown in the **newly added Fig. 3 (left)**, HOG-Diff achieves notably strong performance under these metrics.
>
> Table E (Fig 3, left). Generation results on the SBM dataset
>
> |Method|Deg.↓|Clus.↓|Orb.↓|Spec.↓|Avg.↓|V.U.N.↑|
> |-|-|-|-|-|-|-|
> |GraphRNN|0.0055|0.0584|0.0785|0.0065|0.0372|0.05|
> |GRAN|0.0113|0.0553|0.0540|*0.0054*|0.0315|0.25|
> |EDP-GNN|*0.0011*|0.0552|0.0520|0.0070|0.0288|-|
> |SPECTRE|0.0015|0.0521|**0.0412**|0.0056|*0.0251*|0.53|
> |HSpectre|0.0141|0.0528|0.0809|0.0071|0.0387|0.75|
> |GDSS|0.0212|0.0646|0.0894|0.0128|0.0470|0.05|
> |DiGress|0.0013|**0.0498**|0.0434|0.0400|0.0336|0.74|
> |GruM|0.0015|0.0589|0.0450|0.0077|0.0283|**0.85**|
> |DeFoG|**0.0006**|0.0517|0.0556|*0.0054*|0.0283|0.80|
> |HOG-Diff|0.0028|*0.0500*|*0.0428*|**0.0043**|**0.0249**|*0.83*|
>
>     MOSES experimental details
>
> We thank the reviewer for raising this point. We are indeed aware that using the `test` split can lead to noticeably higher performance. To ensure a fair comparison with prior graph-based generative models such as DiGress and DeFoG, we follow the same evaluation protocol and use the `test_scaffolds` split for MOSES. We understand the reviewer’s concern regarding evaluation fairness, and we would like to emphasize that we have devoted considerable effort to ensuring a rigorous and fair comparison.
> We have clarified this setting in the revised manuscript.
>
> We strongly believe that we have addressed all the concerns of the reviewer, given our very comprehensive manuscript involving numerous empirical studies as well as theoretical exposition.

---

### Official Review · Reviewer_J3EW · 2025-11-03

**Soundness:** 3
**Presentation:** 3
**Contribution:** 3
**Rating:** 6
**Confidence:** 3

**Summary:**

The paper proposes HOG-Diff, a diffusion-based graph generation framework that explicitly incorporates higher-order topological structures (e.g., cell complexes) through a coarse-to-fine generation curriculum. The method leverages generalized Ornstein-Uhlenbeck (GOU) bridges in the spectral domain and demonstrates superior performance on molecular and generic graph generation tasks, supported by theoretical guarantees.

**Strengths:**

Introduces a Higher-Order Guided Diffusion model, which for the first time explicitly incorporates higher-order structures (e.g., cell complexes) into graph generation tasks, addressing the limitations of existing diffusion models in capturing graph topological properties.

Designs a Cell Complex Filtering (CCF) operation to enable a coarse-to-fine generation process, simplifying the modeling of complex graph distributions.

Achieves  superior results on molecular datasets (QM9, ZINC250k) and generic graph datasets.

**Weaknesses:**

The proposed coarse-to-fine process and cell complex filtering could be computationally expensive, especially for large graphs or dense higher-order structures. No runtime is provided.

Although the authors emphasize interpretability as a motivation, there are few visual or quantitative analyses explaining what “higher-order topology” the model actually captures or how it improves the generation quality.

**Questions:**

Can the proposed filtering and bridge approach scale to larger graphs?

How sensitive are results to the number of hierarchical stages？

---

> ### Author Response · Authors · 2025-11-21
>
> We thank the reviewer for the thoughtful, constructive, and positive assessment. We address the specific concerns below.
>
>     W1: The proposed coarse-to-fine process and cell complex filtering could be computationally expensive, especially for large graphs or dense higher-order structures. No runtime is provided.
>
> We thank the reviewer for raising this concern.
> In HOG-Diff, the only additional computation compared to standard graph diffusion models is a one-time CPU preprocessing step that constructs the higher-order skeletons via cell-complex filtering. As detailed in the revised Appendix D, this filter does not enumerate all higher-order cells, but simply checks whether a node or edge belongs to the closure of some $p$-cell, which keeps the preprocessing lightweight.
>
> To provide concrete evidence, we have added an empirical study of the lifting time across datasets of increasing size (see the **newly added Fig. 8** in Appendix D). The corresponding wall-clock times are summarized in the following Table. On the full MOSES benchmark with over $1.9$M molecules, the lifting operation takes around 15 minutes. While this is higher than on smaller datasets, it is still a one-time preprocessing cost that is negligible compared to the overall data processing and training time.
>
> Table A (Fig. 8). Lifting time.
>
> |Dataset|Lifting time|
> |-|-|
> |Ego-small|0.0349 s|
> |Community-small|0.0665 s|
> |Enzymes|0.2983 s|
> |SBM|1.971 s|
> |QM9|22.0843 s|
> |ZINC250k|2 min 44 s|
> |MOSES|15 min 12 s|
>
> After this lifting step, both training and sampling follow the same GPU-bound loop as standard diffusion models. The score network and the per-step operations are identical in complexity to GDSS-style models. In practice, we observe no measurable slowdown in training relative to GDSS or DiGress, and the sampling time of HOG-Diff remains on par with, or even faster than, strong baselines.
>
> Table B (Table 5). Sampling time(s).
>
> |Method|Community-small|Enzymes|Ego-small|
> |-|--|-|-|
> |GDSS|41|110|28|
> |DiGress|8|301|13|
> |HOG-Diff|11|26|15|
>
> Together, the theoretical complexity analysis in Appendix D and the new lifting time measurements in **Fig. 8** demonstrate that the additional overhead introduced by higher-order guidance is modest and that the lifting operation remains practical even at scale.
>
>
>     W2: Although the authors emphasize interpretability as a motivation, there are few visual or quantitative analyses explaining what “higher-order topology” the model actually captures or how it improves the generation quality.
>
> We thank the reviewer for pointing out the need for clearer evidence of what higher-order topology the model captures.
>  In the revised manuscript, we now provide both visual and quantitative analyses.
>
> First, **the newly added Fig. 3 (right)** now visualizes generative trajectories on QM9, ZINC250k, and MOSES, showing that HOG-Diff maintains higher-order structural integrity throughout the intermediate diffusion steps, in contrast to baselines where topology quickly deteriorates under noise.
>
> Second, we added **a new Section 4.3**, Topological Preservation Analysis, where we quantitatively evaluate higher-order structural fidelity using (1) Balanced Forman–Ricci Curvature $\kappa_{FR}$ and (2) Ollivier–Ricci Curvature $\kappa_{OR}$ [1]. These metrics provide principled, geometry-aware measures of higher-order organization. We compute curvature-based distances between the curvature distributions of generated graphs and those of the test data. The results, summarized in the **newly added Table 3** and **Fig. 1 (right)**, show that HOG-Diff consistently achieves the lowest curvature distances across both molecular and generic graph benchmarks, indicating superior preservation of higher-order organization compared to state-of-the-art baselines.
>
> Table C (Table 3). Quantitative evaluation of higher-order topology preservation using Curvature Filtrations.
>
> |Dataset|Method|Forman|Ollivier|
> |-|-|-|-|
> |QM9|GDSS|0.925|0.601|
> |QM9|DiGress|0.251|0.343|
> |QM9|DoFoG|0.177|0.286|
> |QM9|Cometh|0.216|0.314|
> |QM9|HOG-Diff|0.077|0.098|
> |ZINC250k|MiCaM|9.436|7.251|
> |ZINC250k|GDSS|1.781|1.331|
> |ZINC250k|DoFoG|0.728|0.498|
> |ZINC250k|HOG-Diff|0.190|0.098|
> |SBM|DoFoG|3.998|5.206|
> |SBM|Cometh|2.748|3.938|
> |SBM|HOG-Diff|1.453|3.740|
> |Community-small|GDSS|2.311|1.301|
> |Community-small|HOG-Diff|1.679|1.042|
> |Enzymes|GDSS|13.031|12.399|
> |Enzymes|HOG-Diff|12.114|9.830|
> |Ego-small|GDSS|2.311|1.301|
> |Ego-small|HOG-Diff|1.679|1.042|
>
> Taken together, the curvature-based distances (**Tab. 3, Fig. 1 right**) and the trajectory visualizations (**Fig. 3 right**) provide both qualitative and quantitative evidence that HOG-Diff captures and preserves higher-order topology, and that this preservation underlies its performance gains.
>
> [1] J. Southern et al. Curvature Filtrations for Graph Generative Model Evaluation. NeurIPS 2023.

---

> ### Author Response · Authors · 2025-11-21
>
> ```
> Q1: Can the proposed filtering and bridge approach scale to larger graphs?
> ```
>
> We sincerely appreciate this constructive question regarding the scalability of our approach. To further demonstrate scalability, we have added experiments on larger-graph benchmarks.
>
> First, the SBM dataset (**Fig. 3, left**) contains substantially larger node counts than the other graph benchmarks considered, and HOG-Diff operates on it without difficulty. In addition, we now include results on the full MOSES dataset, which contains over 1.9 million molecules, as reported in the **newly added Table 9**.
>
> These experiments show that HOG-Diff can be trained and evaluated on large-scale datasets without introducing a computational bottleneck, indicating that both the filtering procedure and the guided bridge steps scale effectively in practice.
>
> Table D (Table 9). Generation results on the MOSES dataset
>
> |Method|Val.↑|Unique.↑|Nov.↑|Filters↑|FCD↓|SNN↑|Scaf↑|
> |--|-|-|-|-|-|-|-|
> |GraphINVENT|96.4|99.8|-|95.0|1.22|0.54|12.7|
> |DiGress|85.7|100.0|95.0|97.1|1.19|0.52|14.8|
> |DisCo|88.3|100.0|97.7|95.6|1.44|0.50|15.1|
> |Cometh|90.5|99.9|92.6|99.1|1.27|0.54|16.0|
> |HOG-Diff|97.6|100.0|92.7|93.1|1.09|0.51|10.0|
>
>     Q2: How sensitive are results to the number of hierarchical stages?
>
> We appreciate the reviewer’s question about the number of hierarchical stages $K$.
> Conceptually, $K$ controls the granularity of the coarse-to-fine curriculum. We first construct a sequence of higher-order skeletons $\{G_{\tau_K}, \dots, G_{\tau_1}\}$ via cell-complex filtering and then partition the time horizon $[0, T]$ into $K$ windows $[\tau_{k-1}, \tau_k]$. Each window is modeled by a generalized OU bridge that transports the graph from $G_{\tau_k}$ to $G_{\tau_{k-1}}$. Increasing $K$ therefore introduces more intermediate, topologically meaningful “waypoints” along the generative trajectory.
>
> In practice, we treat $K$ as a small discrete hyperparameter and fix $K=2$ across all datasets. On the standard generative benchmarks we consider, cells of dimension $>2$ are extremely sparse (see **Table 10**), and constructing higher‑dimensional cell complexes incurs non‑trivial preprocessing overhead. Moreover, increasing $K$ (i.e., adding extra windows) leads to fragmented guidance and little additional structural information. Consequently, training signals per window diminish and sample quality saturates.
>
> Empirically, we observe that $K=2$ already captures the vast majority of meaningful higher‑order structure in these datasets, provides sufficient guidance, and yields competitive results on these datasets.
> That said, our framework is dimension-agnostic and can seamlessly incorporate larger K when the data warrant it.
> We agree that for large-scale graphs, where richer and more prevalent higher-order structures are expected, using a larger $K$ is a promising ability in our method and worth exploring.
>
> Table E (Table 10). Average counts of higher-order structures across datasets. “–” indicates computation exceeded 10h.
>
> |Dataset|2-simplices|3-simplices|4-simplices|2-cells|3-cells|
> |-|-|-|-|-|-|
> |Community-small|27.45|8.24|0.76|40.92|0.00|
> |Ego-small|4.61|1.64|0.35|5.66|0.00|
> |Enzymes|25.93|3.05|0.01|-|-|
> |SBM|441.25|90.88|4.42|-|-|
> |QM9|0.47|0.00|0.00|1.84|0.10|
> |ZINC250k|0.06|0.00|0.00|2.77|0.00|
>
> We hope our responses address all of the reviewers' concerns. If any additional questions arise, we would be glad to provide further clarification. We also hope that these efforts provide sufficient reason to raise the score.

---

### Author Response · Authors · 2025-11-24

We thank all reviewers for their valuable comments and time.
We are encouraged that the reviewers found our approach novel (**zqT6**, **NCCR**, **ftpE**), interesting (**ftpE**), and refreshing (**zqT6**), while praising its solid mathematical formulation and competitive results (**all reviewers**).
At the same time, the reviewers highlighted the need for stronger empirical evidence regarding scalability and topological preservation, as well as clearer notation and broader baseline comparisons.
In response, we have performed extensive additional experiments and revisions:


- *Quantitative and Visual Topological Analysis* (**J3EW**, **ftpE**).
To address concerns regarding topological fidelity, we introduced Curvature Filtration metrics (Balanced Forman-Ricci and Ollivier-Ricci) in **the newly added Section 4.3**. The new results in [*Fig. 1 (right)*](https://anonymous.4open.science/r/HOG-Diff-anonymous-16AF/figs/Fig%201.png) and [*Table 3*](https://anonymous.4open.science/r/HOG-Diff-anonymous-16AF/figs/Table%203.png) show that HOG-Diff consistently preserves higher-order structures better than baselines. We also added generative trajectory visualizations in [*Fig. 3 (right)*](https://anonymous.4open.science/r/HOG-Diff-anonymous-16AF/figs/Fig%203.png) to explicitly showcase how our model maintains topological integrity throughout the intermediate diffusion steps.

- *Scalability and Computational Efficiency* (**J3EW**, **zqT6**, **ftpE**).
We demonstrated the scalability of our method through three complementary evaluations.
(1) We conducted a detailed theoretical complexity analysis of the novel lifting operation in Appendix D and report the runtime in [*Fig. 8*](https://anonymous.4open.science/r/HOG-Diff-anonymous-16AF/figs/Fig%208.png), verifying that this one-time preprocessing step remains lightweight and computationally efficient even on large datasets.
(2) We compared the training and sampling time of our framework with prior methods, presented in [*Tables 5*](https://anonymous.4open.science/r/HOG-Diff-anonymous-16AF/figs/Table%206.png) and [*6*](https://anonymous.4open.science/r/HOG-Diff-anonymous-16AF/figs/Table%206.png), respectively.
(3) We added experiments on two large-scale benchmarks, MOSES and GuacaMol, as summarized in the newly added [*Tables 9*](https://anonymous.4open.science/r/HOG-Diff-anonymous-16AF/figs/Table%209.png) and [*10*](https://anonymous.4open.science/r/HOG-Diff-anonymous-16AF/figs/Table%2010.png) (**zqT6**).
We emphasize that our study now includes extensive empirical results covering nearly all datasets commonly used in de novo molecular generation.
Additionally, we incorporated the recent strong baseline, Cometh, into molecular generation comparisons, as suggested (**zqT6**).
Across all evaluations, HOG-Diff achieves strong performance. Taken together, these results demonstrate that scalability and computational cost are not bottlenecks for our method.

- *Presentation and Theoretical Strengthening*.
We thoroughly revised the manuscript to harmonize notation (**zqT6**), rewrote the Introduction for clarity (**zqT6**, **NCCR**), and expanded the discussion of limitations and future work (**NCCR**).
In addition, we generalized the proof of Theorem 4, demonstrating that the tighter error bound holds without relying on a simplified assumption (**zqT6**).


In addition to the point-by-point responses here, we used the additional page allowed in the revised manuscript to incorporate these changes. All significant changes are marked in blue in both the main paper and the Appendix for ease of reference.


We thank the reviewers again for their time and valuable feedback. We believe that the strengthened theory, new topological evaluations, expanded large-scale experiments, and clarified exposition address the concerns raised in the reviews, and we kindly ask the reviewers to reconsider their assessments in light of this revised version. We are fully available for discussion should any concerns remain.

---

### Meta-Review · Area_Chair_wAiG · 2026-01-01

**Summary:**

The reviewers initially raised concerns regarding the computational cost of the proposed framework, the motivation for modeling higher‑order topological structures, the soundness of certain theoretical assumptions, the role of hierarchical time windows, and the overall clarity of the paper. I believe these concerns have been largely addressed during the rebuttal phase through additional experiments, revised theoretical analysis, and improved explanations. The new empirical results demonstrate acceptable computational efficiency, justify the use of higher‑order relations, and clarify key design choices. As a result, I am inclined to support acceptance of the paper.

**Reviewer Concerns:**

Computational cost.
The reviewers were concerned about the computational overhead of components such as cell complex filtering. The authors provided additional experimental evidence in the rebuttal showing that the runtime cost is acceptable, which adequately addresses this concern.

Higher‑order topology motivation.
The necessity of modeling higher‑order topological structures was initially unclear. The authors’ explanations and additional curvature‑based experiments help justify the importance of capturing higher‑order relations.

Theoretical issues.
Concerns were raised about assumptions in the theoretical derivations. The authors revised and clarified the theoretical analysis in the rebuttal, which helps resolve these issues.

Hierarchical time windows.
The role of hierarchical time windows was insufficiently explained. The authors added parameter experiments to demonstrate their effect and effectiveness.

Writing and clarity.
The paper initially suffered from clarity and presentation issues. The authors improved the exposition in the rebuttal, which addresses the reviewers’ concerns.

**Reviewer Scores:**

For Reviewer J3EW and ftpE, they have already expressed positive opinions, and I don't think they will change the score.
For Reviewer NCCR, the authors' response is very thorough and convincing. I think he will raise the score to 6.
For Reviewer zqT6,  author's reply is very sincere. However, due to the lack of sufficient discussion, I am not sure whether the reviewer will recognize the author's response, especially in terms of theory. He might change the score to 4 or remain the same, and there is also a small probability that it will be raised to 6.
Overall, I think the overall evaluation of this paper will be positive, but it may be controversial. Therefore, I recommend accepting it, but I don't mind rejecting this paper either.

---

### Decision · Program_Chairs · 2026-01-26

Accept (Poster)